# Morphological and Molecular Diversity among Pin Nematodes of the Genus *Paratylenchus* (Nematoda: Paratylenchidae) from Florida and Other Localities and Molecular Phylogeny of the Genus

**DOI:** 10.3390/plants12152770

**Published:** 2023-07-26

**Authors:** Sergio Álvarez-Ortega, Sergei A. Subbotin, Koon-Hui Wang, Jason D. Stanley, Silvia Vau, William Crow, Renato N. Inserra

**Affiliations:** 1Departamento de Biología y Geología, Física y Química Inorgánica, Universidad Rey Juan Carlos, Campus de Móstoles, 28933 Madrid, Spain; 2Plant Pest Diagnostic Center, California Department of Food and Agriculture, 3294 Meadowview Road, Sacramento, CA 95832, USA; sergei.a.subbotin@gmail.com; 3Center of Parasitology of A.N. Severtsov Institute of Ecology and Evolution of the Russian, Academy of Sciences, Leninskii Prospect 33, 117071 Moscow, Russia; 4Department of Plant and Environmental Protection Sciences, University of Hawaii at Manoa, 3050 Maile Way, Honolulu, HI 96822, USA; koonhui@hawaii.edu; 5Florida Department of Agriculture and Consumer Services, DPI, Nematology Section, P.O. Box 147100, Gainesville, FL 32614-7100, USA; jason.stanley@fdacs.gov (J.D.S.); silvia.vau@fdacs.gov (S.V.); renato.inserra@fdacs.gov (R.N.I.); 6Department of Entomology and Nematology, University of Florida, P.O. Box 110620, Gainesville, FL 32611-0620, USA; wtcr@ufl.edu

**Keywords:** D2–D3 of 28S rRNA gene, ITS rRNA gene, *COI* mtDNA gene, molecular analysis, morphology, morphometrics, new species, phylogeny, taxonomy, systematics

## Abstract

Pin nematodes (*Paratylenchus* spp.) are root parasites found worldwide. They have different life cycles and feeding habits and can damage a wide range of plants. A remarkable diversity of pin nematode species was found in soil samples from Florida and other states of the USA, Canada, and Spain. Using integrative taxonomy, two new species (*Paratylenchus hawaiiensis* sp. n. and *P. roboris* sp. n.), six valid species (*Paratylenchulus acti*, *P. aquaticus*, *P. goldeni*, *P. paralatescens*, *P. minutus* (=*P. shenzhenensis* syn. n.), and *P. straeleni*), and two undescribed species were identified from Florida; *P. goldeni*, *P. hamatus*, *P. hamicaudatus*, *P. holdemani*, and *P. pedrami* were found in California, *P. minutus* in Hawaii, *P. goldeni* in Oregon and Washington, and one new species, *Paratylenchus borealis* sp. n., in Alaska. Outside the USA, *Paratylenchus projectus* was detected in samples from Canada and Spain as well as *P. holdemani* and *Paratylenchus* sp. from Spain. The pin nematode species from Belgium and Russia identified in former studies as *Paratylenchus* sp. F was herein described as a new species with the name of *P. borealis* sp. n., using a population from Alaska. Previously reported molecular type A of *P. aquaticus* from Hawaii was reclassified as *P. hawaiiensis* sp. n., using a population from Florida. *Paratylenchus roboris* sp. n. from Florida has obese sedentary females with a stylet 63–71 µm long. The results of the molecular analysis of *P. shenzhenensis* from Florida and China indicated that it was conspecific with *P. minutus* from Hawaii and considered here as its junior synonym. New 26 D2–D3 expansion segments of 28S rRNA, 17 ITS rRNA, and 20 *COI* gene sequences were obtained in this study. Phylogenetic relationships of *Paratylenchus* are reconstructed using the D2–D3 of 28S rRNA, ITS rRNA, and *COI* gene sequences. Congruence of molecular and morphological evolution and species identification problems are discussed. Obese females were found in two major clades of *Paratylenchus*. The problem of reference materials is discussed, and it is proposed to make more efforts to collect topotype materials of known *Paratylenchus* species for molecular study.

## 1. Introduction

Pin nematodes (*Paratylenchus* spp.) have not been the object of many studies in Florida because their economic relevance in agriculture has been considered negligible. No representatives of this genus were included in a recent revision of plant parasitic nematodes occurring in Florida’s agriculture [1]. Christie [2] did not rank pin nematodes as significant damaging parasites in Florida and reported only an undetermined species causing root lesions on St. Augustine grass (*Stenotaphrum secundatum* (Waltz.) O. Kuntze) in central Florida. Esser [3] and Lehman [4] listed 24 and 36 species, respectively, from Florida (Appendix A). However, these records do not contain morphological descriptions, morphometrics, or illustrations of the identified samples, and, therefore, their identity remains uncertain. Morphological and morphometrical records without molecular data are available for populations of *P. latescens* (Raski, 1976) Siddiqi, 1986 [5,6] and *P. minusculus* Tarjan, 1960 [7], both detected in central Florida. The former was detected on timber bamboo (*Phyllostachys bambusoides* Siebold. & Zucc.) [8,9], and the latter was described as a new species from live oak (*Quercus virginiana* Mill.) [7]. Other species described in Florida include *Paratylenchus elachistus* Steiner, 1949 [10] and *Paratylenchus elegans* (Raski 1962) Siddiqi & Goodey, 1964 [11,12]. The former was found in roots of ramie (*Boehmeria nivea* (L.) Gaud) and the latter was associated with magnolia (*Magnolia grandiflora* L.) roots in south and central Florida, respectively. More recently, a pin nematode population from an unknown host was identified molecularly as *P. shenzhenensis* Wang, Xie, Li, Xu, Yu & Wang, 2013 [13] and reported in central Florida without morphological data [14]. There is a need to obtain reliable taxonomical information and molecular characteristics of pin nematodes in Florida for a better knowledge of nematode biodiversity in this state’s agriculture and environment.

A pin nematode survey was conducted in 2019–2022 in production sites of ornamental plants and botanical gardens in north, central, and south Florida. Additional populations of pin nematodes were collected concomitantly from Canada, Spain, and other states of the United States (Alaska, California, Hawaii, Oregon, and Washington) to study the morphological and phylogenetic relationships among these populations from different geographical areas. In this manuscript, we provide (i) a description of three new species; (ii) a revision of the taxonomic status and morphological descriptions of some known species; (iii) molecular characterization of the newly obtained populations of Paratylenchus, using sequences of the D2–D3 expansion segments of the 28S rRNA, the ITS rRNA, and partial COI mtDNA genes; and (iv) a study of phylogenetic relationships within *Paratylenchus* species using rRNA and *COI* gene sequences.

## 2. Results

### 2.1. Species Identification and Delimitation

Within the studied samples (Table 1), using traditional morphological taxonomic characters integrated with molecular criteria, we distinguished eleven valid species in the sampled localities: *Paratylenchus acti* Eroshenko, 1978 [15]; *P. aquaticus* Merny, 1966 [16]; *P. goldeni* Raski, 1975 [17]; *P. hamatus* Thorne & Allen, 1950 [18]; *P. hamicaudatus* (Cid del Prado Vera & Maggenti, 1988) Brzeski, 1998 [19,20]; *P. holdemani* Raski, 1975 [21]; *P. minutus* Linford, Oliveira & Ishii, 1949 [22]; *P. paralatescens* (Munawar, Cai, Ye, Powers & Zheng, 2018) Munawar, Miao, Castillo, Zheng, 2020 [23,24]; *P. pedrami* Clavero-Camacho, Cantalapiedra-Navarrete, Archidona-Yuste, Castillo & Palomares-Rius, 2021 [25]; *P. projectus* Jenkins, 1956 [26]; and *P. straeleni* (De Coninck, 1931) Ostenbrink, 1960 [27,28]; and three new species: *Paratylenchus borealis* sp. n. from Alaska, *Paratylenchus hawaiiensis* sp. n. from Hawaii and Florida, and *Paratylenchus roboris* sp. n., along with three unidentified species: *Paratylenchus* spp. FL1 and FL2 from Florida and SP1 from Spain. The pin nematode species of the populations from Belgium and Russia identified by Etongwe et al. [29] and Singh et al. [14] as *Paratylenchus* sp. F was herein identified and described as a new species with the name of *P. borealis* sp. n., using a population from Alaska. Previously reported molecular type A of *P. aquaticus* from *Neoregelia* sp. in Hawaii [30] was reclassified here as a new species with the name of *P. hawaiiensis* sp. n., using a population from Florida. Morphological descriptions of unidentified species are not provided due to the reduced numbers of specimens found. A total of ten species (*Paratylenchus hawaiiensis* sp. n., *P. roboris* sp. n., *P. acti*, *P. aquaticus*, *P. goldeni*, *P. paralatescens*, *P. minutus* (=*P. shenzhenensis* syn. n.), *P. straeleni,* and two undescribed species) were found in Florida, along with five species (*P. goldeni*, *P. hamatus*, *P. hamicaudatus*, *P. holdemani*, and *P. pedrami*) from California, three species (*P. holdemani. P. projectus,* and *Paratylenchus* sp.) from Spain, and by one species from Alaska (*P. borealis* sp. n.), Hawaii (*P. minutus*), Oregon (*P. goldeni*), Washington (*P. goldeni*), and Canada (*P. projectus*). The soil samples examined in this study were monospecific. Biological and parasitic habits of some of these species are included in descriptions.

### 2.2. Taxonomic Studies

The morphological and molecular characteristics along with parasitic habits, when possible, of the identified pin nematodes are provided herein.

#### 2.2.1. *Paratylenchus borealis* sp. n.

(Figure 1 and Figure 2 and Table 2). http://zoobank.org/urn:lsid:zoobank.org:act:7BF49655-9C93-4D9A-8083-1AF4EE68F260 (accessed on 18 July 2023).

A population of pin nematodes consisting of vermiform females was found in soil samples collected from Anchorage, Alaska. The population fits morphologically and molecularly an undescribed *Paratylenchus* species indicated as *Paratylenchus* sp. F by Etongwe et al. [29] and Singh et al. [14] and herein is described as a new species.

**Table 2 plants-12-02770-t002:** Morphometrics of *Paratylenchus borealis* sp. n. from Anchorage, Alaska, compared to the populations of *Paratylenchus* sp. F from Belgium [14,29].

Reference	Etongwe et al. [29]	Singh et al. [14]	This Study		Total Range
Population	Zwijnaarde, Belgium	Merendree, Belgium	Alaska, USA (CD3781)		
Character ^a^	♀ (Fixed)	♀ (Fixed)	♀ (Fixed)		♀
			Holotype	Paratypes	
*n*	3	17	1	5	25
L	330–350	300 ± 21.1 (264–339)	320	347.3 ± 13.6 (325.7–362.3)	264–362
Stylet length (St)	27.9–29.7	27.6 ± 1.2 (25.3–29.6)	30.5	31.0 ± 0.5 (30.6–31.6)	25–32
Conus length	19.9–21.8	18.5 ± 0.9 (17.1–20.3)	20.5	20.6 ± 0.7 (20.0–21.8)	17–22
Stylet shaft + knob height	5.6–6.5	-	10.0	10.4 ± 0.7 (9.8–11.6)	5.5–11.5
Knob width	3.9–4.9	3.6 ± 0.3 (3.3–4.1)	4.5	4.7 ± 0.2 (4.5–4.9)	3.5–5.0
Knob height	1.5–1.8	-	2.0	2.2 ± 0.3 (2.0–2.6)	1.5–2.5
DGO	3.6–4.6	-	4.5	4.7 ± 0.2 (4.5–4.9)	3.5–5.0
Median bulb valve length	-	-	6.5	7.5 ± 0.5 (7.0–8.0)	7–8
Median bulb width	-	-	8.0	9.6 ± 0.5 (8.7–10.0)	8–10
Isthmus length	-	-	15.5	14.2 ± 1.3 (13.2–16.5)	13–17
Pharynx length	86.1–91.2	74.8 ± 4.9 (67.7–83.1)	80	85.7 ± 4.0 (80.5–89.1)	68–91
Anterior end to excretory pore (Ep)	71.2–78.0	63.0 ± 5.4 (51.5–70.6)	67	79.1 ± 5.8 (71.0–85.1)	51–85
Max body width	16.7–17.0	14.8 ± 1.1 (13.3–16.6)	14.0	16.4 ± 1.5 (15.0–18.8)	13–19
Body width at vulva	-	-	12.0	13.8 ± 1.0 (13.0–15.5)	13–16
Body width at anus	-	9.0 ± 0.6 (8.0–9.9)	8.5	9.5 ± 0.5 (8.9–10.2)	8–10
Anterior end to median bulb base	-	-	51	58.0 ± 1.5 (56.4–60.3)	56–60
Lateral field width	-	-	3.5	4.4 ± 0.4 (4.0–4.9)	3.5–5.0
Anterior end-vulva distance	-	217 ± 17.2 (217–278)	260	282.3 ± 15.5 (257.4–300.0)	217–300
Vulva-tail terminus distance	-	-	59	65.0 ± 4.2 (60.0–70.2)	60–70
Genital tract length	-	-	89	151.9 ± 53.3 (104.9–223.7)	89–224
Vulva-anus distance	30.0–38.5	-	35	40.4 ± 3.8 (35.3–44.5)	30–45
Tail length	22.3–25.3	23.1 ± 2.4 (20.0–26.5)	22.5	24.6 ± 1.0 (23.0–25.7)	20–27
Stylet base (Stb) to median bulb valve base (v)	-	-	16.0	17.9 ± 0.7 (16.8–18.9)	16–19
PERCENTAGES					
*V*	81–83	82 ± 0.7 (80.9–83.5)	81	81.2 ± 1.6 (79.0–82.8)	79–84
G or T	-	-	28	43.4 ± 13.9 (29.6–61.7)	28–62
Stb-v/St	-	-	52	57.6 ± (54.7–61.7)	52–62
St/L	8.0–9.0	-	9.5	8.9 ± 0.4 (8.4–9.4)	8–9
Ep/L	21.0–22.3	21.0 ± 1.6 (18.6–24.2)	20.9	22.7 ± 1.0 (21.7–24.0)	19–24
RATIOS					
*a*	19.4–20.8	20.3 ± 1.3 (18.4–23.0)	23.1	21.2 ± 1.2 (19.2–22.5)	18–23
*b*	3.7–4.0	4.0 ± 0.3 (3.7–4.6)	4.0	4.0 ± 0.2 (3.8–4.3)	3.5–4.5
*c*	13.8–15.4	12.9 ± 0.8 (11.8–14.0)	14.3	14.1 ± 0.9 (13.2–15.7)	12–16
*c′*	2.1–2.5	2.6 ± 0.2 (2.3–2.9)	2.7	2.6 ± 0.2 (2.2–2.8)	2–3

^a^ All measurements are in micrometers and in the form mean ± standard deviation (range), except the ratios and percentages. Note: *n* = number of measured specimens; *a* = body length/greatest body diameter; *b* = body length/distance from anterior end pharyngeal intestinal valve; *c* = body length/tail length; *c′* = tail length/body diameter at anus or cloaca; L = overall body length; G or T = Genital tract length/body length %; and *V* = distance of anterior body end from the vulva/body length %.

##### Description

*Female*: The body is small and slender, curved ventrad, and open C- to G-shaped body habitus when heat relaxed; the cuticle is finely annulated; the lateral field has four incisures; and the lip region is narrow, usually truncated, sometimes slightly rounded, and bearing very small submedian lobes under LM. Labial framework sclerotization is weak; the pharyngeal region is a typical paratylenchoid type. The stylet is rigid and straight, and the cone is 64–69% of the total stylet length; stylet knobs are rounded; the pharynx occupies almost one-fourth of the total body length; and the dorsal pharyngeal gland opens 4.5–5.0 µm behind stylet knobs. The median pharyngeal bulb is elongated and bearing distinct large valves; the isthmus is slender, surrounded by a nerve ring; the basal bulb is pyriform, and the pharyngeal-intestinal valve is rounded; the excretory pore is situated at the level of the anterior part of the pharyngeal basal bulb; and the hemizonid is situated immediately anterior to the excretory pore. The body is slightly narrower posterior to the vulva; the ovary is outstretched and well developed; the spermatheca and crustaformeria are well developed; the spermatheca is elongated and rounded (11–16 µm long) and filled with sperm; and the vulva has a transverse slit occupying half of the corresponding body width. Vulval lips are not prominent; the advulval flaps are present and very prominent; and the post-uterine sac is present and 6.0–9.5 µm long. The anus difficult to distinguish in some specimens; the tail is slender, conoid, curved ventrad, finely annulated, and tapering gradually to form a bluntly or finely rounded terminus.

*Male*: Not found.

*Juveniles*: Not found.

##### Type Habitat and Locality

The population of this species was collected from a soil sample collected in a park close to the coast area and Westchester Lagoon, in Anchorage, Alaska, USA (61°12′28.8″ N 149°55′19.2″ W).

##### Other Habitats and Localities

Populations of this new species were also found from (i) Blaarmeersen and Merendree, Belgium [29]; (ii) grasses around a beech tree in Merendree, Belgium (51°04′12″ N; 3°34′37″ E) [14]; and (iii) an unknown plant in Olginsky district, Primorsky Krai, Russia [14].

##### Etymology

The specific epithet is derived from the Latin term *borealis* and refers to the occurrence of the species in the Northern Hemisphere.

##### Type Material

The type material was obtained from live specimens that were fixed after morphological examination. The holotype female and thirteen paratype females, mounted on glass slides, are deposited in the nematode collection of the National Museum of Natural Sciences, Madrid, Spain. An additional four paratype females were sent to each of the United States Department of Agriculture Nematode Collection, Beltsville, MD, USA; Plant Protection Service, Wageningen, The Netherlands; and the Nematode Collection of FERA, Sand Hutton YO41 1LZ, UK.

##### Molecular Characterization

One D2–D3 of 28S rRNA, one ITS rRNA, and one *COI* gene sequence were obtained for this species in this study. In the D2–D3 of 28S rRNA (Figure 3) and ITS rRNA (Figure 4) gene trees, the sequences of the Alaska *P. borealis* sp. n. clustered together with other sequences of the species, previously identified as *Paratylenchus* sp. F [14,29], and formed a clade with *P. elachistus *from Belgium. *Paratylenchus borealis* sp. n. differed from *P. elachistus *by 4.2–5.0% (D2–D3 of 28S rRNA), 4.5–5.4% (ITS rRNA), and 8.0–8.4% (*COI*).

##### Diagnosis (Based on All Known Populations) and Relationships

The new species is characterized by the presence of females with four incisures in the lateral field, a short stylet (25–32 µm long), and the presence of prominent advulval flaps. The lip region is truncate or slightly rounded and bearing very small submedian lobes. The excretory pore is situated at the level of the isthmus or the anterior part of the pharyngeal basal bulb. The spermatheca is well developed, elongated and rounded. The tail is slender, conoid, curved ventrad, and finely annulated and gradually tapers to form a bluntly or finely rounded terminus. Males reported by Singh et al. [14] lack a stylet and have spicules about 21.5 µm long. According to the species grouping by Ghaderi et al. [31], the new species belongs to group 3 characterized by a stylet shorter than 40 µm, four incisures in the lateral field, and present advulval flaps.

In having short female stylet length (25–32 µm), very small submedian lobes, the excretory pore situated at the level of the isthmus or the anterior part of the pharyngeal basal bulb, and males lacking stylets, the new species is morphologically and morphometrically nearly identical to *Paratylenchus nanus* Cobb, 1923 [32]. However, *P. borealis* sp. n. differs from *P. nanus* in having females with shorter bodies (264–362 µm) than *P. nanus* populations described by Cobb in the original description (360–410 µm) and in his notes reported by Tarjan [7] for populations from North Dakota and Virginia (390–450 µm). Cobb’s body measurements of *P. nanus* were confirmed by Tarjan [7], who measured original specimens studied by Cobb consisting of one specimen each of *P. nanus* from North Dakota (441 µm) and Virginia (406 µm). The body length of *P. nanus* reported by Subbotin et al. [33] in a recent redescription of *P. nanus* was also greater than that of *P. borealis* sp. n. (356–401 vs. 264–362 µm). Vulval flaps in *P. borealis* sp. n. are extremely prominent, whereas in *P. nanus,* they are present but not prominent. These two species can be easily distinguished by molecular data based on 18S rRNA, D2–D3 of 28S rRNA, ITS rRNA, and *COI* gene sequences. *Paratylenchus borealis* sp. n. also resembles *P. ciccaronei* Raski, 1975 [21], *P. halophilus* Wouts, 1966 [34], *P. lepidus* Raski, 1975 [21], *P. minusculus* Tarjan, 1960 [7], *P. neoamblycephalus* Geraert, 1965 [35], and *P. rotundicephalus* Bajaj, 1988 [36]. It differs from *P. ciccaronei*, a very similar species in its morphometrics, in its lip region usually truncate (vs. rounded), and shorter female tail (22–27 vs. 35 µm) with a bluntly or finely rounded terminus (vs. slight digitate appearance and with a very finely rounded almost acute terminus). It differs from the morphometrically similar *P. halophilus* in that its lip region is usually truncate (vs. rounded), the advulval flaps are very prominent (vs. inconspicuous), and the female tail has a bluntly or finely rounded terminus (vs. finely pointed or digitate terminus). It differs from *P. lepidus* in that its female body is less slender (*a* = 18–23 vs. 23–31), the lip region is usually truncated (vs. rounded), the female tail has a bluntly or finely rounded terminus (vs. subacute terminus), there is a longer spicule (21.5 vs. 16–17 µm), and there are molecular data. It differs from *P. minusculus* in its longer stylet (25–32 vs. 22–27 µm), greater *c* ratio (*c* = 11.8–15.7 vs. 7–12), vulva being more posterior (*V* = 79–84 vs. 69–73%), and longer spicule (21.5 vs. 13–18 µm). It differs from the morphometrically similar *P. neoamblycephalus* in its female tail with a bluntly or finely rounded terminus (vs. subacute terminus) and molecular data. And it differs from *P. rotundicephalus* in its female body being less slender (*a* = 18–23 vs. 23–27), the lip region usually being truncated (vs. rounded), and its longer spicule (21.5 vs. 16–19 µm).

Finally, according to the molecular data, the new species is clustered close to *P. elachistus* Steiner, 1949 [10], but it can be distinguished from this species by longer stylets (25–32 vs. 19–25 µm), female tails with a bluntly or finely rounded terminus (vs. terminus from spicate to pointed or minutely rounded terminus), and molecular data.

##### Remarks

The new population from Alaska fits morphologically, morphometrically, and molecularly very well with populations from distant geographical areas such as Belgium and Russia previously identified as *Paratylenchus* sp. F by Etongwe et al. [29] and Singh et al. [14].

#### 2.2.2. *Paratylenchus hawaiiensis* sp. n.

(=*P. aquaticus* Type A *apud* Van den Berg et al. [30]) (Figure 1 in Van den Berg et al. [30]) (Figure 5 and Figure 6, Table 3). http://zoobank.org/urn:lsid:zoobank.org:act:FCBA10A0-D104-4D15-8278-0791F193A40C (accessed on 18 July 2023).

The Hawaiian population defined by Van den Berg et al. [30] as *P. aquaticus* type A shows remarkable morphological differences from *P. aquaticus sensu lato*, such as a lip region with small and distinct submedian lobes (vs. lacking submedian lobes), a longer stylet length (21 ± 1 (20–23) vs. up to 20 µm), longer stylet conus length (13 ± 0.8 (12–14) vs. up to 11 µm), and the female tail usually ending in a small mucro (vs. lacking a small mucro on tail tip). Moreover, the molecular data, based on sequences of the D2–D3 expansion segments of the 28S rRNA, the ITS of the rRNA, and partial *COI* mtDNA genes, confirm that *P. aquaticus* type A and *P. aquaticus sensu lato* are not conspecific, and, therefore, *P. aquaticus* type A [30] is herein regarded as a new species, and it is named *Paratylenchus hawaiiensis* sp. n.

The population (sample CD619) herein named *P. hawaiiensis* sp. n. was collected from bromeliad in Waimanalo, Hawaii, and initially was identified as *P. aquaticus* by Van den Berg et al. [30]. *Paratylenchus hawaiiensis* sp. n. specimens fitting morphologically and molecularly those of the population from Hawaii were also found in a plant shipment of *Aechmea* sp., from a nursery, Princeton, Florida, USA, which was intercepted by the CDFA. In this paper, we described these Florida specimens as a new species based on morphological and molecular datasets.

##### Description

*Female*: The body is small and slender and curved ventrad when heat relaxed; the cuticle is finely annulated; the lateral field has three incisures (two bands); and the lip region is flattened, with small to distinct submedian lobes (under LM). Labial framework sclerotization is weak; the pharyngeal region is a typical paratylenchoid type. The stylet is rigid and straight, and the cone makes up *ca* 60% of the total stylet length; stylet knobs are rounded; and the pharynx occupies almost one-fourth of the total body length. The median pharyngeal bulb is elongated, bearing distinct large valves; the isthmus is slender, surrounded by a nerve ring; the basal bulb is rounded, and the pharyngeal-intestinal valve is rounded; the excretory pore is situated at the posterior part of the isthmus; and the hemizonid is situated immediately anterior to the excretory pore. The body is slightly narrower posterior to the vulva; the ovary is outstretched and well developed; the spermatheca and crustaformeria are well developed; the spermatheca is well developed and rounded or oblong, with rounded sperm cells; and the vulva has a transverse slit occupying more than half of the corresponding body width. The anterior vulval lip is slightly protruding; the advulval flaps are distinct; and the post-uterine sac is absent. The tail is elongate-conoid, gradually tapering to form a rounded terminus usually bearing a small mucro on the tip.

*Male*: Not found.

*Juveniles*: Not found.

##### Type Habitat and Locality

The type population of this new species was collected from soil anchoring bromeliad (*Aechmea* sp.) plants in a nursery, Princeton, FL, USA.

##### Other Hosts and Localities

A population of this new species was also found in soil and feeder roots of bromeliad (*Neoregelia* sp.) in the Hawaiian Sunshine Nursery, Waimanalo, HI, USA.

##### Etymology

The scientific name of this species is derived from the Latin adjective of the geographical name Hawaii, *hawaiiensis*, where this species was first found.

##### Type Material

The type material was obtained from live specimens that were fixed after morphological examination. The holotype female and fifteen paratype females, mounted on glass slides, are deposited in the nematode collection of the National Museum of Natural Sciences, Madrid, Spain. Additionally, four paratype females were sent to each of the United States Department of Agriculture Nematode Collection, Beltsville, MD, USA; WaNeCo, Plant Protection Service, Wageningen, The Netherlands; and the Nematode Collection of FERA, Sand Hutton YO41 1LZ, UK.

##### Molecular Characterization

One new sequence of the D2–D3 expansion segments of the 28S rRNA gene from the Florida sample of this species was obtained in this study. In the D2–D3 of the 28S rRNA gene tree (Figure 3), this species clustered with *P. minutus* and differed from it by 11–12%. Intraspecific variation for *P. hawaiiensis* sp. n. was up to 2.0%. *P. hawaiiensis* sp. n. within the genus was in a basal position and not resolved in the ITS rRNA gene tree (Figure 4).

##### Diagnosis and Relationships

*Paratylenchus hawaiiensis* sp. n. is characterized by the presence of three incisures in the lateral field and females having advulval flaps and stylets 20–25 µm long. The lip region is flattened, with small to distinct submedian lobes. The excretory pore is situated at the posterior part of the isthmus. The spermatheca is well developed and rounded or oblong, and with rounded sperm cells. The tail is elongate-conoid, gradually tapering to form a rounded terminus usually bearing a small mucro on the tip. According to the species grouping by Ghaderi et al. [31], the *P. hawaiiensis* sp. n. belongs to group 2 characterized by a stylet shorter than 40 µm, three incisures in the lateral field, and present advulval flaps.

The new species is morphologically and morphometrically very similar to *P. aquaticus* (see above). Moreover, in having short female stylet length (20–25 µm), a lip region bearing small submedian lobes, the excretory pore situated at the isthmus level, and distinct advulval flaps, *P. hawaiiensis* sp. n. shares with *P. leptos* Raski, 1975 [17] a lateral filed marked by three incisures but differs from this species in having females with smaller ratio c values (12–14 vs. 15–18). This new species resembles *P. holdemani*, *P. microdorus* Andrássy, 1959 [37], *P. minutus*, *P. sheri* (Raski, 1973) Siddiqi, 1986 [6,38], and *P. veruculatus* Wu, 1962 [39], but it differs from all of these species in the presence of three (vs. four) incisures in the lateral field and molecular data based on rRNA and/or partial *COI* mtDNA genes.

#### 2.2.3. *Paratylenchus roboris* sp. n.

(Figure 7, Figure 8, Figure 9, Figure 10A and Figure 11A, Table 4, Table 5 and Table 6). http://zoobank.org/urn:lsid:zoobank.org:act:7009A874-0AE5-45D9-9093-895C0495D720 (accessed on 18 July 2023).

A population of pin nematode consisting of motile vermiform females, males and J2, sedentary swollen females, and quiescent coiled J3 and J4 was extracted from soil and root samples collected from the rhizosphere of live oak trees in a tree nursery in Florida. The population differed morphologically and molecularly from other *Paratylenchus* species and is described as a new species.

**Figure 7 plants-12-02770-f007:**
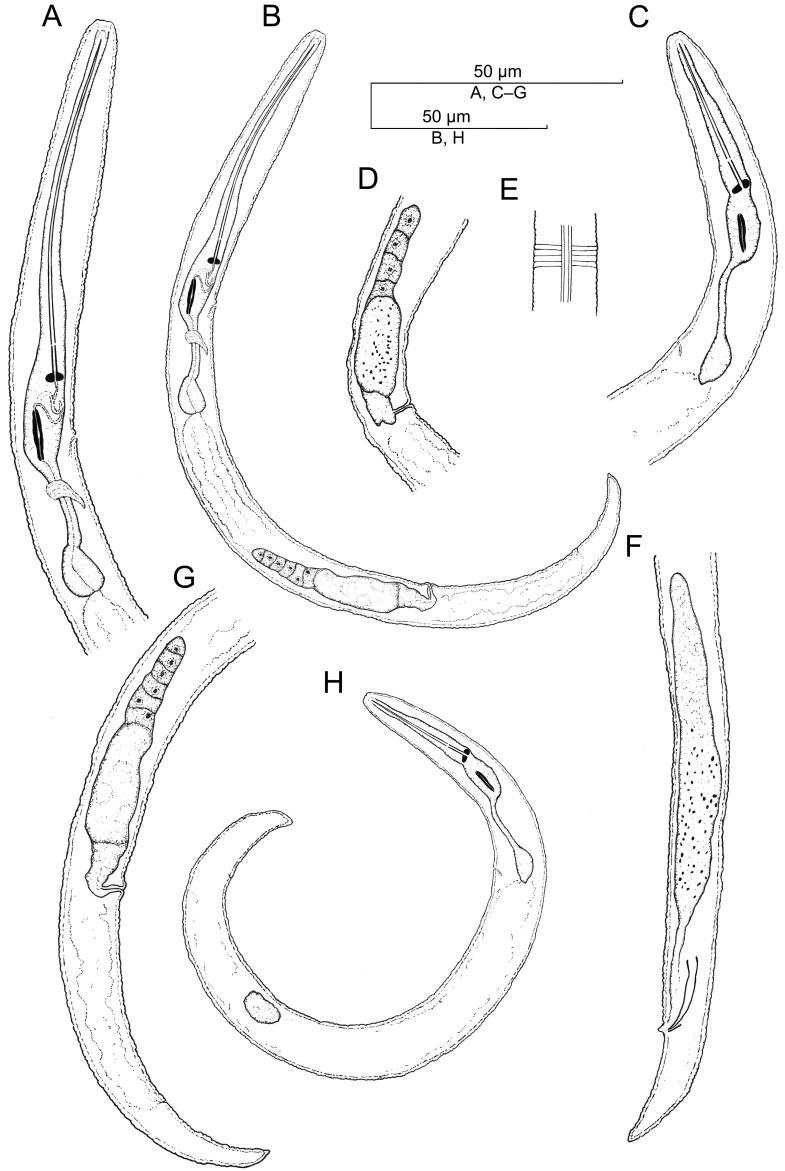
Camera lucida drawings of *Paratylenchus roboris* sp. n. from *Quercus virginiana* in Florida. (**A**): Female pharyngeal region; (**B**): Female entire body; (**C**): Second-stage juvenile pharyngeal region; (**D**): Female vulval region and genital tract; (**E**): Female lateral field marked by four incisures; (**G**): Female posterior body; (**F**): Male posterior body; and (**H**): Second-stage juvenile entire body.

**Figure 8 plants-12-02770-f008:**
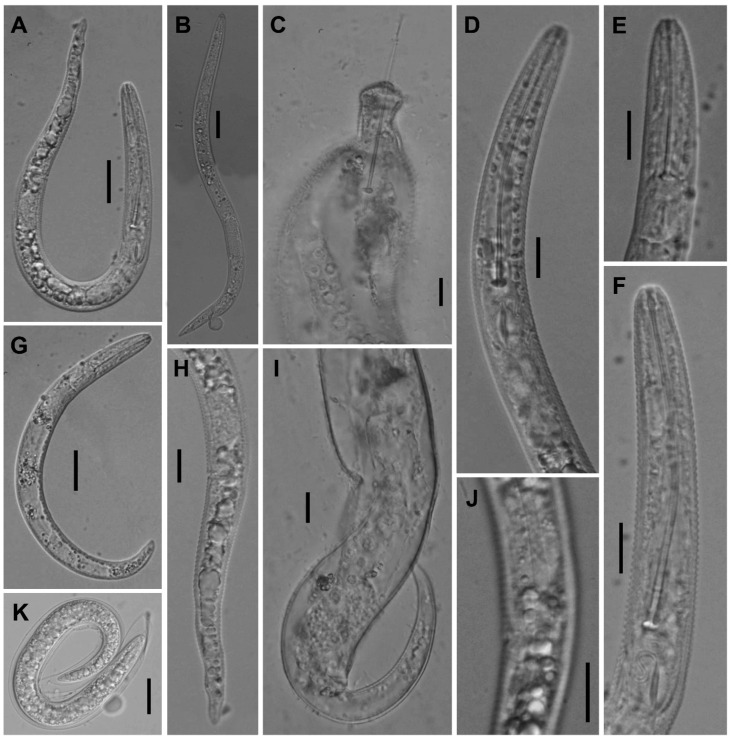
Light microscopic photos of *Paratylenchus roboris* sp. n. female, male, and juveniles (live specimens) from *Quercus virginiana* in Florida. (**A**): Entire body of vermiform female; (**B**): Entire body of male; (**C**): Anterior body of obese female; (**D**,**F**): Anterior body of vermiform female; (**E**): Anterior body of second-stage juvenile; (**G**): Entire body of second-stage juvenile; (**H**,**I**): Posterior body of vermiform and obese female, respectively; (**J**): Lateral field of female; and (**K**): Coiled and non-feeding fourth-stage juvenile encased in the molted cuticles of third- and second-stage juveniles. (Scale bars: (**A**,**B**,**G**) = 20 μm; (**C**–**F**,**H**–**K**) = 10 μm).

**Figure 9 plants-12-02770-f009:**
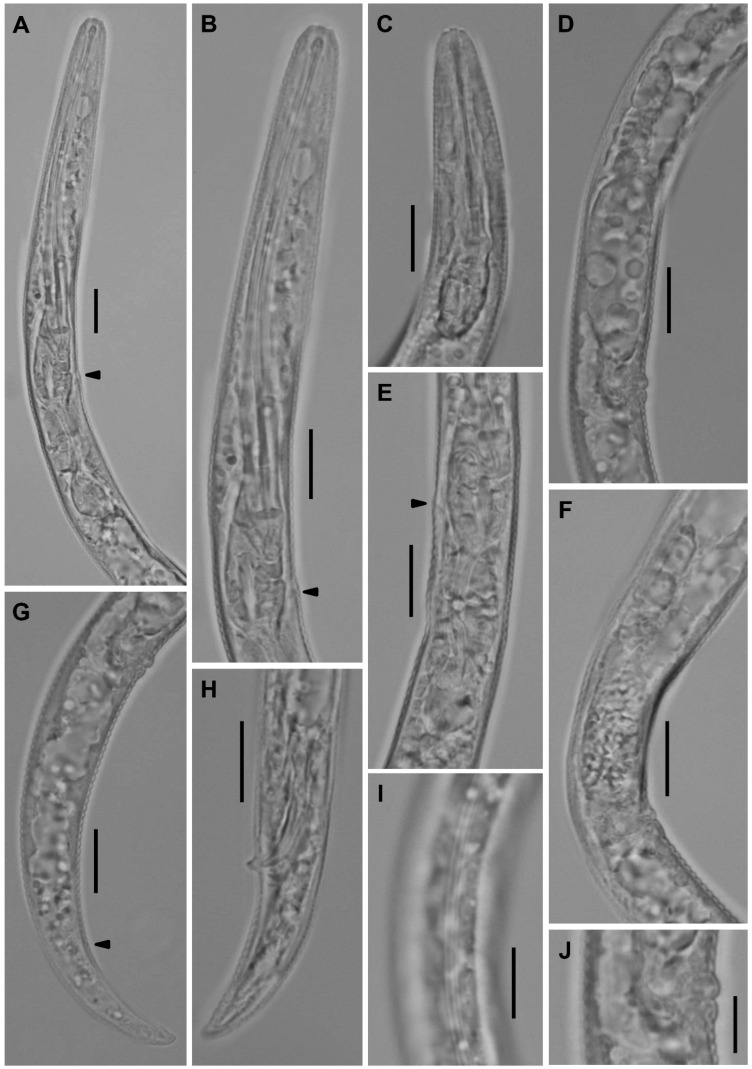
Light microscopic photos of *Paratylenchus roboris* sp. n. vermiform female, male, and second-stage juvenile (fixed specimens) from *Quercus virginiana* in Florida. (**A**): Female pharyngeal region. Note excretory pore (arrowed); (**B**): Female anterior region. Note excretory pore (arrowed); (**C**): Second-stage juvenile pharyngeal region; (**D**): Female genital tract; and (**E**): Female median bulb. Note excretory pore (arrowed); (**F**): Female vulval region showing large uterus and ovary; and (**G**): Female posterior body. Note anus (arrowed); (**H**): Male posterior body; (**I**): Female lateral field marked by four incisures or three bands; and (**J**): Vulval region showing post-vulval uterine sac. (Scale bars: (**A**–**I**) = 10 μm; (**J**) = 5 μm).

**Figure 10 plants-12-02770-f010:**
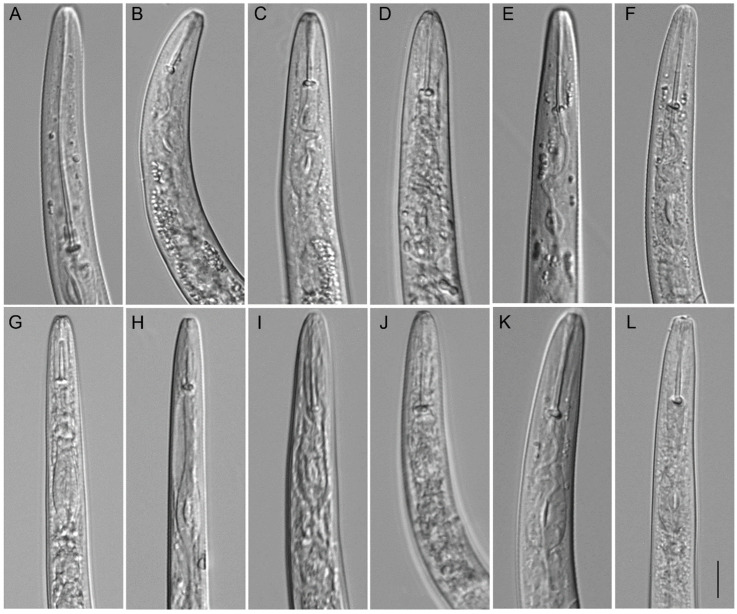
Anterior part of body. Light microscopic photos of *Paratylenchus* species. (**A**): *P. roboris* sp. n. (CD3450); (**B**,**C**): *P. minutus* (CD3465); (**D**): *P. holdemani* (CD3469); (**E**,**F**): *P. goldeni* (CD3470); (**G**,**H**): *P. aquaticus* type C (CD3480); (**I**,**J**): *P. pedrami* (CD3609); (**K**): *P. projectus* (CD3623); and (**L**): *P. hawaiiensis* sp. n. (CD3688). (Scale bar = 10 μm).

##### Description

*Vermiform female*: The body is small and slender, curved ventrad, and has an open C-shaped body habitus when heat relaxed; the cuticle is finely annulated; the lateral field has four incisures; and the lip region is narrow, truncated, and bearing small submedian lobes. Labial framework sclerotization is weak; and the pharyngeal region is a typical paratylenchoid type. The stylet is flexible, elongated, and ventrally curved, and the cone is 88–90% of the total stylet length; stylet knobs are rounded; the pharynx occupies almost two-fifths of the total body length; and the dorsal pharyngeal gland opens 7.0–8.0 µm behind stylet knobs. The median pharyngeal bulb is elongated, bearing distinct large valves; the isthmus is slender, surrounded by a nerve ring; the basal bulb is pyriform, the pharyngeal-intestinal valve is rounded; the excretory pore is situated at the level of the valve of the median pharyngeal bulb or at base of the pharyngeal bulb in 8% of the examined specimens; and the hemizonid is situated immediately anterior to the excretory pore. The body is slightly narrower posterior to the vulva; the ovary is outstretched and well developed; the spermatheca and crustaformeria are well developed; the spermatheca is elongated (15–20 µm long) and rounded; and the vulva has a transverse slit occupying half of the corresponding body width. Vulval lips are slightly prominent, with the anterior lip protruding further than the posterior lip but not prominent in fresh specimens; the advulval flaps are absent; and the post-uterine sac is absent. The anus is difficult to distinguish in some specimens; and the tail is slender, conoid, curved ventrad, finely annulated, and gradually tapers to form a subacute terminus.

*Male*: The body is more slender than the female body, tapering towards both ends, and curved to open C-shaped when heat relaxed. The cuticle is apparently smooth with fine annulations; the labial region is similar to that of females but narrower and slightly truncated, continuous with the body, and sclerotization in the labial region is weak; and the stylet is lacking. The pharynx is rudimentary and non-functional, procorpus, and metacorpus, and basal bulb is inconspicuous; and the excretory pore is located 60–82 µm away from the anterior end. The testis is outstretched, with small spermatozoa; the spicule is slender and slightly curved towards the end; the gubernaculum is curved; and the bursa is absent. The tail is elongate-conoid, tapering gradually to a finely pointed tip.

**Figure 11 plants-12-02770-f011:**
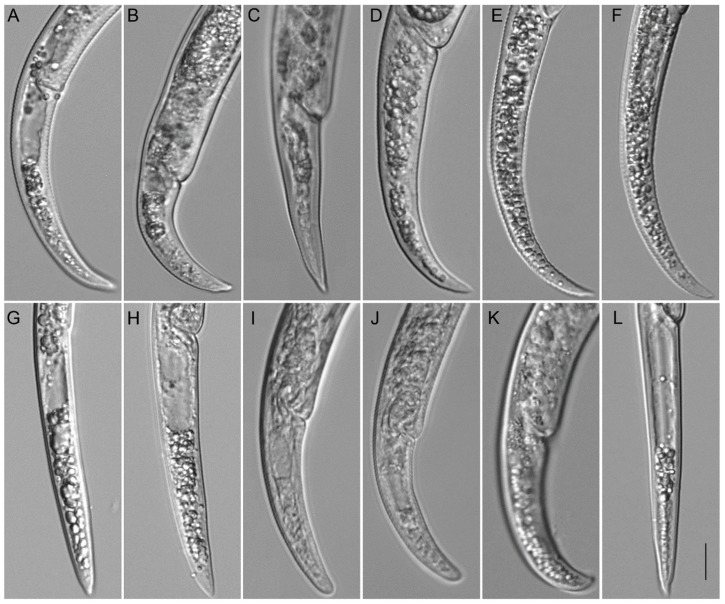
Posterior part of body. Light microscopic photos of *Paratylenchus* species. (**A**): *P. roboris* sp. n. (CD3450); (**B**,**C**): *P. minutus* (CD3465); (**D**): *P. holdemani* (CD3469); (**E**,**F**): *P. goldeni* (CD3470); (**G**,**H**): *P. aquaticus* type C (CD3480); (**I**,**J**): *P. pedrami* (CD3609); (**K**): *P. projectus* (CD3623); and (**L**): *P. hawaiiensis* sp. n. (CD3688). (Scale bar = 10 μm).

**Table 4 plants-12-02770-t004:** Morphometrics of the females of *Paratylenchus roboris* sp. n. from *Quercus virginiana* in Florida.

Population	High Springs, Alachua County (CD3450—N21-00211)				
Character	Vermiform ♀ (Live)	Vermiform ♀ (Fixed)		Total Range Vermiform ♀	Obese ♀ (Dead)
		Holotype	Paratypes		
*n*	7	1	9	17	5
L	262.7 ± 6.7 (257.0–275.2)	283.8	275.4 ± 7.6 (261.0–283.0)	257.0–283.8	266.8 ± 17.7 (243.0–288.0)
Stylet length (St)	65.3 ± 1.1 (63.3–66.8)	71	66.9 ± 2.3 (64.0–70.0)	63.3–71.0	64.0 (n = 1)
Conus length	57.5 ± 0.7 (56.4–58.4)	63	59.4 ± 1.9 (57.0–62.0)	56.4–63.0	-
Stylet shaft + knob height	7.7 ± 1.0 (6.1–9.2)	8	7.4 ± 0.5 (7.0–8.0)	6.1–9.2	-
Knob width	3.9 ± 0.9 (3.5–4.0)	3.5	3.0 ± 0.1 (3.0–3.2)	3.0–4.0	-
Knob height	1.9 ± 0.0 (1.9–2.0)	1.5	1.5 ± 0.0 (1.5–1.5)	1.5–2.0	-
DGO	7.6 ± 0.4 (7.0–8.0)	7.1	7.12 ± 0.13 (7.00–7.30)	7.0–8.0	-
Median bulb valve length	9.8 ± 0.4 (8.9–10.0)	8.5	8.8 ± 0.3 (8.5–9.0)	7.0–10.0	9.0 (n = 1)
Median bulb width	9.7 ± 0.7 (9.0–10.8)	15	14.8 ± 0.4 (14.0–15.5)	8.5–10.8	15.0, 20.0 (n = 2)
Isthmus length	14.1 ± 0.5 (13.6–15.0)	3.5	3.4 ± 0.2 (3.0–3.5)	13.6–15.5	-
Pharynx length	108.6 ± 3.0 (103.9–113.8)	117	108.3 ± 3.4 (102.0–114.0)	102.0–117.0	-
Anterior end to excretory pore (Ep)	74.9 ± 2.3 (72.0–78.2)	83.5	75.8 ± 2.8 (72.0–81.0)	72.0–83.5	-
Max body width	14.5 ± 0.6 (13.3–15.3)	14	14.2 ± 0.6 (13.0–15.0)	13.0–15.3	44.2 ± 11.8 (30.0–65.0)
Body width at vulva	12.6 ± 0.3 (12.2–13.0)	12.5	12.4 ± 0.5 (11.5–13.0)	11.5–13.0	20.0 ± 2.9 (17.0–25.0)
Body width at anus	7.2 ± 0.3 (6.9–7.9)	6	6.2 ± 0.7 (5.5–7.0)	5.5–7.9	-
Anterior end to median bulb base	82.2 ± 3.4 (77.7–89.1)	90	84.3 ± 2.2 (81.0–87.0)	77.7–90.0	-
Lateral field width	2.8 ± 0.2 (2.5–3.0)	1.8	1.8 ± 0.0 (1.8–1.9)	1.8–3.0	4.3 ± 0.4 (4.0–5.0)
Anterior end-vulva distance	189.9 ± 4.1 (186.0–197.0)	211.2	200.9 ± 7.2 (185.0–207.1)	185.0–211.2	199.6 ± 11.9 (252.0–288.0)
Vulva-tail terminus distance	72.9 ± 3.1 (68.3–78.2)	72.7	74.8 ± 4.0 (68.2–82.2)	68.2–82.2	67.2 ± 5.9 (60.0–75.0)
Genital tract length	51.8 ± 3.8 (48.0–60.3)	54.5	50.0 ± 4.8 (45.0–60.0)	45.0–60.3	-
Vulva-anus distance	48.5 ± 5.9 (39.0–57.4)	48	47.2 ± 6.0 (40.0–59.0)	39.0–59.0	-
Tail length	20.7 ± 1.3 (18.8–22.7)	21	20.4 ± 1.3 (19.0–23.0)	18.8–23.0	-
Stylet base (Stb) to median bulb valve base (v)	15.6 ± 1.5 (14.0–17.8)	15	14.2 ± 0.4 (14.0–15.0)	14.0–17.8	-
PERCENTAGES					
*V*	72.2 ± 0.6 (71.4–73.4)	74.4	72.9 ± 1.4 (70.9–74.6)	70.9–74.6	74.1 ± 1.3 (71.7–75.3)
G or T	19.5 ± 1.5 (18.4–22.9)	19.2	18.2 ± 1.9 (15.9–22.5)	15.9–22.9	-
Stb-v/St	23.9 ± 2.6 (21.4–28.1)	21.1	21.3 ± 1.0 (20.0–23.1)	20.0–28.1	-
St/L	24.8 ± 0.9 (23.0–25.9)	25.0	24.3 ± 0.8 (22.6–25.1)	22.6–25.9	-
Ep/L	28.6 ± 1.3 (26.4–30.4)	29.4	27.6 ± 1.2 (25.8–29.1)	25.8–30.4	-
RATIOS					
*a*	18.1 ± 0.8 (17.2–19.3)	20.3	19.5 ± 0.6 (18.7–20.2)	17.2–20.3	6.5 ± 1.6 (4.0–8.5)
*b*	2.4 ± 0.1 (2.2–2.4)	2.4	2.5 ± 0.1 (2.4–2.7)	2.2–2.7	-
*c*	12.7 ± 0.9 (11.9–14.6)	13.5	13.5 ± 1.1 (11.3–14.9)	11.3–14.9	-
*c′*	2.8 ± 0.2 (2.4–3.1)	3.5	3.3 ± 0.3 (2.9–3.8)	2.4–3.8	-

**Table 5 plants-12-02770-t005:** Morphometrics of males of *Paratylenchus roboris* sp. n. from *Quercus virginiana* in Florida.

Population	High Springs, Alachua County (CD3450—N21-00211)		
Character	♂ (Live)	♂ (Fixed)—Paratypes	Total Range ♂
*n*	4	8	12
L	299.0 ± 6.2 (291.0–305.9)	258.5 ± 12.7 (245.0–281.0)	245.0–305.9
Pharynx length	88.1 ± 5.1 (82.1–96.0)	71.6 ± 2.7 (68.0–76.0)	68.0–96.0
Anterior end to excretory pore (Ep)	77.8 ± 2.9 (74.2–82.1)	63.3 ± 2.2 (60.0–66.0)	60.0–82.1
Max body width	12.6 ± 0.4 (12.0–13.0)	10.6 ± 0.6 (9.5–11.0)	9.5–13.0
Body width at anus	10.0 ± 0.5 (9.4–10.8)	7.8 ± 0.5 (7.0–8.0)	7.0–8.0
Genital tract length	105.9 ± 8.8 (96.0–118.8)	100.9 ± 4.6 (96.6–109.2)	96.6–118.8
Tail length	27.2 ± 1.1 (25.7–28.7)	20.8 ± 0.7 (19.5–22.0)	19.5–28.7
Spicule	17.6 ± 0.2 (17.4–17.8)	16.4 ± 0.4 (16.0–17.0)	16.0–17.8
Gubernaculum	4.1 ± 0.1 (4.0–4.2)	3.4 ± 0.2 (3.0–3.5)	3.0–4.2
PERCENTAGES			
G or T	35.4 ± 3.5 (31.5–40.2)	39.6 ± 2.9 (35.4–44.6)	31.5–44.6
Ep/L	26.0 ± 0.6 (25.4–27.0)	24.5 ± 0.9 (23.1–25.6)	23.1–27.0
RATIOS			
*a*	23.7 ± 1.1 (22.3–25.4)	24.5 ± 1.8 (22.3–26.5)	22.3–26.5
*b*	3.4 ± 0.2 (3.1–3.5)	3.6 ± 0.1 (3.4–3.7)	3.1–3.7
*c*	11.0 ± 0.3 (10.6–11.3)	12.4 ± 0.6 (11.7–13.3)	10.6–13.3
*c′*	2.7 ± 0.2 (2.4–3.0)	2.7 ± 0.2 (2.4–3.0)	2.4–3.0

*Obese female*: The description of the morphology of this life stage is incomplete because we were not able to detect females attached to the roots. The partially described specimens were damaged during their extraction with pressurized water from the roots because their stylet is cemented in the root tissues and breaks inside the tissues when specimens are detached. This life stage is characterized by a lemon- or reniform-shaped body with the anterior portion often bent and the posterior portion after the vulva narrow and projecting like an opened sickle. The body is marked by annuli 1.0–1.5 µm wide and a lateral field 4.0 µm wide with indistinct inner lateral incisures. The cuticle is 3.5 µm thick. The undamaged stylet was 64 µm long. The median pharyngeal bulb was enlarged (15–25 µm wide and 20 µm long); the isthmus and basal bulb were missing in our specimens. The genital tract consisted of a very enlarged and coiled ovary occupying almost the entire body. The uterus was enlarged, containing segmented eggs 40 µm long and 20 µm wide. We were not able to detect egg masses.

*Juveniles*: Second-stage juvenile (J2) is similar in morphology to the adult females. It is characterized by a well-developed stylet. The pharynx shows a prominent median pharyngeal bulb with a valve 4.5 µm long. The isthmus and basal bulb (8.5 µm wide and 14 µm long) are visible in live specimens. The excretory pore opening is at the level of the isthmus. the genital primordium is undeveloped; the anus and rectum are undiscernible in some specimens; and posterior body has a rounded terminus. Third (J3) and fourth (J4) juvenile stages are coiled, dark, and without a discernable stylet, pharynx, and gonads. These non-feeding resistant stages retain the molted cuticles that, in some specimens, show the attached molted stylet of the J2. They are commonly found in the soil to resume their development into adults under favorable environmental conditions.

##### Type Habitat and Locality

The population of this species was collected from soil and feeder roots of live oak (*Quercus virginiana* Mill.) in a tree farm in Alachua, FL, USA (latitude 29°90′83″ N, longitude 82°61′11″ W).

**Table 6 plants-12-02770-t006:** Morphometrics of second-stage juveniles of *Paratylenchus roboris* sp. n. from *Quercus virginiana* in Florida.

Population	High Springs, Alachua County (CD3450—N21-00211)		
Character	J2 (Live)	J2 (Fixed)	Total Range J2
*n*	6	4	10
L	210.2 ± 7.1 (200.0–219.7)	251.8 ± 5.5 (246.8–258.3)	200.0–258.3
Stylet length (St)	32.9 ± 0.7 (32.1–34.1)	31.8 ± 0.3 (31.5–32.0)	31.5–34.1
Conus length	27.1 ± 0.8 (26.5–28.8)	26.3 ± 0.3 (26.0–26.5)	26.0–28.8
Stylet shaft + knob height	5.8 ± 0.5 (5.3–6.6)	5.5 ± 0.0 (5.5–5.5)	5.3–6.6
Knob width	3.8 ± 0.9 (3.5–3.9)	3.5 ± 0.0 (3.5–3.5)	3.5–3.9
Knob height	1.7 ± 0.1 (1.5–1.9)	1.5 ± 0.0 (1.5–1.5)	1.5–1.9
DGO	4.8 ± 0.1 (4.7–5.0)	4.5, 4.7 (n = 2)	4.5–5.0
Median bulb valve length	4.9 ± 0.1 (4.7–5.0)	5.0 ± 0.0 (5.0–5.0)	4.7–5.0
Median bulb width	8.4 ± 0.5 (7.5–8.9)	6.5 ± 0.5 (6.0–7.0)	6.0–8.9
Isthmus length	12.3 ± 2.3 (8.0–15.5)	12.7 ± 3.2 (9.0–15.0)	8.0–15.5
Pharynx length	84.1 ± 4.1 (79.0–91.5)	73.3 ± 0.5 (73.0–74.0)	73.0–91.5
Anterior end to excretory pore (Ep)	65.5 ± 3.5 (60.0–70.3)	63.0 ± 1.0 (62.0–64.0)	60.0–70.3
Max body width	13.4 ± 0.3 (13.0–13.9)	15.4 ± 0.3 (15.0–15.5)	13.0–15.5
Body width at anus	8.3 ± 0.3 (7.8–8.8)	8.6 ± 0.3 (8.5–9.0)	7.8–9.0
Anterior end to median bulb base	56.0 ± 3.4 (50.0–61.3)	48.4 ± 1.3 (47.0–50.0)	47.0–61.3
Genital primordium length	12.6 ± 1.9 (10.0–15.0)	11.5, 13.0 (n = 2)	10.0–15.0
Tail length	18.8 ± 1.2 (17.5–21.3)	19.3 ± 0.5 (19.0–20.0)	17.5–21.3
Stylet base (Stb) to median bulb valve base (v)	14.9 ± 1.9 (12.0–17.8)	14.5 ± 1.9 (13.0–17.0)	12.0–17.8
PERCENTAGES			
Stb-v/St	45.2 ± 5.3 (37.2–52.9)	45.7 ± 6.2 (40.6–54.0)	37.2–54.0
St/L	15.6 ± 0.6 (14.8–16.5)	12.6 ± 0.4 (12.2–13.0)	12.2–16.5
Ep/L	30.9 ± 1.4 (28.5–32.2)	24.9 ± 0.4 (24.4–25.2)	24.4–32.2
RATIOS			
*a*	15.7 ± 0.4 (15.0–16.2)	16.4 ± 0.5 (15.9–17.0)	15.0–17.0
*b*	2.5 ± 0.1 (2.0–2.5)	3.4 ± 0.1 (3.4–3.5)	2.0–3.5
*c*	11.2 ± 0.5 (10.2–11.7)	13.1 ± 0.6 (12.3–13.6)	10.2–13.6
*c′*	2.2 ± 0.2 (2.0–2.5)	2.2 ± 0.0 (2.2–2.2)	2.0–2.5

##### Etymology

Specific epithet derived from the Latin term *robur* = oak tree.

##### Type Material

The type material was obtained from live specimens that were fixed after morphological examination. The holotype female, ten paratype females, and six paratype males, mounted on glass slides, are deposited in the nematode collection of the National Museum of Natural Sciences, Madrid, Spain. An additional four female and two male paratypes were sent to the United States Department of Agriculture Nematode Collection, Beltsville, MD, USA.

##### Molecular Characterization

One D2–D3 of 28S rRNA, one ITS rRNA, and one *COI* gene sequence were obtained for this species. In the D2–D3 of the 28S rRNA (Figure 3) and ITS rRNA (Figure 4) gene trees, the sequences of the Florida *P. roboris* sp. n. formed clades with undescribed *Paratylenchus* sp.FL1 and *Paratylenchus* sp.FL2 species from Florida, from which it differed by 5.7% and 3.7% and 7.4% and 5.0%, respectively. In the *COI* gene tree (Figure 12), *P. roboris* sp. n. also clustered with these two unidentified species from Florida and unidentified *Paratylenchus* sp. from Missouri.

##### Diagnosis and Relationships

The new species is characterized by the presence of vermiform females with four incisures in the lateral field, a long stylet (63–71 µm long), slightly prominent vulval lips, and the lack of advulval flaps. The lip region is truncate and bearing small submedian lobes. The excretory pore is situated at the level of the median pharyngeal bulb. The spermatheca is well developed, elongated, and rounded. The tail is elongate-conoid, gradually tapering to form a subacute terminus. In addition, the species is charaterized by the presence of obese sedentary females; males lacking stylets; J2 having a well-developed stylet (31.5–34.1 µm long); and J3 and J4 encased in cuticles as resting stages. According to the species grouping by Ghaderi et al. [31], the new species belongs to group 11 characterized by a stylet longer than 40 µm, four incisures in the lateral field, and the lack of advulval flaps.

In having long female stylet length (63–71 µm), small submedian lobes, the excretory pore situated at the level of the median pharyngeal bulb, and males lacking stylets, the new species is morphologically and morphometrically very similar to *Paratylenchus teres* (Raski, 1976) Siddiqi, 1986 [5,6]. However, it can be distinguished from this species by its smaller female body (257–284 vs. 290–350 μm), lip region bearing small submedian lobes (vs. prominent submedian lobes), smaller values of ratio a (17.2–19.3 vs. 19–25), and longer stylets of J2 (32–34 vs. 11–15 µm). *Paratylenchus roboris* sp. n. also resembles *P. laocaiensis* Nguyen, Baldwin & Choi, 2004 [40], *P. steineri* Golden, 1961 [41], and *P. vitecus* (Pramodini, Mohalil & Dhanachand, 2006) Ghaderi, Kashi & Karegar, 2014 [31,42]. It differs from *P. laocaiensis* in its longer (257–284 vs. 220–250 μm) and less slender female body (*a* = 17.2–20.3 vs. 21–25), comparatively shorter female stylet (St/L = 22.6–25.9 vs. 30–33%), *c* ratio (11–15 vs. 9–11), and the male being present (vs. absent); it differs from *P. steineri* in having the excretory opening at level of the valve of the pharyngeal bulb vs. ‘in the vicinity of the base of the unprotruded stylet’; and it differs from *P. vitecus* in its less slender female body (*a* = 17.2–20.3 vs. 20–26) and comparatively longer female stylet (St/L = 22.6–25.9 vs. 17–20%).

Other species of group 11 have either longer stylets (length more than 72 μm in *P. enatus* (Raski, 1976) Siddiqi, 1986 [5,6], *P. macrodorus* Brzeski, 1963 [43], and *P. oostenbrinki* Misra & Edward, 1971 [44]) or shorter stylets (length up to 62 μm in *P. acti*, *P. micoletzkyi* Edward, Misra & Singh, 1967 [45], *P. mutabilis* Colbran, 1969 [46], and *P. raskii* (Phukan & Sanwal, 1979) Siddiqi, 1986 [6,47]).

##### Remarks

*Paratylenchus roboris* sp. n. is morphologically similar to *P. steineri* and differs from it by the position of the excretory pore, a character that is variable in some specimens. Unfortunately, *P. steineri* was described using vermiform females only, and no morphological information is available on other life stages of this species. We prefer considering *P. roboris* sp. n. a separate species from *P. steineri* until topotype material of this species is available to disprove our decision.

There are no reports of the presence of obese females in *Paratylenchus* species belonging to group 11. Obese females of *P. roboris* sp. n. were detached from live oak roots using a pressurized water spray and were missed when soil was processed without roots. The morphological features of this life stage resemble those of other *Paratylenchus* spp. such as *P. paralatescens*. Non-feeding, resistant, and coiled J4 is the most common life stage in the soil but does not possess useful diagnostic characters for the identification of this species. However, measurements of the molted stylet in the retained J2 cuticle that remains attached to J4 can be useful in a tentative diagnosis of this species. Vermiform females, males, and J2 have the most important diagnostic characters for the morphological identification of this species.

##### 2.2.4. *Paratylenchus acti* Eroshenko, 1978 [15]

(Figure 13, Table 7)

*Paratylenchus acti* was described from pine forests by Eroshenko [15] in the south of Sakhalin Island, Russia. The life stages of *P. acti* described in the original description of the species include only vermiform life stages. No sedentary swollen females are reported. Recently, a pin nematode population morphologically similar to *P. acti* was found in regulatory nematode samples collected from the rhizosphere of broomsedge in central Florida. This population was mixed with specimens of *Tylenchulus graminis* Inserra, Vovlas, O’ Bannon & Esser, 1988 [48], a parasite of broomsedge. The examined *P. acti* adults and juveniles were encased in the molted cuticles of the juveniles. Headless bodies of dead swollen and sedentary females were also detected in the soil with these life stages. Information on the morphological and molecular characters of these populations are provided in the following sections.

##### Description

*Female*: The body is small and slender, curved ventrad in a closed C-shaped body habitus when heat relaxed; the cuticle is finely annulated; the lateral field has four incisures; the lip region is rounded and lacking submedian lobes (under LM). Labial framework sclerotization is weak; the pharyngeal region is a typical paratylenchoid type. The stylet is flexible, and the cone is 88–91% of the total stylet length; the junction of the stylet cone with the shaft is thick, and stylet knobs are rounded; and the dorsal pharyngeal gland opening (DGO) is 5–7 µm behind stylet knobs. The median pharyngeal bulb is elongated, bearing distinct large valves; the isthmus is slender, surrounded by a nerve ring; and the excretory pore is situated at the level of the pharyngeal median bulb. The body is slightly narrower posterior to the vulva; the ovary is outstretched and well developed; the spermatheca is not well defined in some specimens or an oval 9 × 6 µm in size, and the crustaformeria is well developed; and the vulva has a transverse slit occupying more than half of the corresponding body width. Vulval lips are not protruding; the advulval flaps are absent; and the post-uterine sac is absent. The anus is difficult to distinguish in some specimens; the tail is slender, tapering in a finely rounded terminus.

*Sedentary swollen female*: The description of the morphology of this life stage is incomplete because we were not able to detect live females attached to the roots. The dead specimens we examined were headless because their stylet is cemented in the root tissues and usually breaks inside the tissues when specimens become detached. This life stage is characterized by a reniform-shaped body with the anterior portion often bent and the posterior portion after the vulva narrow and projecting like an opened sickle. The body is marked by annuli 1–3 µm wide, and the lateral field is 3.3 µm wide, showing three distinct bands. The anatomical structure of the internal organs was not discernable.

*Male*: The body is more slender than the female body, tapering towards both ends, curved to a closed C-shaped when heat relaxed. The cuticle is apparently smooth with fine annulations; the labial region is similar to that of females but narrower and continuous with the body, and sclerotization in the labial region is weak; the stylet is lacking. The pharynx is rudimentary and non-functional, procorpus, and metacorpus, and the basal bulb is inconspicuous; the excretory pore is located 74–82 µm away from the anterior end. The testis is outstretched, with small and rounded spermatozoa; the spicule is slender and slightly curved towards the end; the gubernaculum is curved; and the bursa is absent. The tail is conoid, tapering gradually to a finely pointed tip.

*Juveniles*: No second-stage juveniles were found. Third- and fourth-stage juveniles present in our population encased in the J2 and J3 cuticles that retained the molted stylet of the J2. No J3 stylets were observed.

##### Habitat and Locality

The population of *P. acti* was collected from soil and feeder roots of broomsedge (*Andropogon virginicus* L.) in a hammock used for the extraction of peat in Avon Park, FL, USA (latitude 27°64′044″ N, longitude 81°47′109″ W).

##### Molecular Characterization

D2–D3 of 28S rRNA, ITS rRNA, and *COI* gene sequences (one for each gene) were obtained for this population. In the D2–D3 of the 28S rRNA gene tree (Figure 3), the sequence of the Florida *P. acti* had a sister relationship with that of an unidentified *Paratylenchus* sp.SK (MW413580) from South Korea, and they were different from each other by 1.7%. These two sequences formed a clade with sequences of *P. aciculus* Brown, 1959 [49] (Spain), *P. colinus* (Huang & Raski, 1986) Brzeski, 1998 [20,50] (Iran), *P. aculentus* Brown, 1959 [49] (Belgium, Myanmar, China), and *P. paralatescens* (China, USA, Florida). In the ITS rRNA gene tree (Figure 4), the Florida *P. acti* formed a clade with *P. aciculus* (Spain) and *P. aculentus* types A and B (Belgium and China) and differed from them by 1.3–4.6%. In the *COI* gene tree (Figure 12), the Florida *P. acti* clustered with *P. aciculus* (Spain), *P. aculentus* type A (Belgium), and *P. paralatescens* (Florida), and it differed from the first two species by 12.4–12.8%.

##### Diagnosis (Based on Specimens Examined)

Florida *P. acti* is characterized by the presence of vermiform and swollen sedentary females. Vermiform females lack advulval flaps and show four incisures in the lateral field and a stylet 55–64 µm long. They have a rounded lip region, which lacks submedian lobes; the excretory pore is situated level to the pharyngeal median bulb, and the tail is elongate-conoid, gradually tapering to form a finely rounded terminus. Males lack stylets. According to the species grouping by Ghaderi et al. [31], *P. acti* belongs to group 11 characterized by a stylet longer than 40 µm, four incisures in the lateral field, and absent advulval flaps. The Florida population of *P. acti* is morphologically closely related to *P. aciculus*, *P. aculentus*, and *P. paralatescens*, but it differs from these three species in having four vs. three incisures in the lateral field.

##### Remarks

The Florida vermiform specimens herein studied agree well with morphometrics data of the original population of *P. acti*. *Paratylenchus acti* is an incomplete characterized species. However, its main morphometric and morphological features fit well those of the Florida population herein studied, especially the stylet and spicule length and vulva position. Swollen sedentary females having ectoparasitic sedentary habits were detected for the first time in this species. In Florida, *P. acti* is found commonly in uncultivated lands used as mine operations for the extraction of tropical peat. This detection is the second record of this species and the first report in Florida and in a location outside of the type locality.

#### 2.2.5. *Paratylenchus aquaticus* (type C) Merny, 1966 [16]

(Figure 10G,H, Figure 11G,H and Figure 14, Table 8 and Appendix A)

St. Augustine grass is a turfgrass native to Florida, where it is infested by many root-lesion nematode species such as *Pratylenchus hippeastri* Inserra, Troccoli, Gozel, Bernard, Dunn & Duncan, 2007 [51], which is a prevalent *Pratylenchus* species occurring in sod operations and landscape sites. Root samples of this turf grass collected in south and central Florida were found infested by a mixed population of *P. hippeastri* with a pin nematode morphologically resembling *P. aquaticus*, a species originally described from the Ivory Coast and reported in other countries in Africa, the Americas, and Asia, using morphologically variable populations. Two morphologically different genotypes A and B of *P. aquaticus* were characterized by Van den Berg et al. [30]. The genotype A represents a new species that is described in previous sections of this study as *P. hawaiiensis* sp. n. The genotype B corresponds to a population from Kansas. The morphological and molecular characterization of Florida pin nematode specimens from St. Augustine grass and the results of comparisons of their morphological and molecular features with those of *P. aquaticus* type B from Kansas, another Florida population from *Zoysia* sp., and other populations reported in the literature are presented below.

**Figure 14 plants-12-02770-f014:**
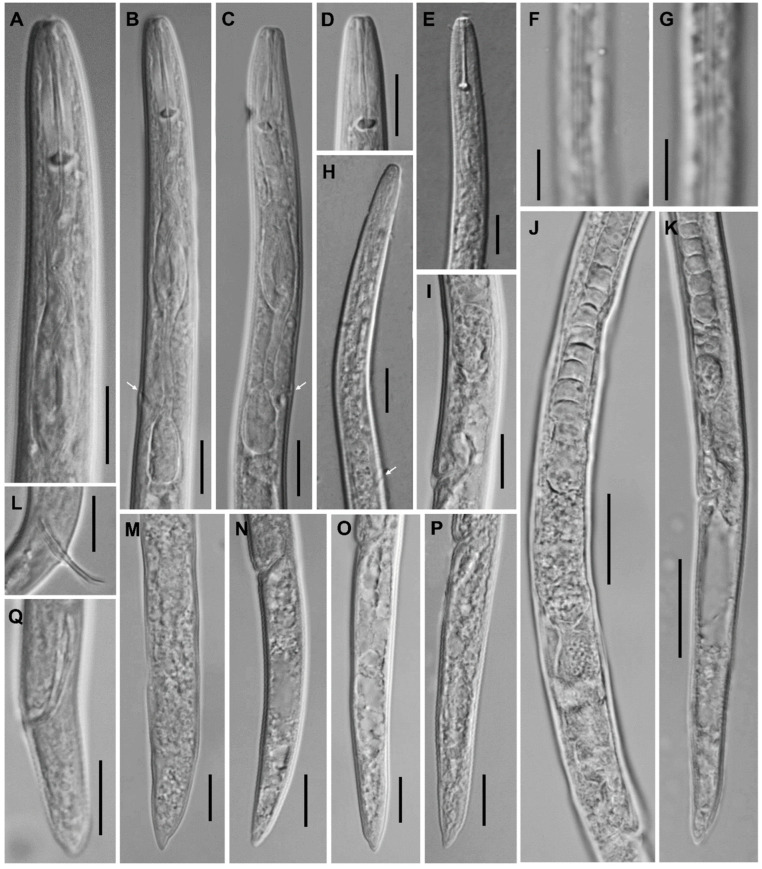
Light microscopic photos of *Paratylenchus aquaticus sensu lato* (defined as *P. aquaticus* type C) female, male, and juveniles from *Stenotaphrum secundatum* in Florida. (**A**–**D**): Views of anterior body of female. Note excretory pore (arrowed); (**E**): View of anterior body of second-stage juveniles; (**F**,**G**): Lateral field marked by three incisures. The central line is not well visible in (**G**); (**H**): Anterior body of male. Note excretory pore (arrowed); (**I**): Vulvar region showing advulval flap and spermatheca; (**J**,**K**): Views of female genital tract; (**L**,**Q**): Posterior body of male showing spicules; and (**M**–**P**): Views of posterior body of female with bluntly pointed and round tail terminus. (Scale bars = 10 μm).

**Table 8 plants-12-02770-t008:** Morphometrics of live females and males of a population of *Paratylenchus aquaticus sensu lato* (*P. aquaticus* type C) from St. Augustine grass (*Stenotaphrum secundatum*) in Florida.

Population	Okeechobee, Okeechobee County (CD3480—N21-00445)		
Character	♀ (Live)	♂ (Live)	J2 (Live)
*n*	8	4	3
L	364 ± 26.3 (321.7–409.8)	340.5 ± 15.8 (314.8–355.4)	225.0 ± 22.4 (196.0–250.4)
Stylet length (St)	16.9 ± 0.3 (16.2–17.0)	-	16.5 ± 0.6 (16.2–17.5)
Conus length	10.2 ± 0.4 (9.5–10.5)	-	10.0 ± 0.1 (9.9–10.2)
Stylet shaft + knob height	6.7 ± 0.3 (6.3–7.1)	-	6.2 ± 0.1 (6.0–6.3)
Knob width	3.5 ± 0.9 (3.2–3.8)	-	3.3 ± 0.9 (3.1–3.5)
Knob height	1.8 ± 0.1 (1.6–1.9)	-	1.5 ± 0.0 (1.5–1.5)
DGO	6.7 ± 0.3 (6.1–7.0)	-	5.0 ± 0.1 (4.9–5.2)
Median bulb valve length	6.9 ± 0.3 (6.2–7.0)	-	4.8 ± 0.2 (4.5–5.0)
Median bulb width	8.0 ± 1.2 (6.9–9.9)	-	6.9 ± 0.3 (6.5–7.1)
Isthmus length	17.3 ± 2.0 (14.0–19.5)	-	14.0 ± 2.2 (12.0–17.0)
Pharynx length	90.6 ± 3.9 (85.0–97.0)	88.9 ± 2.4 (87.1–93.0)	72.4 ± 6.3 (67.0–81.2)
Anterior end to excretory pore (Ep)	78.1 ± 5.1 (71.3–87.1)	71.6 ± 1.8 (70.0–74.2)	57.3 ± 1.4 (55.0–58.4)
Max body width	13.1 ± 2.3 (11.0–17.8)	10.6 ± 0.2 (10.4–10.8)	9.9 ± 0.5 (9.3–10.5)
Body width at vulva	11.1 ± 1.9 (9.0–14.5)	-	-
Body width at anus	8.1 ± 1.0 (6.4–8.9)	9.1 ± 0.4 (8.8–9.9)	6.0 ± 0.9 (5.0–7.0)
Anterior end to median bulb base	58.6 ± 2.4 (55.4–62.3)	-	46.8 ± 3.5 (43.0–51.4)
Lateral field width	3.8 ± 0.3 (3.5–4.0)	-	-
Anterior end-vulva distance	296.0 ± 23.2 (257.4–324.7)	-	-
Vulva-tail terminus distance	68.2 ± 7.7 (60.0–87.1)	-	-
Genital tract length	125.9 ± 45.9 (90.0–202.5)	96.7 ± 13.6 (75.2–112.8)	-
Vulva-anus distance	44.4 ± 3.6 (40.0–53.0)	-	-
Tail length	21.7 ± 1.9 (17.0–23.7)	24.1 ± 3.0 (19.8–28.0)	22.6 ± 1.9 (20.0–24.5)
Stylet base (Stb) to median bulb valve base (v)	32.6 ± 2.3 (30.0–35.6)	-	22.6 ± 1.8 (21.0–25.2)
Spicule length	-	17.4 ± 0.4 (17.0–17.9)	-
Gubernaculum length	-	3.9 ± 0.1 (3.8–4.0)	-
PERCENTAGES			
*V*	80.6 ± 1.7 (78.5–84.3)	-	-
G or T	34.4 ± 11.3 (23.0–53.0)	28.4 ± 4.0 (22.0–31.7)	-
Stb-v/St	1.9 ± 0.1 (1.7–2.1)	-	1.3 ± 0.1 (1.2–1.4)
St/L	4.6 ± 0.4 (4.1–5.2)	-	7.4 ± 0.6 (7.0–8.2)
Ep/L	21.2 ± 0.7 (19.8–22.1)	21.3 ± 0.7 (20.5–22.2)	25.5 ± 3.0 (22.9–29.7)
RATIOS			
*a*	26.9 ± 3.8 (21.7–32.1)	31.7 ± 1.2 (29.9–32.9)	22.6 ± 1.8 (21.0–25.2)
*b*	4.0 ± 0.4 (3.3–4.4)	3.8 ± 0.1 (3.6–4.0)	3.1 ± 0.2 (2.9–3.3)
*c*	16.9 ± 2.3 (15.2–22.3)	14.3 ± 1.1 (12.6–15.8)	18.2 ± 1.3 (16.3–19.2)
*c′*	2.7 ± 0.5 (1.9–3.5)	2.6 ± 0.3 (2.3–3.1)	2.2 ± 0.4 (1.7–2.6)

##### Description

*Female*: The body is small and slender, curved ventrad, and exhibits an open C-shaped body habitus when heat relaxed; the cuticle is finely annulated; the lateral field has three incisures, sometimes obscure; and the lip region is narrow, truncated, and lacking submedian lobes (under LM). Labial framework sclerotization is weak; the pharyngeal region is a typical paratylenchoid type. The stylet is rigid and straight, and the cone is *ca* 60% of the total stylet length; stylet knobs are rounded; the pharynx occupies almost one-fourth of the total body length; and the dorsal pharyngeal gland opening is 6.1–7.0 µm behind stylet knobs. The median pharyngeal bulb is elongated, bearing distinct large valves; the isthmus is slender, surrounded by a nerve ring; the basal bulb is rounded, the pharyngeal-intestinal valve is rounded; the excretory pore is situated at the level of the pharyngeal basal bulb; and the hemizonid is situated immediately anterior to excretory pore. The body is slightly narrower posterior to the vulva; the ovary is outstretched and well developed; the spermatheca and crustaformeria are well developed; the spermatheca (11–21 µm long) is rounded and filled with sperm in most specimens; and the vulva has a transverse slit occupying more than half of the corresponding body width. Vulval lips are not protruding; the advulval flaps are distinct; and the post-uterine sac is absent. The anus is difficult to distinguish in some specimens; and the tail is slender, conoid, almost straight or very slightly curved ventrad, finely annulated, and gradually tapers to form a rounded terminus.

*Male*: The body is more slender than the female body, tapering towards both ends, and curved to open C-shaped when heat relaxed. The cuticle is apparently smooth with fine annulations; the labial region is similar to that of females but narrower and slightly truncated, continuous with the body, sclerotization in the labial region is weak; the stylet is lacking. The pharynx rudimentary and non-functional, procorpus, and metacorpus, and the basal bulb is inconspicuous; and the excretory pore is located 70–74 µm away from anterior end. The testis is outstretched, with small and rounded spermatozoa; the spicule is slender, slightly curved towards the end; the gubernaculum is curved; the bursa is absent. The tail is elongate-conoid, tapering gradually to a finely pointed tip.

*Juveniles*: They are similar in morphology to the adult females. Second-stage juveniles are characterized by the presence of a well-developed stylet of the same length as that in adult stage; prominent pharynx components; an underdeveloped genital primordium; an indistinct anus; and a posterior body with the presence of a mucro on the tail terminus. The observed J3–J4 lacked stylets and were encased inside the cuticle of the J2.

##### Habitat and Locality

Florida *P. aquaticus* populations that were analyzed morphologically and molecularly in this study were collected from soil and feeder roots of St. Augustine grass (*S. secundatum*) in samples from a sod farm in Okeechobee, FL, USA (latitude 27°64′554″ N, longitude 80°84′482″ W), and roots from zoysia grass (*Zoysia* sp.) collected in a sod farm in Felda, FL, USA (latitude 26°30′262″ N, longitude 81°42′494″ W). The latter population was analyzed only molecularly.

##### Molecular Characterization

D2–D3 of 28S rRNA gene sequences were obtained for both Florida populations, whereas the ITS rRNA and *COI* gene sequences were obtained only from the Okeechobee population. In the D2–D3 of the 28S rRNA gene tree (Figure 3), the sequence of the two Florida *P. aquaticus* grouped together, confirming that they were conspecific and formed a clade with sequences of *P. rostrocaudatus* Huang & Raski 1987 [52] and a population of *P. aquaticus* type B from Kansas. The sequence difference between Florida *P. aquaticus* and *P. rostrocaudatus* was 10.5% and that between the two populations from Florida and Kansas was 8.5%. In the ITS rRNA gene tree (Figure 4), the sequence of Florida *P. aquaticus* formed a clade with *P. rostrocaudatus*. The sequence difference between these two species was 13.8%. In the *COI* gene tree (Figure 12), the sequence of Florida *P. aquaticus* type C clustered with those of a population of *P. aquaticus* type B from Kansas and an unidentified *Paratylenchus* sp. from Missouri. The sequence difference between the Florida and Kansas populations was 10.8% and that between Florida *P. aquaticus* and an undescribed Missouri *Paratylenchus* sp. was 12.3%.

##### Diagnosis (Based on Specimens Examined)

Florida *P. aquaticus* is characterized by the presence of three incisures in the lateral field and females having advulval flaps and stylets 16–17 µm long. The lip region is truncate and lacking submedian lobes. The excretory pore is situated at the level of the pharyngeal basal bulb. The spermatheca is well developed and rounded. The tail is elongate-conoid, gradually tapering to form a rounded terminus. Males lack stylets, and J2 has a well-developed stylet and a mucro on the tail terminus. According to the species grouping by Ghaderi et al. [31], *P. aquaticus* belongs to group 2 (characterized by stylets shorter than 40 µm, three incisures in the lateral field, and present advulval flaps).

##### Remarks

In the phylogenetic trees using D2–D3 of 28S rRNA and ITS rRNA gene sequences, Florida *P. aquaticus* formed a clade with *P. rostrocaudatu*s, a morphologically different species for having females with shorter stylets (15–16 vs. 16–17 µm) and excretory pore-anterior end distance (41–55 vs. 71–82 µm). In D2–D3 of the 28S rRNA gene tree and in that using *COI* gene sequences, Florida *P. aquaticus* formed a clade with *P. aquaticus* type B from Kansas. However, their sequences that differed by 8.5 and 10.5%, respectively, did not cluster together in the clade, indicating that these two *P. aquaticus* populations were not conspecific. These two populations also differed morphologically. The population from Kansas had a shorter excretory-anterior end distance (58–72 vs. 71–87 µm) and tail (17–24 vs. 22–26 µm) and greater c values (15–22 vs. 11–14) than those of the Florida population.

The Florida specimens herein studied agree morphologically well with the original population of *P. aquaticus* from the Ivory Coast, and only some morphometric differences were observed in the spicule length (17–18 vs. 21–22 µm). This minor difference found between the males of both populations might be considered intraspecific. They match well also with those of the Mexican population described and illustrated by Brzeski [53], in which the male of the species was also found. These males studied by Brzeski [53] match those of our Florida population and have shorter spicules than those of the type population (16–18 vs. 21–22 µm). Because of these morphological similarities, these two populations from Mexico and Florida are herein considered as conspecific, although no supporting molecular datasets are available for this population from Mexico.

These findings confirm that *P. aquaticus* is a species complex consisting of cryptic species. Nevertheless, considering that no molecular data are available for the population from the type locality of *P. aquaticus*, we cannot determine which population (from Kansas or from Florida) belongs to *P. aquaticus sensu stricto*. Therefore, for taxonomical convenience and until molecular data are available for the type population of *P. aquaticus* from the Ivory Coast, we regard the populations from Florida (in this study), Mexico (studied by Brzeski [53]), Kansas (classified by Van den Berg et al. [30] as *P. aquaticus* type B), and other similar populations of *P. aquaticus* as *P. aquaticus sensu lato*. Nonetheless, because of the currently available molecular and morphological data, we consider the Kansas population of *P. aquaticus* type B, as defined by Van den Berg et al. [30], a separate genotype from the Florida population defined in this study, and indicate the Florida population as *P. aquaticus* type C.

The Hawaiian population defined by Van den Berg et al. [30] as *P. aquaticus* type A does not fit the morphological pattern of *P. aquaticus sensu lato* and represents a new species described above as *Paratylenchus hawaiiensis* sp. n.

We would like to point out that the Florida populations of *P. aquaticus* type C have endoparasitic migratory habits and can be extracted from the infested St. Augustin grass roots kept moist in jars for longer than 10 days. This species has been detected in subsequent nematode surveys in sod operations of both St. Augustine and zoysia grasses located in distant localities in Florida. The common occurrence of this species on St. Augustine grass in sod operations indicates that, very probably, the pin nematode reported by Christie [2] as a parasite of St. Augustine grass in Florida is *P. aquaticus* type C.

#### 2.2.6. *Paratylenchus goldeni* Raski, 1975 [17] Species Complex

(Figure 10E,F, Figure 11E,F, Figure 15, Figure 16 and Appendix A, Table 9, Table 10 and Appendix A)

**Figure 15 plants-12-02770-f015:**
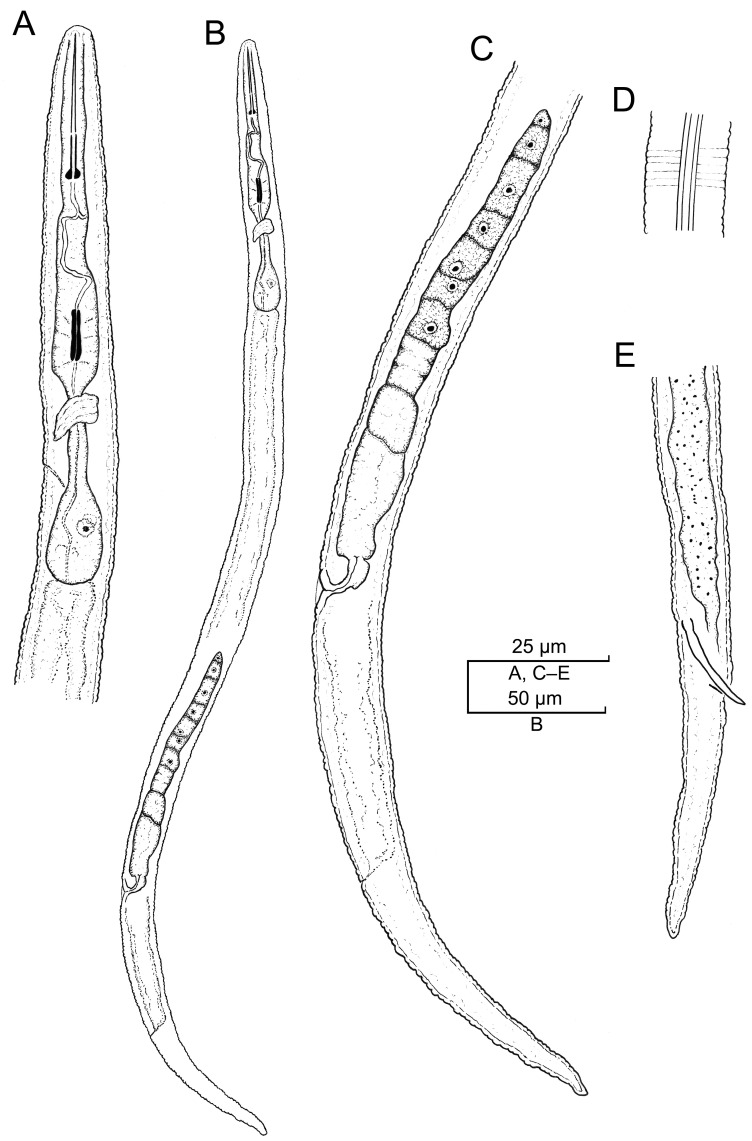
Camera lucida drawings of *Paratylenchus goldeni* from *Zoysia* sp. in Florida (CD3651). (**A**): Female pharyngeal region; (**B**): Female entire body; (**C**): Female posterior body; (**D**): Female lateral field; and (**E**): Male posterior body.

**Figure 16 plants-12-02770-f016:**
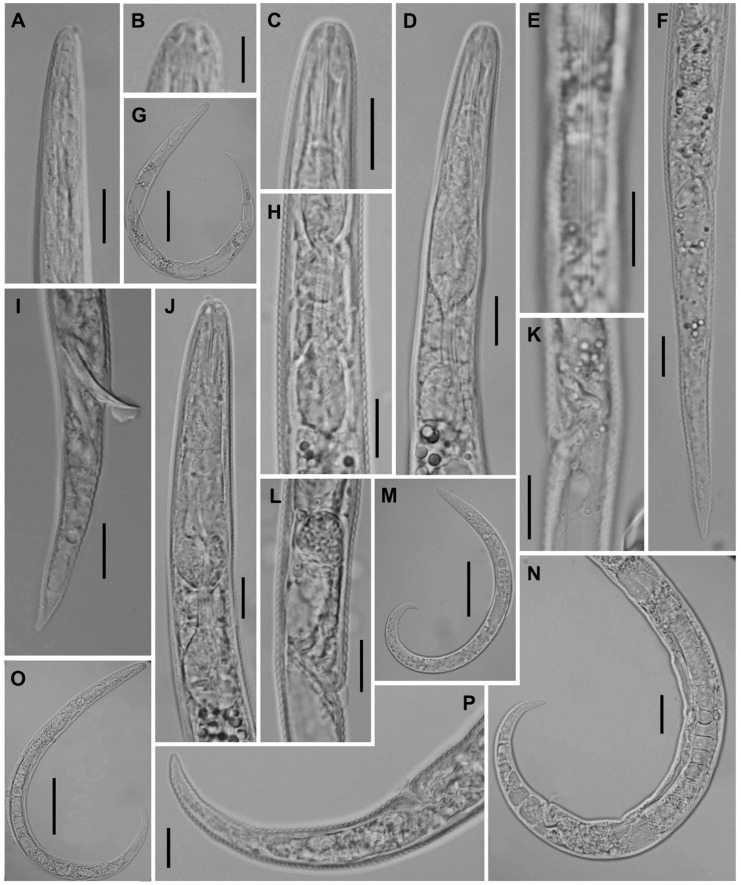
Light microscopic photos of *Paratylenchus goldeni* female, male and juvenile from *Ilex* sp. (CD3443) and *Enydra* sp. (CD3453) in Florida. (**A**–**I**,**K**): Population from *Ilex* sp.; (**J**,**L**–**P**): Population from *Enydra* sp. (**A**): Anterior body region of male; (**B**,**C**): Anterior body region of female; (**D**,**H**,**J**): Pharyngeal region of female; (**E**): Lateral field marked by four incisures; (**F**,**P**): Posterior body region of female; (**G**,**O**): Entire body of female; (**I**): Posterior body region of male; (**K**,**L**): Vulval region of female; (**M**): Non-feeding fourth-stage juvenile; and (**N**): Genital tract of female. (Scale bars: (**A**,**C**–**F**,**H**–**L**,**P**) = 10 μm; (**B**) = 5 μm; (**G**,**M**,**O**) = 50 μm; (**N**) = 20 μm).

**Table 9 plants-12-02770-t009:** Morphometrics of vermiform and egg-laying females of *Paratylenchus goldeni* associated with *Zoysia* sp. in Florida compared to those of the population from North Carolina by Zeng et al. [54].

Population	Madison, Madison County (CD3651—N21-01426-4)			North Carolina Zeng et al. [54]
Character	Vermiform ♀ (Live)	Egg-Laying ♀ (Live)	Vermiform ♀ (Fixed)	Vermiform ♀ (Fixed)
*n*	10	2	5	10
L	434.6 ± 31.5 (385.1–478.1)	520, 488	373.5 ± 42.1 (309.7–414.8)	374.9 ± 17.1 (358.9–399.8)
Stylet length (St)	26.8 ± 0.7 (25.7–27.7)	25.2, 27.2	25.2 ± 0.8 (24.0–26.0)	19.5 ± 0.8 (18.2–20.2)
Conus length	16.9 ± 0.4 (16.3–17.5)	16.9, 16.5	17.2 ± 0.4 (17.0–18.0)	-
Stylet shaft + knob height	9.9 ± 0.6 (8.8–11.0)	8.3, 10.7	8.0 ± 0.7 (7.0–8.7)	-
Knob width	3.7 ± 0.9 (3.1–4.0)	3.9, 3.9	3.0 ± 0.0 (3.0–3.0)	-
Knob height	1.9 ± 0.1 (1.7–2.0)	1.9, 1.7	1.6 ± 0.1 (1.5–1.7)	-
DGO	7.5 ± 0.6 (6.3–8.4)	7.8, 6.9	6.5 ± 0.4 (6.0–7.0)	-
Median bulb valve length	9.7 ± 0.4 (9.0–10.0)	9.4, 9.4	8.4 ± 0.2 (8.0–8.5)	-
Median bulb width	9.8 ± 0.6 (8.8–10.8)	14.9, 14.8	8.0 ± 0.7 (7.0–8.5)	-
Isthmus length	16.8 ± 1.0 (15.0–18.0)	18.6, 20.0	15.8 ± 0.8 (15.0–17.0)	-
Pharynx length	113.7 ± 5.8 (101.0–122.7)	101.9, 109.8	93.1 ± 3.2 (89.5–98.0)	90.6 ± 4.7 (82.9–94.8)
Anterior end to excretory pore (Ep)	90.1 ± 6.1 (75.0–96.0)	97.0, 98.0	69.2 ± 4.6 (65.0–75.0)	74.8 ± 4.1 (68.5–79.2)
Max body width	16.1 ± 0.9 (14.5–17.8)	24.7, 21.2	14.4 ± 2.4 (11.0–16.5)	17.5 ± 0.4 (16.9–18.1)
Body width at vulva	13.7 ± 0.8 (12.4–14.8)	18.0, 16.5	11.3 ± 1.7 (9.0–13.0)	14.1 ± 0.6 (13.5–15.0)
Body width at anus	9.8 ± 0.7 (8.9–11.0)	12.3, 9.9	9.1 ± 1.5 (7.0–10.5)	9.6 ± 0.4 (9.1–10.3)
Anterior end to median bulb base	76.6 ± 3.0 (71.3–82.1)	74.2, 77.2	61.2 ± 2.6 (57.0–64.0)	-
Lateral field width	3.4 ± 0.4 (3.0–4.0)	-	2.9 ± 0.3 (2.5–3.0)	-
Anterior end-vulva distance	334.8 ± 22.3 (302.0–363.3)	416.7, 386.1	281.0 ± 29.2 (238.8–310.9)	-
Vulva-tail terminus distance	99.0 ± 11.6 (83.1–115.6)	103.9, 101.9	93.3 ± 12.7 (72.0–104.0)	-
Genital tract length	104.9 ± 16.0 (85.0–136.6)	297.0, 261.3	88.5 ± 13.3 (69.0–99.0)	-
Vulva-anus distance	45.9 ± 4.7 (40.0–55.4)	54.0, 51.5	43.1 ± 5.2 (38.5–49.0)	40.3 ± 2.8 (37.6–44.4)
Tail length	52.9 ± 4.8 (44.5–58.4)	49.9, 50.4	56.5 ± 4.7 (53.0–63.0)	39.0 ± 3.2 (34.2–43.0)
Stylet base (Stb) to median bulb valve base (v)	38.8 ± 2.4 (34.0–42.5)	35.6, 37.6	29.9 ± 2.1 (27.5–32.0)	-
PERCENTAGES				
*V*	77.2 ± 1.3 (75.7–79.4)	79.1, 80.1	75.3 ± 1.0 (74.6–77.1)	78.9 ± 0.9 (78.1–80.3)
G or T	24.0 ± 2.7 (20.6–28.7)	57.1, 53.5	22.6 ± 2.4 (19.5–24.8)	-
Stb-v/St	1.4 ± 0.1 (1.2–1.6)	1.4, 1.3	118.4 ± 7.2 (109.8–128.0)	-
St/L	6.2 ± 0.4 (5.5–6.6)	4.8, 5.5	6.8 ± 0.6 (6.3–7.7)	-
Ep/L	20.7 ± 1.2 (18.7–22.4)	18.6, 20.0	18.7 ± 1.9 (15.7–21.0)	-
RATIOS				
*a*	27.1 ± 1.9 (24.4–30.1)	21.0, 23.0	27.5 ± 3.6 (24.2–32.2)	21.4 ± 1.3 (20.2–23.6)
*b*	3.8 ± 0.3 (3.4–4.2)	5.1, 4.4	4.0 ± 0.4 (3.5–4.3)	4.1 ± 0.2 (3.8–4.4)
*c*	8.3 ± 0.7 (6.8–9.6)	10.4, 9.4	6.9 ± 0.5 (6.3–7.4)	9.7 ± 0.9 (8.9–11.2)
*c′*	5.3 ± 0.6 (4.0–6.5)	4.0, 5.2	6.3 ± 0.9 (5.6–7.6)	4.1 ± 0.4 (3.6–4.7)

**Table 10 plants-12-02770-t010:** Morphometrics of live vermiform females of a population of *Paratylenchus goldeni* associated with *Ilex* sp. and another recovered from *Enydra* sp. roots in two localities in Florida compared to those of the paratypes from North Carolina.

Population	*Ilex* sp. Morriston, Levy County (CD3443—N21-00015-4)		*Enydra* sp. Milton, Santa Rosa County (CD3453—N21-00238-7)	Paratypes North Carolina Raski [17]	
Character	♀ (Live)	♂ (Live)	♀ (Live)	♀ (Fixed)	♂ (Fixed)
*n*	4	1	5	24	7
L	389.2 ± 19.2 (367.1–413.8)	396.9	369.6 ± 34.1 (308.0–409.8)	380.0 (350.0–430.0)	330.0 (300.0–380.0)
Stylet length (St)	17.5 ± 0.2 (17.3–17.8)	-	19.6 ± 1.0 (18.3–21.2)	17.0 (14.0–19.0)	-
Conus length	10.0 ± 0.4 (9.5–10.5)	-	11.2 ± 0.6 (10.5–11.9)	11.0 (10.0–11.0)	-
Stylet shaft + knob height	7.5 ± 0.5 (6.8–8.1)	-	8.4 ± 0.5 (7.8–9.3)	-	-
Knob width	2.4 ± 0.9 (2.3–2.5)	-	2.9 ± 0.9 (2.4–3.3)	-	-
Knob height	1.4 ± 0.2 (1.1–1.6)	-	1.7 ± 0.2 (1.4–2.0)	-	-
DGO	6.8 ± 0.7 (6.0–7.9)	-	8.7 ± 0.6 (8.0–9.5)	5.0	-
Median bulb valve length	8.7 ± 0.4 (8.0–8.9)	-	9.4 ± 0.4 (9.0–9.9)	-	-
Median bulb width	9.1 ± 0.5 (8.7–9.9)	-	10.6 ± 1.0 (9.0–11.8)	-	-
Isthmus length	14.8 ± 1.4 (12.8–16.2)	-	14.3 ± 0.7 (13.3–15.5)	-	-
Pharynx length	98.2 ± 1.7 (96.0–99.9)	90.0	107.6 ± 2.7 (105.0–112.8)	93.0	-
Anterior end excretory pore (Ep)	79.3 ± 3.9 (75.0–85.6)	74.7	80.8 ± 8.6 (65.0–89.1)	77.0 (67.0–83.0)	63.0 (54.0–66.0)
Max body width	15.1 ± 0.4 (14.8–15.8)	13.8	17.1 ± 1.6 (15.3–19.9)	13.3	-
Body width at vulva	12.7 ± 0.2 (12.3–12.8)	-	13.6 ± 1.3 (12.0–15.5)	-	-
Body width at anus	8.9 ± 0.0 (8.8–8.9)	10.8	8.7 ± 1.0 (7.5–9.9)	-	-
Anterior end to median bulb base	65.1 ± 1.3 (64.3–67.3)	-	59.6 ± 3.6 (54.4–65.0)	-	-
Lateral field width	3.4 ± 0.4 (3.0–3.7)	-	4.6 ± 0.2 (4.4–4.8)	-	-
Anterior end-vulva distance	300.7 ± 14.2 (282.1–316.8)	-	295.7 ± 29.1 (243.0–331.6)	320	-
Vulva-tail terminus distance	88.7 ± 5.5 (82.2–97.0)	-	74.0 ± 5.5 (65.0–79.2)	80	-
Genital tract length	101.2 ± 4.7 (97.0–108.9)	96	118.2 ± 12.5 (99.0–136.6)	140	-
Vulva-anus distance	41.3 ± 2.2 (37.6–43.5)	-	38.4 ± 3.3 (33.1–42.5)	-	-
Tail length	37.0 ± 1.9 (34.0–39.0)	50.4	34.7 ± 2.8 (30.5–38.6)	-	-
Stylet base (Stb) to median bulb valve base (v)	37.4 ± 1.3 (35.6–38.6)	-	38.2 ± 2.5 (35.0–42.0)	-	-
Spicule length	-	21.3	-	-	22.0 (21.0–23.0)
Gubernaculum length	-	4.9	-	-	4.0 (3.0–5.0)
PERCENTAGES					
*V*	77.2 ± 0.6 (76.5–78.0)	-	74.1 ± 1.3 (71.7–75.3)	80.0 (78.0–81.0)	-
G or T	25.7 ± 0.4 (25.1–26.3)	24.2	31.9 ± 1.1 (30.1–33.3)	35	32.0 (29.0–36.0)
Stb-v/St	2.1 ± 0.1 (2.0–2.2)	-	2.0 ± 0.2 (1.7–2.2)	-	-
St/L	4.5 ± 0.2 (4.2–4.8)	-	5.3 ± 0.3 (5.0–5.9)	-	-
Ep/L	20.3 ± 0.6 (19.2–20.8)	18.8	23.0 ± 3.2 (20.4–29.3)	-	-
RATIOS					
*a*	25.7 ± 0.7 (24.8–26.7)	26.8	21.6 ± 1.6 (19.3–23.3)	27.0 (19.0–33.0)	29.0 (27.0–29.0)
*b*	3.9 ± 0.2 (3.7–4.1)	4.4	3.4 ± 0.3 (2.8–3.6)	4.2 (3.9–5.0)	5.4
*c*	10.5 ± 0.2 (10.1–10.7)	7.8	10.7 ± 0.8 (9.4–11.9)	-	9.0 (8.0–10.0)
*c′*	4.2 ± 0.2 (3.8–4.4)	4.6	3.9 ± 0.3 (3.5–4.3)	-	2.3 ± 0.2

*Paratylenchus goldeni* is a species described from the rhizosphere of boxwood (*Buxus* sp.) in North Carolina and an unknown host in Georgia (USA). Morphological characters were used for this description, but no molecular analysis was conducted. In Florida, populations of *P. goldeni* were reported by Esser [3] and Lehman [4] in Hendry County. However, these reports cannot be confirmed because no morphological data on these populations are available. Subsequently, a population of this species was detected in North Carolina on centipede grass (*Eremochloa ophiuroides* (Munro) Hack.) and characterized morphologically and molecularly by Zeng et al. [54] and Zeng et al. [55]. In this study, a population genetically similar to that from North Carolina was collected from an unidentified host in California and analyzed only molecularly. Three additional populations of *P. goldeni*, morphologically variable but with high molecular similarity (see under remarks), were collected from geographical distant localities in north-central Florida. In the following sections, the morphological characters of these Florida populations are analyzed and compared with those of *P. goldeni* reported in the literature. Furthermore, a phylogenetic analysis was conducted to assess the relationship among the populations of *P. goldeni* from California, Florida, and North Carolina and also among these *P. goldeni* populations and other populations from Oregon and Washington reported by Sing et al. [14].

##### Description

*Immature female*: The body is small and slender, curved ventrad, and usually exhibits an open C-shaped body habitus when heat relaxed; the cuticle finely annulated; the lateral field has four incisures; and the lip region is narrow, rounded and lacking submedian lobes (under LM). Labial framework sclerotization is weak; the pharyngeal region is a typical paratylenchoid type. The stylet is rigid and straight, and the cone is 60–70% of the total stylet length; stylet knobs are rounded; the pharynx occupies *ca* 25–30% of the total body length; and the dorsal pharyngeal gland opening is 6.0–9.5 µm behind stylet knobs. The median pharyngeal bulb is elongated, bearing distinct large valves; the isthmus is slender, surrounded by a nerve ring; the basal bulb is pyriform, and the pharyngeal-intestinal valve is rounded; the excretory pore is situated at the anterior end of the pharyngeal basal bulb or at the isthmus level; and the hemizonid is situated immediately anterior to the excretory pore. The body is slightly narrower posterior to the vulva; the ovary is outstretched and well developed; the spermatheca and crustaformeria are well developed; the spermatheca is ovate or rounded, usually filled with sperm; and the vulva has a transverse slit occupying half of the correspond ing body width. Vulval lips are not protruding; the advulval flaps are distinct; and the post-uterine sac is absent. The anus is difficult to distinguish in some specimens; and the tail is slender, conoid, curved ventrad, finely annulated, and gradually tapers to form a finely rounded terminus.

*Egg-laying female* (only found in population CD3651, see Table 1): This life stage differs from immature females (from same population) in having a longer (504.0 (488.0–520.0) vs. 436.4 (385.1–478.1) µm) and wider (23.0 (21.2–24.7) vs. 16.1 (14.5–17.8) µm) body; a wider median bulb (14.9 (14.8–14.9) vs. 9.8 (8.8–10.8) µm); and smaller values of ratio *a* (22.0 (21.0–23.0) vs. 27.1 (24.4–30.1)).

*Male*: The body is more slender than the female body, tapering towards both ends, curved to open C-shaped when heat relaxed. The cuticle is apparently smooth with fine annulations; the labial region is similar to that of females but narrower and rounded, continuous with the body, sclerotization in the labial region is weak; and the stylet is lacking. The pharynx is rudimentary and non-functional, procorpus, and metacorpus, and the basal bulb inconspicuous; and the excretory pore is located 75.0–91.5 µm away from anterior end. The testis is outstretched, with small spermatozoa; the spicule is slender, slightly curved towards the end; the gubernaculum is almost straight or slightly curved; and the bursa is absent. The tail is elongate-conoid, tapering gradually to a finely pointed tip.

*Juveniles*: Second-stage juveniles (J2, only found in population (CD3651) are similar in morphology to the adult females. They are characterized by a well-developed stylet that is shorter than that of females (21.1 (20.7–21.7) vs. 26.8 (25.7–27.2) µm). The median pharyngeal bulb is prominent and with a valve 5.8 (5.5–6.1) µm long. The isthmus and basal bulb are visible in live specimens. The excretory pore opening is at the level of the anterior end of the basal bulb. The genital primordium is undeveloped; the anus and rectum are undiscernible in some specimens; and the tail is curved in the posterior portion, with a rounded terminus. Third (J3) is coiled, without a discernable stylet, pharynx, and gonads. Fourth-stage juveniles (J4) are similar in morphology to the adult females. They are characterized by the absence of stylets; underdeveloped pharynx components; an underdeveloped genital primordium; an indistinct anus; and a posterior body with a rounded terminus. These non-feeding stages retain the molted cuticles and are commonly found in the soil often in association with motile stages.

##### Habitat and Locality

The first of the populations of this species was associated with *Ilex* sp. in Morriston, Florida (latitude 29°01′378″ N, longitude 82°15′505″ W). The second was extracted by incubating the root and soil of *Enydra* sp. collected in Milton, Florida, in the vicinity of the state of Alabama (latitude 30°78′940″ N, longitude 87°06′862″ W), and the last one was collected from soil and feeder roots of zoysia grass (*Zoysia* sp.) in a farm in Madison County, FL, USA (latitude 30°61′091″ N, longitude 83°48′089″ W), from an unspecified plant in a forest, near a stream, in Madera County, CA, USA (latitude 37°24′596″ N, longitude 119°37′329″ W).

##### Molecular Characterization

Four D2–D3 of 28S rRNA, three ITS rRNA and five *COI* gene sequences were obtained for all the studied populations of this species from different localities and also for the two unidentified populations from Oregon and Washington. In D2–D3 of the 28S rRNA gene tree (Figure 3), the sequences of the Florida *P. goldeni* grouped together with those from California and also with the sequences obtained previously by Sing et al. [14] for *Paratylenchus* sp. J from Washington and Oregon, which are considered here as representatives of *P. goldeni* (see remarks below). These sequences formed a clade with *P. zurgenerus* Clavero-Camacho, Cantalapiedra-Navarrete, Archidona-Yuste, Castillo & Palomares-Rius, 2021 [25]. Intraspecific sequence variation for *P. goldeni* was up to 0.4%, and the sequence difference between *P. goldeni* and the related species, *P. zurgenerus,* was 4.0–4.5%. In the ITS rRNA gene tree (Figure 4), the sequences of Florida populations and those of North Carolina grouped together and formed a clade with *P. zurgenerus*, from which they differed by 5.5–7.3%. Differences in the ITS rRNA gene sequences between Florida and North Carolina were 0.2–1.1%. In the *COI* gene tree (Figure 12), the sequences of the Florida populations formed a clade with those of the populations from Oregon and Washington. However, the three Florida populations were grouped into two subclades, which were separated from the subclade containing the populations from Oregon and Washington. Intraspecific *COI* gene sequence variation for *P. goldeni* was up to 11.5%.

##### Diagnosis (Based on Specimens Examined)

*Paratylenchus goldeni* is characterized by a vermiform female having a lip region that is rounded and lacking submedian lobes at LM, with a stylet that is 17–27 µm long, the excretory pore situated at the anterior end of the pharyngeal basal bulb, four incisures in the lateral field, advulval flaps in the vulvar region, an ovate or rounded spermatheca, and a tail elongate-conoid in shape, tapering gradually to form a finely rounded terminus. Males and J4 lack stylets. According to the species grouping by Ghaderi et al. [31], *P. goldeni* belongs to group 3 characterized by a stylet shorter than 40 µm, four incisures in the lateral field, and the presence of advulval flaps.

##### Remarks

The three studied Florida populations (CD3443, CD3453 and CD3651; see Table 8, Table 9 and Table 10) show some remarkable morphometric differences, when they are compared, in the stylet length (17.3–17.8 vs. 18.3–21.2 vs. 25.7–27.7 µm). These differences in the stylet length between the Florida populations might suggest that they belong to morphologically different *Paratylenchus* species; however, the molecular data obtained for these populations show that they are the members of *P. goldeni* species complex.

The Florida specimens herein studied agree well with the original population of *P. goldeni* from North Carolina and also the population from North Carolina identified by Zeng et al. [54]. These populations share the lip region shape, stylet length, tail length and shape, and spicules length. Moreover, molecular data obtained for the Florida populations show high similarity in the ITS rRNA gene sequences with those of the North Carolina population [55], reaching 99.77% for the populations CD3443 and CD3453 and 98.84% for the population CD3651. These findings confirm that Florida and North Carolina populations are conspecific and representatives of *P. goldeni* until sequences of the population of this species from the type locality, Salemburg, North Carolina, become available disproving this identification.

The molecular data herein obtained for *P. goldeni* from Florida (CD3443, CD3453, and CD3651) and California (CD3470) populations show that the similarity in the available sequences of the D2–D3 of the 28S rRNA gene with those of *Paratylenchus* sp. J is very high (99.71–99.86%). Furthermore, the new sequences of the *COI* gene for *Paratylenchus* sp. J also show a high similarity with those from Florida populations (CD3443, CD3453, and CD3651), confirming that these populations from the Pacific Northwest are likely conspecific with those from Florida and are representatives of the *P. goldeni* species complex.

The three Florida populations of *P. goldeni* have ectoparasitic migratory habits.

#### 2.2.7. *Paratylenchus minutus* Linford, Oliveira & Ishi, 1949 [22]

(Figure 10B,C, Figure 11B,C, Appendix A and Appendix A, Table 11 and Appendix A)

This species was described by Linford et al. [22] from pineapple in Oahu and other islands (Molokai and Maui) in Hawaii, where, in addition to pineapple, it was found feeding inside the root tissues of numerous plants including agronomic crops, vegetables, weeds, and ornamentals. Subsequently, a redescription of *P. minutus* was published by Raski [17] using topotype specimens. This redescription includes morphological characters and morphometrics of other populations from Hawaii and distant geographical areas on different hosts, including *Coffea* sp. in Brazil. However, no molecular analyses were conducted on these type populations of *P. minutus.* In Florida, populations of *P. minutus* were reported by Esser [3] and Lehman [4] in Monroe County. Nevertheless, these reports cannot be confirmed because no morphological data on these populations are available. Subsequently, a population identified as *P. shenzhenensis*, a closely related species to *P. minutus*, was reported from a plant nursery in Apopka, Florida, by Singh et al. (2021), who identified this population by molecular analysis only.

*Paratylenchus shenzhenensis* was described by Wang et al. [13] from the roots of *Anthurium andreanum* in Shenzhen, Guangdong Province, China. This species was considered to be closely related to *P. minutus.* Wang et al. [13] morphologically separated *P. shenzhenensis* from *P. minutus* by comparing *P. shenzhenensis* with a population of *P. minutus* from Taiwan without using topotype material of this species from Hawaii. The authors state that *Paratylenchus shenzhenensis* had ‘submedian lobes and a slight depression at the oral area’ at SEM, whereas *P. minutus* has ‘tiny lips which are prominent near the oral aperture and form a truncate tip’ at LM. This comparison is not correct because the lip sectors in the two species should be observed and compared using the same microscopy, SEM or LM. In any case, both species at LM have a truncate lip region. Furthermore, *P. shenzhenensis* was separated molecularly from *P. minutus* by comparing ITS rRNA gene sequences of the paratypes from China with those of a *P. minutus* population from Taiwan without obtaining DNA sequences from topotype specimens from Hawaii. These morphological and molecular differences between the two populations without using topotype material are not enough for the separation of *P. shenzhenensis* from *P. minutus* and cast doubt on the validity of *P. shenzhenensis* as a separate taxon from *P. minutus*. In this study, a pin nematode population morphologically matching *P. minutus* was detected from a daylily stand in Jasper, central Florida. A preliminary analysis indicated that the DNA sequences of this population were congruous with those of *P. shenzhenensis* from China and the population mentioned above from an unknown host, in Central Florida, reported by Singh et al. [14]. To clarify the taxonomic status of this daylily population, we obtained a population of *P. minutus* that was collected from coffee in the Hawaiian archipelago. In the following sections, we compare the morphological characters and DNA sequences of this Florida population from daylilies with those of *P. shenzhenensis* from China and Florida reported by Sing et al. [14] and the population of the closely related species *P. minutus* from Hawaii.

##### Description (Based on the Hawaiian (from Roots of *Coffea* sp.) and Florida (from Roots of *Hemerocallis* sp.) Populations)

*Female*: The body is small and slender, curved ventrad, and exhibits a C- or G-shaped body habitus when heat relaxed; the cuticle is finely annulated; the lateral field has four incisures; and the lip region is narrow, truncated, and bearing very small submedian lobes (under LM). Labial framework sclerotization is weak; the pharyngeal region is a typical paratylenchoid type. The stylet is rigid and straight, and the cone is *ca* 60–65% of the total stylet length; stylet knobs are rounded; the pharynx occupies almost two-sevenths of the total body length; and the dorsal pharyngeal gland opening is 4.0–4.7 µm behind stylet knobs. The median pharyngeal bulb is elongated, bearing distinct large valves; the isthmus is slender, surrounded by a nerve ring; the basal bulb is pyriform, and the pharyngeal-intestinal valve is rounded; the excretory pore is situated at the level of or anterior to the pharyngeal basal bulb; and the hemizonid is situated immediately anterior to the excretory pore. The body is slightly narrower posterior to the vulva; the ovary is outstretched and well developed; the spermatheca and crustaformeria are well developed; the spermatheca is rounded (11–16 µm long) and filled with sperm; and the vulva has a transverse slit occupying more than half of the corresponding body width. Vulval lips are not protruding; the advulval flaps are distinct; and the post-uterine sac is small, less than one vulval body width in length. The anus is difficult to distinguish in some specimens; and the tail is slender, conoid, curved ventrad, finely annulated, and gradually tapers to form a finely rounded terminus.

*Male*: The body is more slender than the female body, tapering towards both ends, and curved to open C-shaped when heat relaxed. The cuticle is apparently smooth with fine annulations; the labial region is similar to that of females but narrower and slightly truncated, continuous with the body; sclerotization in the labial region is weak; and the stylet is lacking. The pharynx is rudimentary and non-functional, procorpus, and metacorpus, and the basal bulb is inconspicuous; the excretory pore is located 50–59 µm away from the anterior end. The testis is outstretched, with rounded refractive spermatozoa; the spicule is slender, slightly curved towards the end; the gubernaculum is curved; and the bursa is absent. The tail is elongate-conoid, tapering gradually to a finely pointed tip.

*Juveniles*: They are similar in morphology to the adult females. However, they are characterized by the presence of well-developed stylets in all juveniles we studied as reported by Wang et al. [13] for the population from China. The stylets of the fourth-stage juveniles were slightly shorter than those of females (16 vs. 16.9–18.8 µm). They are characterized by underdeveloped pharynx components; an underdeveloped genital primordium; an indistinct anus; and a posterior body with a rounded terminus.

##### Habitats and Localities

The populations of this species were extracted from (i) the roots of *Coffea* sp., in Kalaheo, Kauai Island, Hawaii (latitude 29°19′0″ N, longitude 159°54′9″ W), where populations of *P. minutus* were collected since 1951, and (ii) from soil and feeder roots of daylily (*Hemerocallis* sp.) in a nursery in Jasper, FL, USA (latitude 30°49′138″ N, longitude 83°12′693″ W).

**Table 11 plants-12-02770-t011:** Morphometrics of *P. minutus* populations from *Coffea* sp. in Hawaii and from daylily (*Hemerocallis* sp.) in Florida.

Population	*Coffea* sp. (CD3538—N21-01001) Hawaii	*Hemerocallis* sp. (CD3465—N21-00389) Florida	
Character	♀ (Live)	♀ (Live)	♂ (Live)
*n*	4	8	5
L	228.5 ± 9.7 (220.6–244.5)	253.3 ± 12.7 (239.5–279.5)	242.5 ± 20.7 (210.8–269.2)
Stylet length (St)	17.5 ± 1.2 (16.3–19.0)	18.0 ± 0.3 (16.9–18.8)	-
Conus length	11.0 ± 0.6 (10.3–12.0)	10.8 ± 0.3 (10.0–11.2)	-
Stylet shaft + knob height	6.5 ± 0.7 (5.6–7.3)	7.2 ± 0.5 (6.6–8.0)	-
Knob width	3.3 ± 0.9 (3.1–3.5)	3.5 ± 0.9 (3.1–3.9)	-
Knob height	1.5 ± 0.0 (1.4–1.6)	1.6 ± 0.2 (1.3–1.8)	-
DGO	4.2 ± 0.1 (4.0–4.3)	4.4 ± 0.2 (4.0–4.7)	-
Median bulb valve length	5.1 ± 0.4 (4.5–5.5)	5.3 ± 0.2 (5.0–5.5)	-
Median bulb width	7.5 ± 0.9 (6.5–8.9)	8.2 ± 1.1 (7.0–9.9)	-
Isthmus length	13.5 ± 2.2 (11.3–16.5)	12.4 ± 2.0 (8.0–14.8)	-
Pharynx length	71.0 ± 1.7 (69.0–73.0)	73.8 ± 3.1 (70.3–79.2)	70.7 ± 3.6 (64.3–74.2)
Anterior end to excretory pore (Ep)	56.8 ± 1.6 (54.5–59.0)	60.7 ± 4.3 (52.0–65.3)	56.6 ± 3.1 (50.5–59.4)
Max body width	11.5 ± 1.1 (10.3–13.0)	14.1 ± 1.8 (12.3–16.8)	9.7 ± 0.4 (9.2–10.3)
Body width at vulva	9.9 ± 0.7 (9.0–10.5)	10.2 ± 0.8 (9.0–11.8)	-
Body width at anus	6.7 ± 1.0 (5.5–8.0)	7.3 ± 0.5 (6.4–8.0)	7.5 ± 0.4 (6.9–8.0)
Anterior end to median bulb base	45.5 ± 2.3 (43.0–49.0)	41.3 ± 1.7 (39.6–45.0)	-
Anterior end-vulva distance	187.8 ± 6.5 (182.0–198.0)	212.0 ± 11.7 (198.0–238.0)	-
Vulva-tail terminus distance	40.8 ± 3.3 (38.5–46.5)	41.3 ± 2.6 (38.0–47.5)	-
Genital tract length	68.1 ± 9.4 (60.0–81.2)	99.0 ± 22.6 (29.0–135.6)	86.5 ± 14.9 (62.0–105.9)
Vulva-anus distance	30.4 ± 3.7 (27.1–35.6)	23.5 ± 2.1 (21.0–27.7)	-
Tail length	17.3 ± 2.5 (15.0–21.4)	17.5 ± 1.5 (15.8–21.0)	17.7 ± 2.6 (15.0–21.7)
Stylet base (Stb) to median bulb valve base (v)	21.0 ± 0.1 (20.8–21.0)	23.2 ± 2.1 (20.7–27.0)	-
Spicule length	-	-	17.4 ± 0.5 (16.8–18.1)
Gubernaculum length	-	-	3.0 ± 0.1 (3.0–3.2)
PERCENTAGES			
*V*	82.2 ± 0.7 (81.0–82.8)	83.7 ± 1.0 (82.0–85.1)	-
G or T	30.4 ± 3.7 (27.2–35.6)	38.7 ± 8.2 (24.3–50.4)	34.9 ± 4.5 (29.4–42.4)
Stb-v/St	1.2 ± 0.0 (1.1–1.2)	1.3 ± 0.2 (1.1–1.5)	-
St/L	7.6 ± 0.7 (6.7–8.3)	7.1 ± 0.5 (6.4–7.7)	-
Ep/L	24.9 ± 0.7 (24.1–25.9)	23.9 ± 1.2 (21.7–25.8)	23.4 ± 1.3 (21.6–25.2)
RATIOS			
*a*	20.1 ± 0.1 (20.8–21.0)	18.2 ± 2.1 (14.8–21.3)	24.8 ± 1.3 (22.9–27.1)
*b*	3.2 ± 0.1 (3.0–3.3)	3.4 ± 0.2 (3.2–3.9)	3.4 ± 0.2 (3.1–3.7)
*c*	13.4 ± 1.3 (11.5–15.2)	14.5 ± 0.8 (13.3–15.4)	13.8 ± 1.3 (12.4–15.6)
*c′*	2.6 ± 0.4 (2.0–3.0)	2.4 ± 0.2 (2.1–2.8)	2.3 ± 0.4 (2.0–2.8)

##### Molecular Characterization

Two D2–D3 of the 28S rRNA gene and two ITS rRNA gene sequences were obtained in this study. In the partial 28S rRNA gene phylogenetic tree (Figure 3), the populations of *P. minutus* from Florida and Hawaii grouped together, with the Florida population identified as *P. shenzhenensis* by Singh et al. [14]. These populations formed a clade with *P. hawaiiensis* sp. n. and also *P. pestis* (Thorne, 1943) Goodey, 1963 [56,57] (=*Cacopaurus pestis*). In the ITS rRNA gene tree (Figure 4) the Florida and Hawaii populations of *P. minutus* grouped together and formed a clade with samples of unidentified *Paratylenchus* sp. from China and Malaysia, which are identified here as representatives of *P. minutus.* For the D2–D3 of the 28S rRNA gene and the ITS rRNA gene sequences, intraspecific sequence variations for *P. minutus* were up to 1.7% and 3.8%, respectively.

##### Diagnosis (Based on Specimens Examined)

*Paratylenchus minutus* is characterized by the presence of four incisures in the lateral field, present advulval flaps, and a female stylet length of 16–19 µm. The lip region is truncate and bearing very small submedian lobes. The excretory pore is situated at the level of or anterior to the pharyngeal basal bulb. The spermatheca is well developed and rounded. The tail is elongate-conoid, gradually tapering to form a finely rounded terminus. And there is a presence of males lacking stylets and juvenile stages having a well-developed stylet. According to the species grouping by Ghaderi et al. [31], *P. minutus* belongs to group 3 characterized by a stylet shorter than 40 µm, four incisures in the lateral field, and present advulval flaps.

##### Remarks

The populations from Hawaii and Florida herein studied perfectly fit morphologically and morphometrically the original description by Linford et al., 1949 and the redescription by Raski [17] of *P. minutus*.

On the other hand, the specimens herein studied perfectly fit, morphologically and morphometrically, the original description of *P. shenzhenensis.* Furthermore, the molecular data obtained in this and the previous study by Singh et al. [14] based on D2–D3 of 28S rRNA and ITS rRNA gene sequences indicate that the Hawaiian population from coffee and that from daylily in Florida of *P. minutus* are conspecific with the Florida population of *P. minutus* identified in Singh et al. [14] as *P. shenzhenensis*, which was found to be conspecific with original population of *P. shenzhenensis* from China in that study. Therefore, all of these populations from China, Florida, and Hawaii are conspecific and represent *P. minutus* since it was described before *P. shenzhenensis*. In the D2–D3 of the 28S rRNA gene tree, the populations from Hawaii and Florida of *P. minutus* cluster in a highly supported (100%) clade with *P. pestis*. This relationship between *P. minutus* and *P. pestis* was weaker in the ITS rRNA gene tree.

##### On the Identity of *P. shenzhenensis*

The description of *P. shenzhenensis* does not differ from the redescription of *P. minutus* by Raski [17], who states that in the females of this species, ‘Tiny lips protrude slightly near the oral aperture giving a truncate anterior tip. Lateral field with four incisures, inner two slightly less distinct than the outer two. Vulval flaps present. Tail evenly conoid, ventrally arcuate with a rounded tip or blunt or slightly misshapen to subdigitate’.

The specimens of the Florida population from daylily agree very well with those described by Wang et al. [13] for *P. shenzhenensis* from *Anthurium andreanum* in China [13] and also with those in the original description of *P. minutus* from pineapple in Hawaii [17,22] and the others from coffee in Hawaii characterized in this study. Females showed the most important characters of these two species and populations such as a small and ventrally curved body, truncate lip region, excretory pore located at the level of the pharyngeal basal bulb, a prominent spermatheca filled with sperm, and the portion of the body posterior to the vulva tail curved ventrally. Body and stylet sizes were similar (239.5–279.5 vs. 249.9–302.0 for the paratypes from China; 240.0–310.0 and 220.6–244.5 μm for the paratypes and the coffee population of *P. minutus* from Hawaii; and 16.9–18.8 vs. 17.0–21.0, and 16.0–21.0 and 16.3–19.0 μm long in both species, respectively). Tail shape varied from conoid with a bluntly pointed and smooth terminus to tapered with a finely rounded terminus or sub-digitate as in *P. shenzhenensis* and *P. minutus*. Males of the Florida population showed spicules of the same length as those of *P. shenzhenensis* and *P. minutus* (16.8–18.1 vs. 16.0–19.0 and 16.0–19.0 μm long, respectively). Unfortunately, the small number of specimens obtained of the population from coffee in Hawaii did not include males. All of the life stages of the Florida population were extracted from daylily roots and were endoparasitic like *P. shenzhennesis* from China and *P. minutus* from Hawaii. Taking into consideration that *P. minutus* was described by Linford et al. [22] as an endoparasite and has morphological characters that do not differ from those of *P. shenzhenensis* and the populations from daylilies in Florida and coffee in Hawaii, we consider these populations conspecific and belonging to the species *P. minutus*. Additionally, the molecular data obtained in this study, based on the analysis of D2–D3 of 28S rRNA and ITS rRNA gene sequences, clearly confirm that the Hawaiian population of *P. minutus* from the roots of *Coffea* sp, the original population of *P. shenzhenensis* from the roots of *Anthurium andreanum*, and the Florida population of *P. shenzhenensis* from the roots of daylilies are conspecific. As a conclusion, *P. shenzhenensis* should be regarded as a junior synonym of *P. minutus*.

##### Emended Diagnosis of *Paratylenchus minutus* (=*Paratylenchus shenzhenensis* syn. n.)

This species is characterized by the presence of four incisures in the lateral field, present advulval flaps, and a female stylet length of 16–21 μm. The lip region is truncate and bearing very small submedian lobes. The excretory pore is situated at the level of or anterior to the pharyngeal basal bulb. A prominent rounded spermatheca is filled with sperm, and a post-uterine sac is present and small. The portion of the body posterior to the vulva tail is curved ventrally. The tail elongate-conoid, gradually tapering to form a finely rounded terminus. There is a presence of males lacking stylets with spicules 15–19 μm long and juvenile stages having a well-developed stylet.

#### 2.2.8. *Paratylenchus paralatescens* (Munawar, Cai, Ye, Powers & Zheng, 2018) Munawar, Miao, Castillo, Zheng, 2020 [23,24]

(Figure 1, Figure 2, Figure 3, Figure 4 and Figure 5 in Troccoli et al. [8] is identified as *G. latescens* Raski, 1976 [5]; Figure 2 in Inserra et al. [9] shows *G. latescens*; Figure 1, Figure 2 and Figure 3 in Munawar et al. [23] show *G. paralatescens*; and Figure 17 and Table 12, Appendix A in this study).

This species was described from unspecified bamboo plants in China by Munawar et al. [23] using a population consisting of vermiform females and males and was included in the group of pin nematodes lacking obese females. Juveniles were not detected in these samples. In our survey, we found a morphologically similar pin nematode population parasitizing black bamboo (*Phyllostachys nigra*) in north Florida. This population contained vermiform adults along with obese females and juveniles. DNA sequences of this population matched those of the specimens of *P. paralatescens* in China, indicating that the two populations are conspecific. In previous studies by Troccoli et al. [8] and Inserra et al. [9], another population with obese females from timber bamboo (*Phyllostachys bambusoides* Siebold & Zucc.) was discovered in central Florida and, despite differences in the morphometrics of the second-stage juveniles of the two populations, it was identified incorrectly as *P. latescens*, a species described from mesquite (*Prosopis* sp.) in Texas by Raski [5]. Although no DNA sequences of the population from timber bamboo described by Troccoli et al. [8] are available, morphological comparisons of the life stages of the two Florida pin nematode populations from fish pole and timber bamboos, as shown in the following sections of this study, indicate that both populations do not differ morphologically and should be considered *P. paralatescens.*

**Figure 17 plants-12-02770-f017:**
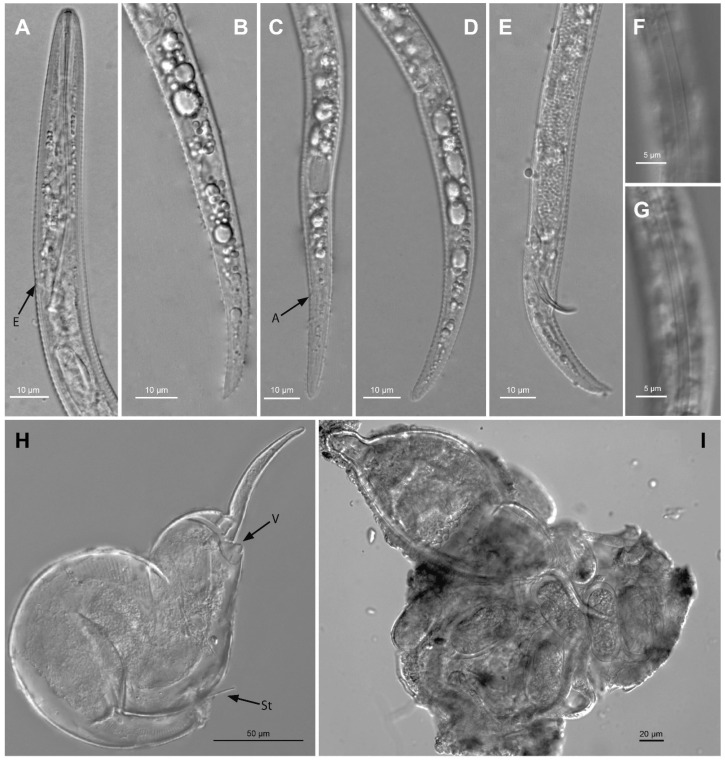
Light microscopic photos of *Paratylenchus paralatescens* female and male from *Phyllostachys nigra* in Florida. (**A**): Anterior body of vermiform female. Note excretory pore (arrowed); (**B**–**D**): Posterior body of vermiform female. Note different shapes of tail terminus from wedge-like to bluntly pointed and anus (arrowed); (**E**): Posterior body of male; (**F**,**G**): Lateral field marked by three incisures; and (**H**): Entire body of a coiled swollen female. Note stylet (St) and vulva (V); (**I**): Swollen female with attached egg mass.

**Table 12 plants-12-02770-t012:** Morphometrics of females of *Paratylenchus paralatescens* populations from *Phyllostachys nigra* (Florida) and *Phyllostachys bambusoides* (Florida).

Population	*Phyllostachys nigra* (CD3396—N20-01262-1) Florida		*Phyllostachys bambusoides* (as *P. latescens*) Troccoli et al. [8] Florida
Character	♀ (Live)	♀ (Fixed)	♀ (Fixed)
*n*	7	2	15
L	294.7 ± 16.7 (277.2–326.6)	260.0, 294.0	279.0 ± 10.9 (264.0–300.0)
Stylet length (St)	74.6 ± 3.4 (67.0–77.7)	71.0, 80.0	72.0 ± 2.8 (67.5–76.5)
Conus length	68.2 ± 3.1 (61.5–71.2)	65.5, 74.5	66.0 ± 3.3 (59.5–70.0)
Stylet shaft + knob height	6.4 ± 08 (5.5–8.1)	5.5, 5.5	6.0 ± 0.9 (4.5–7.5)
Knob width	3.1 ± 0.9 (2.9–2.4)	2.5, 2.1	2.5 ± 0.4 (2.3–5.0)
Knob height	1.5 ± 0.2 (1.3–1.9)	1.3, 1.2	-
Median bulb valve length	9.8 ± 0.6 (9.0–10.5)	9.9, 9.4	9.0 ± 0.7 (7.5–10.0)
Median bulb width	9.1 ± 0.6 (8.0–9.9)	9.0, 10.0	8.0 ± 0.4 (7.5–8.5)
Isthmus length	15.5 ± 1.6 (13.8–17.8)	15.0, 14.0	16.0 ± 1.7 (12.0–18.5)
Pharynx length	130.6 ± 4.6 (123.7–135.6)	115.8, 134.5	127.0 ± 4.3 (120.0–132.0)
Anterior end to excretory pore (Ep)	68.7 ± 3.9 (61.3–72.9)	65.4, 69.3	68.0 ± 3.5 (64.0–78.0)
Max body width	13.6 ± 0.4 (13.0–14.3)	12.0, 12.6	11.5 ± 0.6 (11.0–13.0)
Body width at vulva	12.0 ± 0.5 (11.1–12.8)	11.0, 11.8	-
Body width at anus	6.6 ± 0.4 (5.9–7.1)	6.4, 6.9	6.0 ± 0.4 (5.5–6.5)
Anterior end to median bulb base	100.7 ± 3.1 (94.0–103.9)	90.0, 103.1	-
Lateral field width	2.4 ± 0.4 (2.0–2.9)	1.9, 2.2	2.2 ± 0.3 (2.0–2.5)
Anterior end-vulva distance	207.7 ± 11.3 (197.0–230.6)	182.0, 201.0	198 ± 9.2 (185.0–218.0)
Vulva-tail terminus distance	87.0 ± 6.0 (79.2–96.0)	78.0, 98.0	87.2 ± 1.7 (73.0–132.0)
Genital tract length	44.0 ± 6.0 (41.6–45.5)	37.0, 38.0	37.0 ± 3.5 (31.0–42.0)
Vulva-anus distance	60.7 ± 5.6 (53.5–71.2)	54.0, 58.0	55.0 ± 3.1 (47.0–59.0)
Tail length	25.2 ± 0.7 (23.7–26.0)	26.2, 24.7	24.5 ± 2.5 (21.5–31.5)
Stylet base (Stb) to median bulb valve base (v)	22.6 ± 3.6 (18.0–29.7)	17.8, 15.8	-
PERCENTAGES			
*V*	70.5 ± 0.8 (68.6–71.9)	70.0, 68.4	71.0 ± 1.4 (69.0–74.0)
G or T	14.9 ± 0.7 (13.9–16.0)	14.2, 12.9	13.0 ± 1.4 (11.0–15.0)
Stb-v/St	29.7 ± 5.6 (22.5–42.3)	23.0, 22.6	-
St/L	25.3 ± 1.4 (23.6–26.9)	27.3, 27.2	-
Ep/L	23.3 ± 0.9 (21.8–24.5)	25.1, 23.5	-
RATIOS			
*a*	21.7 ± 1.4 (19.7–24.3)	21.6, 23.3	23.9 ± 1.8 (21.5–27.3)
*b*	2.3 ± 0.1 (2.1–2.5)	2.2, 2.2	2.2 ± 0.1 (2.1–2.4)
*c*	11.7 ± 0.5 (11.0–12.5)	10.0, 11.9	11.7 ± 1.2 (8.4–14.0)
*c′*	3.8 ± 0.3 (3.4–4.3)	4.0, 3.5	4.1 ± 0.4 (3.5–4.8)

##### Description

See Munawar et al. [23].

##### Habitat and Locality

The studied population was extracted from the roots of black bamboo (*Phyllostachys nigra* (Lodd ex Lindt.) Munro) in Gainesville, Florida (latitude 27°61′224″ N, longitude 82°40′799″ W).

##### Molecular Characterization

One D2–D3 of 28S rRNA and one *COI* gene sequence were obtained in this study. In the D2–D3 of the 28S rRNA gene tree (Figure 3), *P. paralatescens* from Florida clustered with *P paralatescens* from China. Intraspecific sequence variation for *P. paralatescens* was up to 0.1%. In the *COI* gene tree, *P. paralatescens* formed a clade with *P. acti*, *P. aculentus* type A, and *P. aciculus*.

##### Diagnosis (Based on Florida Specimens Examined)

*Paratylenchus paralatescens* is characterized by the presence of vermiform females with three lateral incisures in lateral field, a long stylet (67–80 µm long), and absence of advulval flaps. The lip region is truncate and lacking small submedian lobes. The excretory pore is situated near the level of stylet knobs. The spermatheca is well developed. The tail is elongate-conoid, gradually tapering to form a subacute terminus. In addition, it is characterized by the presence of obese sedentary females, males lacking stylets, and J2 having a well-developed stylet. According to the species grouping by Ghaderi et al. [31], this species belongs to group 9 characterized by a stylet longer than 40 µm, three incisures in the lateral field, and a lack of advulval flaps.

##### Remarks

Among the species of group 9, *P. latescens* is the species morphologically closest to *P. paralatescens.* The morphological characters for the separation of these two species are discussed hereafter. The measurements of the two Florida populations from fishpole and timber bamboo fit those of the paratypes of *P. paralatescens*. These populations share the same morphological characters. We would like to point out that the position of the excretory pore in the middle of the stylet shaft as indicated in the diagnosis of this species is not a consistent character because in 15% of the examined Florida specimens, the excretory pore is located near the base of the stylet as in *P. latescens*. The wedge-like shape of the tail terminus of vermiform females, which is another character of diagnostic value in the original description of this species, is not a consistent feature because about 40% of the Florida specimens we examined have a tapered tail with a bluntly pointed terminus. Another diagnostic character that is not consistent across the populations of *P. paralatescens* is the value of ratio *a*. The range of this character varies from 22.0–29.0 in the paratypes to 19.7–24.3 in the population from fishpole bamboo. The latter range is very close to that of *P. latescens* (14.0–23.0). Our observations indicate that the rectum and anus are more visible in live than fixed specimens. Obese females were not found in the original description, but they were detected in the two Florida populations. They were like those reported for *P. latescens*. Their body is lemon-shaped, with the anterior portion often bent and the posterior portion after the vulva narrow and projecting like an opened sickle. Measurements of their stylet are available for one specimen only because in other specimens, the stylet that is cemented in the root tissues breaks inside the tissues when specimens are detached from the roots. An obese female’s metacorpus is enlarged, and its width can reach 36 µm; the isthmus is slender, slightly shorter than that of vermiform females, and the bulb has a dimension like that of immature females. The genital tract consists of a very enlarged and coiled ovary connected to a prominent spermatheca and uterus. The posterior body is embedded in an egg mass, which contains embryonated eggs 59–69 µm long and 28–36 µm wide. Egg masses adhere tightly to the cuticle and often mask the female body attached to the roots. J2 from fish pole has a stylet 11 µm long just as that (8.5–11.5 µm) from timber bamboo reported by Troccoli et al. [8] and unlike that of *P. latescens*, which is longer (13–16 µm). The detection in *P. paralatescens* of obese females like those of *P. latescens* complicates the separation of these two species. The results of our study, however, confirm the diagnostic value of some morphological characters in the vermiform females reported by Munawar et al. [23], which are useful for the separation of *P. paralatescens* from *P. latescens* such as the truncate shape of the lip region vs. rounded in *P. latescens*; the absence of submedian lobes vs. presence; and vulvar lips non-protruding vs. protruding. Across all of the populations of *P. paralatescens*, males and J2 have shorter spicules (17.5–19.5 vs. 18.0–23.0 µm) and stylets (8.5–11.5 vs. 13.0–16.0 µm), respectively, than those of *P. latescens*. In the absence of J2 and males, the morphological identification of *P. paralatescens* is not reliable and needs to be confirmed by molecular analysis.

In group 9, *P. aciculus* and *P. aculentus* are other species phylogenetically close to *P. paralatescens*. These two species can be separated from each other by the stylet length, which is longer in *P. paralatescens* than in *P. aciculus* and *P. aculentus* (67.0–77.7 vs. 61–69 and 55.5–64.3 µm, respectively).

In Florida, *P. paralatascens* has been detected in bamboo stands grown in gardens and in nurseries.

#### 2.2.9. *Paratylenchus straeleni* (De Coninck, 1931) Oostenbrink, 1960 [27,28]

(Appendix A, Table 13)

This species was described with the name *Procriconema straeleni* from moss by De Conink [27] in Liege, Belgium. Subsequently, it was transferred to the genus *Paratylenchus* Micoletzky, 1922 [58] by Oostenbrink [28]. The geographical distribution of *P. straeleni* includes many countries other than Europe, such as Canada, Iran, South Africa, Turkey, the United States, and others [31,59]. In Florida, Tarjan [7] described a morphologically similar species from Lake Alfred, Polk County, under the name of *P. sarissus*, which was considered a junior synonym of *P. straeleni* by Geraert [35]. Lehman [4] listed another *P. straeleni* population from north Florida in association with Sabal Palmetto. However, this report cannot be confirmed because no morphological data on this population are available. Recently, pin nematode populations morphologically similar to *P. straeleni* were found in regulatory nematode samples collected from live oaks in central Florida. Phylogenetic studies conducted by Sing et al. (2021) indicated that DNA sequences of the Belgian population of *P. straeleni* matched those of populations from California and North Carolina in the United States. The phylogenetic relationships of the Florida populations with others from different geographical areas were not known up to this study. Information on the morphological and molecular characters of these populations is provided in the following sections.

##### Description

*Female*: The body is small and slender, curved ventrad, and exhibits an open C-shaped body habitus when heat relaxed; the cuticle is finely annulated; the lateral field has four incisures; the lip region is narrow, truncated, and lacking submedian lobes (under LM). Labial framework sclerotization is weak; the pharyngeal region is a typical paratylenchoid type. The stylet is rigid and straight, and the cone is 75–80% of the total stylet length; stylet knobs are rounded; and the dorsal pharyngeal gland opening is 5.4–6.9 µm behind stylet knobs. The median pharyngeal bulb is elongated, bearing distinct large valves; the isthmus is slender, surrounded by a nerve ring; and the excretory pore is situated at the anterior end of the pharyngeal basal bulb. The body is slightly narrower posterior to the vulva; the ovary is outstretched and well developed; the spermatheca is not well defined in our specimens, while the crustaformeria is well developed; and the vulva has a transverse slit occupying more than half of the corresponding body width. Vulval lips are not protruding; the advulval flaps are distinct; and the post-uterine sac is absent. The anus is difficult to distinguish in some specimens; the tail is slender, conoid, almost straight or very slightly curved ventrad, finely annulated, and gradually tapers to form a finely rounded terminus.

**Table 13 plants-12-02770-t013:** Morphometrics of females, males, J2, and J4 of *Paratylenchus straeleni* associated with *Quercus virginiana* in Florida.

Population	Ocala, Marion County (CD3633—N21-01260-3)			
Character	♀ (Live)	♂ (Live)	J2 (Live)	J4 (Live)
*n*	10	7	2	4
L	333.7 ± 9.3 (317.7–352.0)	362.2 ± 14.2 (305.0–347.4)	226.7, 207.0	287.6 ± 48.1 (245.5–355.0)
Stylet length (St)	54.6 ± 2.2 (51.4–57.9)	-	38.6, 37.6	44.0 ± 0.8 (43.5–44.5)
Conus length	42.5 ± 1.1 (41.0–44.5)	-	28.7, 27.7	33.4 ± 0.8 (32.6–34.1)
Stylet shaft + knob height	12.1 ± 1.4 (10.4–14.4)	-	9.9 (n = 2)	10.7 ± 0.3 (10.4–10.9)
Knob width	3.7 ± 0.9 (3.1–3.9)	-	2.8, 2.9	3.0 ± 0.9 (2.9–3.1)
Knob height	1.8 ± 0.1 (1.5–1.9)	-	1.1, 1.3	1.3 ± 0.1 (1.2–1.3)
DGO	6.0 ± 0.5 (5.4–6.9)	-	-	5.0
Median bulb valve length	6.2 ± 0.5 (5.5–7.0)	-	4.5 (n = 2)	5.0 ± 0.0 (4.9–5.0)
Median bulb width	9.2 ± 0.6 (8.3–9.9)	-	8.0 (n = 2)	7.3 ± 0.4 (7.0–7.9)
Isthmus length	11.6 ± 1.6 (9.0–13.8)	-	-	10.5 ± 0.6 (9.9–11.0)
Pharynx length	104.4 ± 5.0 (95.0–111.8)	90.0 ± 1.4 (88.6–93.2)	77.0, 89.0	87.6 ± 0.5 (87.0–88.1)
Anterior end to excretory pore (Ep)	85.4 ± 3.7 (80.0–90.9)	77.9 ± 1.8 (75.2–80.2)	63.0, 60.4	74.2 ± 6.4 (68.3–83.0)
Max body width	15.8 ± 1.0 (14.8–18.3)	12.3 ± 0.3 (11.9–12.8)	11.0, 12.0	14.9 ± 1.6 (12.9–17.3)
Body width at vulva	13.8 ± 0.8 (12.8–14.8)	-	-	-
Body width at anus	9.0 ± 0.6 (8.0–9.8)	9.0 ± 0.4 (8.1–9.4)	7.0, 7.4	7.9
Anterior end to median bulb base	77.7 ± 2.6 (72.3–81.2)	-	49.5 (n = 1)	63.2 ± 2.2 (61.0–65.3)
Lateral field width	3.5 ± 0.2 (3.2–4.0)	-	-	-
Anterior end-vulva distance	274.8 ± 9.4 (257.4–291.0)	-	-	-
Vulva-tail terminus distance	58.9 ± 2.0 (54.0–61.0)	-	-	-
Genital tract length	110.5 ± 19.8 (89.0–141.5)	96.4 ± 9.1 (84.1–113.8)	-	-
Vulva-anus distance	31.7 ± 2.7 (27.0–35.6)	-	-	-
Tail length	23.4 ± 1.6 (21.0–25.2)	26.3 ± 2.1 (24.7–30.0)	18.0 (n = 2)	19.4 ± 0.6 (18.8–20.0)
Stylet base (Stb) to median bulb valve base (v)	15.6 ± 1.8 (13.0–17.8)	-	10.0 (n = 1)	12.4 ± 2.4 (10.0–14.8)
Spicule length	-	19.8 ± 0.3 (19.3–20.2)	-	-
Gubernaculum length	-	4.6 ± 0.2 (4.5–5.0)	-	-
PERCENTAGES				
*V*	82.3 ± 0.8 (81.0–83.8)	-	-	-
G or T	33.0 ± 5.6 (26.4–40.7)	29.5 ± 2.5 (26.8–34.5)	-	-
Stb-v/St	28.6 ± 3.0 (23.4–33.6)	-	26.5 (n = 1)	28.2 ± 5.8 (22.4–34.0)
St/L	16.3 ± 0.6 (15.2–17.1)	-	17.0, 18.6	17.4 ± 0.4 (17.0–17.7)
Ep/L	25.5 ± 1.0 (24.0–27.6)	24.6 ± 1.7 (22.9–28.2)	27.7, 29.1	26.1 ± 2.0 (23.3–27.8)
RATIOS				
*a*	21.1 ± 1.2 (18.4–22.7)	26.5 ± 0.5 (25.6–27.1)	20.6, 17.3	21.2 ± 2.6 (18.3–23.6)
*b*	3.2 ± 0.1 (2.9–3.3)	3.6 ± 0.1 (3.3–3.7)	2.9, 2.3	3.3 ± 0.5 (2.8–4.0)
*c*	14.3 ± 1.2 (13.0–16.3)	12.4 ± 0.9 (11.2–13.8)	12.5, 11.5	13.8 ± 1.0 (13.0–15.2)
*c′*	2.6 ± 0.3 (2.2–3.0)	2.9 ± 0.2 (2.6–3.3)	2.5, 2.4	2.3

*Male*: The body is more slender than the female body, tapering towards both ends and curved to open C-shaped when heat relaxed. The cuticle is apparently smooth with fine annulations; the labial region is similar to that of females but narrower and slightly truncated, continuous with the body; sclerotization in the labial region is weak; and the stylet is lacking. The pharynx is rudimentary and non-functional, procorpus, and metacorpus, and the basal bulb is inconspicuous; and the excretory pore is located 75–80 µm away from the anterior end. The testis is outstretched, with small and rounded spermatozoa; the spicule is slender, slightly curved towards the end; the gubernaculum is curved; and the bursa is absent. The tail is conoid, tapering gradually to a finely pointed tip.

*Juveniles*: Second-, third-, and fourth-stage juveniles are similar in morphology to the adult females. Detailed information on their morphological characteristics were provided by Brzeski & Háněl [60], who observed the presence of stylets in all the juvenile stages in a population from the Czech Republic. Van den Berg & Tiedt [61] report the morphometrics of a J4 from South Africa. The Florida population we examined contained J2 and J4 with stylets, but we missed the J3. The J4 was encased in the J2 and J3 cuticles that retained the molted stylet of the J2. Morphometrics of J2 and J4 did not differ from those of the populations from the Czech Republic and South Africa. J2 and J4 stylets are shorter than that of the adult females; the pharynx is well developed; and the anus is almost indistinct. The tail is conoid and curved ventrad, ending in a finely pointed tip.

##### Habitat and Locality

The population of this species was collected from the rhizosphere of live oak (*Quercus virginiana* Mill.) in a tree farm in Ocala, Florida, USA (latitude 29°23′520″ N, longitude 82°50′161″ W). Other populations with similar morphological characteristics were detected in hard forests in central Florida.

##### Molecular Characterization

Two D2–D3 of 28S rRNA, two ITS rRNA, and one *COI* gene sequence were obtained for this species. In the D2–D3 of the 28S rRNA gene tree (Figure 3), Florida *Paratylenchus straeleni* clustered with other representatives of this species from distant geographical areas in separated subclades. In the ITS rRNA gene tree (Figure 4), two sequences of this species (Iran and USA) stand apart. In the *COI* tree (Figure 12), the sequences of the Florida populations were molecularly distant from those of other populations from other geographic areas.

##### Diagnosis (Based on Specimens Examined)

*Paratylenchus straeleni* is characterized by the presence of four incisures in the lateral field, a long stylet (51–58 µm long), and the presence of advulval flaps. The lip region is truncate and lacking small submedian lobes. The excretory pore is situated at the anterior end of the pharyngeal basal bulb. The tail is elongate-conoid, gradually tapering to form a finely pointed tip. In addition, the stylet is lacking in males but is present in J2–J4. According to the species grouping by Ghaderi et al. [31], this species belongs to group 10 characterized by a stylet longer than 40 µm, four incisures in the lateral field, and the presence of advulval flaps.

##### Remarks

The Florida specimens herein studied agree well with the populations of *P. straeleni* reported in the literature from different geographical areas for having the same ectoparasitic migratory habits and morphological features. However, these populations are not genetically homogeneous.

### 2.3. Molecular Characterization and Morphological Illustrations of Paratylenchus spp. Populations from Localities Other than Florida

#### 2.3.1. *Paratylenchus hamatus* Thorne & Allen, 1950 [18]

This species was described from a fig tree (*Ficus carica* L.) in California and is characterized by females having a stylet 25–33 µm long and a tail often curved, ending in a finely to broadly rounded terminus, and molecularly characterized from the type locality by Van den Berg et al. [30]. According to the species grouping by Ghaderi et al. [31], *P. hamatus* belongs to group 3 characterized by a stylet shorter than 40 µm, four incisures in the lateral field, and the presence of advulval flaps. Populations of *P. hamatus* have been reported from many geographical areas worldwide [31,59]. In this study, three populations of *P. hamatus* from fruit and ornamental trees were collected in California and another undetermined locality in the USA. These populations were used only for molecular and phylogenetic analyses without illustrations of their morphology.

##### Description

See Thorne & Allen [18].

##### Habitat and Locality

Two of the studied populations were collected from peach (*Prunus persica* L.) and grapevine (*Vitis vinifera* L.) in Kern County, California, and another from ornamental pear (*Pyrus malus* L.) in an undetermined locality in the USA.

##### Molecular Characterization

One sequence of D2–D3 of the 28S rRNA gene was obtained for each of the three populations. In addition, one ITS rRNA gene sequence was obtained for the population from an undetermined locality in the USA. In D2–D3 of the 28S rRNA phylogenetic tree (Figure 3), the sequences of three populations of *P. hamatus* grouped together and clustered in a clade containing two unidentified species and a population of *P. tenuicaudatus* Wu, 1961 [62], a morphologically closely related species. In the ITS rRNA gene tree (Figure 4), the population of *P. hamatus* clustered also in a clade containing a population of *P. tenuicaudatus* and others of unidentified species.

#### 2.3.2. *Paratylenchus hamicaudatus* (Cid del Prado Vera & Maggenti, 1988) Brzeski, 1998 [19,20]

(Figure 18)

This species was described from the roots of *Sequoia sempervirens* (D. Don) Endl. in California and is characterized by having obese egg-laying females that are often hook-like in shape, with the post-vulvar portion of the body constricted and shaped finger-like. According to the species grouping by Ghaderi et al. [31], *P. hamicaudatus* belongs to group 10 characterized by a stylet longer than 40 µm, four incisures in the lateral field, and the presence of advulval flaps.

No DNA sequences of this species have been previously deposited in the GenBank. In this study, a population with the same morphological characters was collected from the same host and type locality and used for molecular and phylogenetic analyses.

**Figure 18 plants-12-02770-f018:**
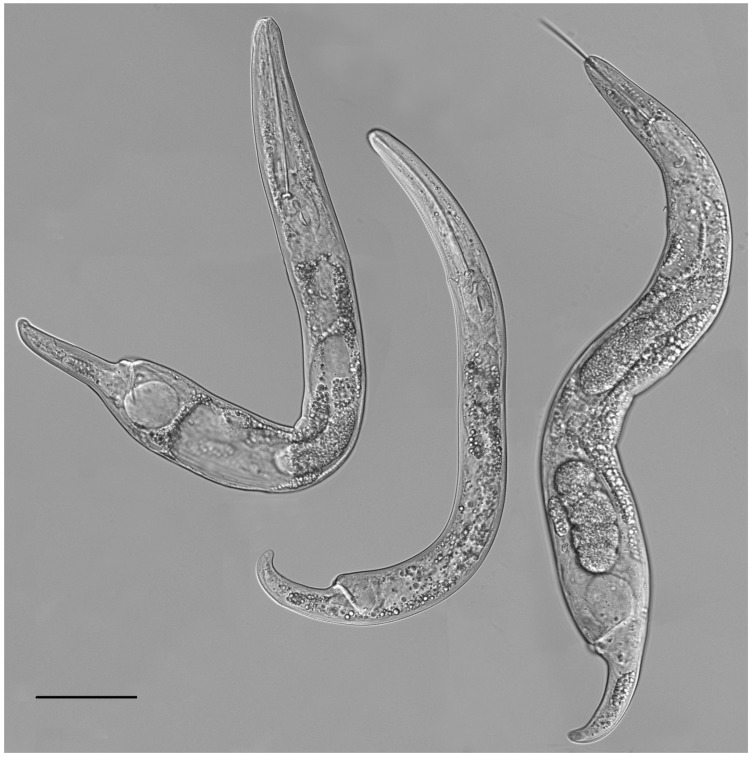
Light microscopic photos of females of *P. hamicaudatus* (CD3447) collected from rhizosphere soil of *Sequoia sempervirens*, Lagunitas Lake (type locality), San Rafael, Marin County, CA, USA. (Scale bar = 50 μm).

##### Description

See Cid del Prado Vera & Maggenti [19].

##### Habitat and Locality

The studied population was extracted from the roots of redwood (*Sequoia sempervirens* (D. Don) Endl. in Lagunitas Lake, in Martin County, California (latitude 37°56′37.0″ N, longitude 122°35′49.2″ W).

##### Molecular Characterization

One sequence each of D2–D3 of 28S rRNA, ITS rRNA, and *COI* genes were obtained for the population of this species. In D2–D3 of the 28S rRNA (Figure 3) and *COI* (Figure 12) gene phylogenetic trees, *P. hamicaudatus* clustered in a clade containing species having stylets longer than 70 µm and obese females such as *P. ilicis* Singh, Lokker, Couvreur, Bert & Karssen, 2022 [63].

#### 2.3.3. *Paratylenchus holdemani* Raski, 1975 [21]

(Figure 10D and Figure 11D)

This species was described from coffee (*Coffea arabica* L.) in La Florida, El Salvador. It is closely related morphologically to *P. hamatus,* from which it can be separated by the shorter stylet of 22–23 vs. 25–33 µm. In the classification by Ghaderi et al. [31], *P. holdemani* belongs to group 3 characterized by a stylet shorter than 40 µm, four incisures in the lateral field, and the presence of advulval flaps. In this study, one population each of this species was collected in California and Spain. Morphological illustrations were obtained only for the California population.

##### Description

See Raski [21].

##### Habitat and Locality

The studied populations were collected from an unidentified host in Marin County, California, and holly (*Quercus ilex* L.) in Villaviciosa de Odón, Madrid, Spain (latitude 40°22′16.6″ N, longitude 3°56′41.0″ W).

##### Molecular Characterization

One sequence of D2–D3 of the 28S rRNA gene was obtained for each of the two populations. In addition, one sequence each of ITS rRNA and *COI* gene were obtained for the population from an undetermined locality in Spain. In D2–D3 of the 28S rRNA tree (Figure 3), the sequences of the two populations grouped together with other populations of this species and clustered in a clade containing *P. labiosus* Anderson & Kimpinski, 1977 [64], a species belonging to the same classification group 3. In the ITS rRNA (Figure 4) and *COI* (Figure 12) gene phylogenetic trees, the population of *P. holdemani* from Spain grouped with other populations of this species from other geographical areas.

#### 2.3.4. *Paratylenchus pedrami* Clavero-Camacho, Cantalapiedra-Navarrete, Archidona-Yuste, Castillo & Palomares-Rius, 2021 [25]

(Figure 10I,J and Figure 11I,J)

This species was described from almond trees in Córdoba, Spain. It belongs to group 3 in the classification proposed by Gadheri et al. [31] for having females with stylets shorter than 40 µm, four incisures in the lateral field, and the presence of advulval flaps. It is related morphologically to some of the species in this group such as *P. baldaccii* Raski, 1975 [21]. In this study, two populations of this species were collected in California. Morphological illustrations were obtained for one (CD3609) of these populations.

##### Description

See Clavero-Camacho et al. [25].

##### Habitat and Locality

The studied populations were collected from grapevine (*Vitis vinifera* L.) and an undetermined host in Yolo and San Jose Counties, California.

##### Molecular Characterization

One sequence each of D2–D3 of 28S rRNA, ITS rRNA, and *COI* genes were obtained for the population from Yolo County as well as one sequence each of D2–D3 of 28S rRNA and *COI* genes for that of San Jose County. In D2–D3 of the 28S rRNA gene tree (Figure 3), the sequences of the two populations grouped together with other populations of this species from Spain and clustered in a clade containing *P. baldaccii*. The close relationship of *P. pedrami* populations from California and Spain with *P. baldaccii* was also observed in the *COI* gene tree (Figure 12).

#### 2.3.5. *Paratylenchus projectus* Jenkins, 1956 [26]

(Figure 10K and Figure 11K)

This species was described from a pasture grass, in Maryland, USA. It has a wide geographical distribution and host range and is a representative of group 3 in the classification proposed by Gadheri et al. [31] for having females with stylets shorter than 40 µm, four incisures in the lateral field, and the presence of advulval flaps. In this study, one population of this species was intercepted in Florida in a consignment of strawberry runners from Canada, and another was collected in Spain. Morphological illustrations were obtained only for the Canada population.

##### Description

See Jenkins [26].

##### Habitat and Locality

The studied populations were collected in strawberry runners (*Fragaria x ananassa* Duchesne) in a consignment from Symcoe, Ontario, Canada, and rhizosphere of a grapevine (*Vitis vinifera* L.) in Villamanta, Madrid, Spain (latitude 40°20′01.1″ N, longitude 4°05′52.1″ W).

##### Molecular Characterization

One sequence each of D2–D3 of the 28S rRNA gene was obtained for the two populations from Canada and Spain as well as one additional sequence of ITS rRNA and *COI* genes for the latter population. In D2–D3 of the 28S rRNA gene tree (Figure 3), the sequence of the population from Canada grouped together with other sequences of populations of this species and clustered in a clade that contained the sequence of the population from Spain, which clustered together with sequences of *P. neoprojectus* Wu & Hawn, 1975 [65], indicating that the identification of this species as *P. projectus* may be incorrect. In the ITS rRNA (Figure 4) and *COI* (Figure 12) genes trees, this population from Spain did not group together with those of *P. projectus*, confirming the uncertainty of its taxonomic status.

### 2.4. Phylogenetic Relationships of Paratylenchus Species

The D2–D3 of the 28S rRNA gene alignment included 474 sequences of 66 valid and 11 unidentified *Paratylenchus* species and 3 sequences of outgroup taxa and was 739 bp in length. Twenty-six new sequences were included in this analysis. The Bayesian 50% majority rule consensus tree inferred from the partial 28S rRNA gene alignment is presented in Figure 3. Phylogenetic relationships within *Paratylenchus* are generally congruent to those given by several authors [14,26,30,66,67,68]. The sequences of *Paratylenchus* species are distributed among four major clades previously designated by Singh et al. [14], and *P. jasmineae* Phani, Somvanshi, Rao & Khan 2019 [69] formed a separate clade (IIIb).

The ITS rRNA gene alignment included 319 sequences of 58 valid and 7 unidentified *Paratylenchus* species and three sequences of outgroup taxa and was 1169 bp in length. Seventeen new sequences were included in this analysis. The Bayesian 50% majority rule consensus tree inferred from the ITS rRNA gene alignment is presented in Figure 4. The sequences of *Paratylenchus* species are distributed among four major clades, and *P. hawaiiensis* sp. n. formed a separate clade (IIIc).

The *COI* alignment included 354 sequences of 48 valid and 22 unidentified *Paratylenchus* species and 3 sequences of outgroup taxa and was 419 bp in length. Twenty new sequences were included in this analysis. The Bayesian 50% majority rule consensus tree inferred from the *COI* gene alignment is presented in Figure 12. All sequences of *Paratylenchus* were distributed among four major clades.

The combined alignment included 67 sequences of valid *Paratylenchus* species and 3 sequences of the outgroup taxa and was 2072 bp in length. The Bayesian 50% majority rule consensus tree inferred from this alignment is presented in Figure 19 with mapping of stylet length, lateral field pattern, and presence of swollen and obese females. The sequences of *Paratylenchus* species are distributed among four major clades, and *P. hawaiiensis* sp. n. formed a separate clade (IIIc). Clade I contains *Paratylenchus,* having vermiform females with four incisures and stylets shorter than 40 µm, except for species belonging to three subclades: (i) *P. straeleni*, *P. parastraeleni*; (ii) *P. plesiostraeleni*; and (iii) *P. goodeyi*, all having longer stylets. Clades II and IV contain *Paratylenchus*, having vermiform and swollen and obese females with three or four incisures and stylets longer than 40 µm. Clade III contains *Paratylenchus*, having vermiform females with three or four incisures and stylets shorter than 40 µm. Thus, obese females were reported in two *Paratylenchus* clades with long stylets.

## 3. Discussion

The results of this and previous studies indicated that in Florida, environments contain numerous *Paratylenchus* species. Apart from *Paratylenchus hawaiiensis* sp. n. (=*P. aquaticus* type A), *P. minutus* (=*P. shenzhenensis* syn. n.), and *Paratylenchus* sp.3, which have been characterized molecularly in previous taxonomic work [14], a total of ten species: *Paratylenchus hawaiiensis* sp. n., *P. roboris* sp. n., *P. acti*, *P. aquaticus*, *P. goldeni*, *P. paralatescens*, *P. minutus* (=*P. shenzhenensis* syn. n.), and *P. straeleni* and two undescribed species were identified in this study. Other species reported (Appendix A) lack DNA sequences, and they were identified based on their morphological characteristics only. These morphological identifications supplemented by molecular characterization should stand until phylogenetic analyses using DNA sequences of the topotype material of these species are conducted, confirming or disproving their validity.

Some of the species such as *P. paralatescens* and *P. minutus* (=*P. shenzhenensis* syn. n.) with ectoparasitic sedentary and endoparasitic migratory habits, respectively, may have been introduced into Florida from the Far East in the roots of *Phyllostachys* sp. and *Hemerocallis* sp. propagative material. The latter species damages anthurium plants in China and pineapple in Hawaii. However, its impact on daylily growth in Florida has not been assessed. Other species, such as *P. goldeni,* may be native to southeastern USA, where it was associated with many plants. The role of the other species in damaging their hosts is not known. There is a great diversity in the parasitic habits of the species we found. *Paratylenchus aquaticus* (type C) and *P. minutus* (=*P. shenzhenensis* syn. n.) have endoparasitic migratory habits and vermiform egg-laying females. The parasitic habit of *P. hawaiiensis* sp. n. is not known. In contrast, *P. goldeni*, and *P. straeleni* are ectoparasitic migratory species and also have vermiform egg-laying females and coiled and quiescent J3 and J4. The remaining *P. acti*, *P. paralatescens*, and *P. roboris* sp. n. are ectoparasites but with sedentary and obese females and coiled and quiescent J3 and J4.

In this study, the host status of the plants we sampled was verified with certainty for the nematode with endoparasitic migratory or ectoparasitic sedentary habits. There is no certainty for the host status of plants associated with ectoparasitic migratory species because of the difficulty in localizing the nematode in the roots. The results of our study elucidate the molecular characteristics of only four species (*P. aquaticus* (type C), *P. goldeni*, *P. minutus,* and *P. straeleni*) on the long list of pin nematodes reported from Florida by Esser [3] and Lehman [4]. Molecular data were obtained for additional two new species, *P. hawaiiensis* sp. n. and *P. roboris* sp n. and *P. acti,* reported for the first time in Florida, along with two pin nematode species (*Paratylenchus* spp. FL1 and FL2, see Appendix A), which are still unidentified.

Except for the topotype population of *P. hamicaudatus* from California and the Florida populations of *P. acti* and *P. hawaiiensis* sp. n. belonging to groups 10, 11, and 2, respectively, in the classification of Gadheri et al. [31], all of the other studied populations of *Paratylenchus* species from localities other than Florida belong to group 3. In the phylogenetic trees, these species clustered in clades containing morphologically related species also belonging to group 3, indicating that *Paratylenchus* species morphologically and biological related for having only sessile females with stylet lengths less than 40 and with ectoparasitic migratory habits are also related genetically. This morphological and genetic relationship was also observed for species having obese females with stylets longer than 40 µm and ectoparasitic sedentary habits such as *P. hamicaudatus* that grouped together with *P. ilicis*, a species having a similar body shape to the swollen females and the same parasitic habits. However, these two species having the hook-shaped body of swollen females and lateral fields marked by four incisures or three bands did not cluster in the same clade as other species having ectoparasitic sedentary habits and swollen females with a lateral field marked by four or three incisures and a lemon-shaped body narrowing after the vulva and projecting like an opened sickle, such as *P. acti* and *P. paralatescens.* We would like to point out that former *Cacopaurus pestis,* a species recently reclassified as *Paratylenchus pestis* by Singh et al. [63] (see below), did not cluster in the clades containing *P. hamicaudatus* and *P. ilicis*, which have a similar body shape to swollen females and parasitic habits.

Regarding to the identity of some populations of pin nematodes, there are some contradictions. For example, in the D2–D3 of the 28S rRNA gene tree (Figure 3), the only available sequence of *Paratylencus colinus* clustered together with *P. aciculus* in a clade containing a subclade with *P. acti*. We would like to point out that this sequence belongs to a population from Iran that was identified incorrectly as *P. colinus* by Mokaram Hesar et al. [70], despite the morphology and morphometrics of this population fitting those of *P. aciculus*, and then we herein regarded the Iranian population as *P. aciculus*. Ghaderi et al. [31] accommodated *P. colinus* in group 8 because this species has advulval flaps in addition to stylets shorter than 40 μm and three incisures in the lateral field. However, these advulval flaps are not shown in the drawing of the single specimen used for the description of this species from Brazil by Huang & Raski [50]. Moreover, there is no mention of the presence of advulval flaps in the redescription of this species using a population from Argentina by Doucet [71]. In this redescription, many differential morphological characters such as the position of the excretory pore and anus are missed. The insufficient number of specimens used for the original description of *P. colinus* and the lack of molecular and morphological differential characters such as the position of the excretory pore and anus in the redescription by Doucet [71] make the identification of this species unreliable. Therefore, the taxonomic position of this species remains uncertain until supplemental morphological and molecular data of the original population of *P. colinus* from Brazil are obtained. On the other hand, Zhuo et al. [72] identified a Chinese population of *Paratylenchus* as *P. aculentus* and provided molecular data for this population. According to these authors, the sequence with GenBank accession number KR270600 belongs to the D2–D3 of the 28S rRNA gene and the sequence KR270597 to 18S rRNA gene. This information is incorrect because the sequences are labeled incorrectly; the sequence KR270597 really belongs to the D2–D3 of the 28S rRNA gene, and KR270600 really belongs to 18S rRNA gene. This population is herein identified in the molecular study as *P. aculentus* type B (see below).

The genus *Gracilacus* Raski, 1962 [11] was proposed for representatives of the *Paratylenchus* species with stylet lengths longer than 48 µm and the excretory pore generally situated near or anteriorly to metacorpus [11]. Brzeski [43] and Siddiqi and Goodey [12] synonymized *Gracilacus* with *Paratylenchus*; however, this action was not accepted by many authors. Siddiqi [73] treated it as a subgenus of *Paratylenchus*. After the reconstruction of phylogenetic relationships using rRNA gene sequences, Van den Berg et al. [30] concluded that the position of representatives with long stylets was not resolved, and the validity of *Gracilacus* could be rejected using available datasets at that time. With the inclusion of additional sequences of different *Paratylenchus* species, Mokaram Hesar et al. [70], Clavero-Camacho et al. [25,66], and Palomares-Rius et al. [68] concluded that molecular data support the synonymy of *Gracilacus* with *Paratylenchus* because stylet length in *Paratylenchus* has evolved independently several times during the evolution of this genus. The present study also confirmed this conclusion.

The genus *Cacopaurus* Thorne, 1943 [56] was proposed and distinguished from *Paratylenchus* by the obese female body, tubercles on the annuli of the female cuticle, and sessile parasitism [56]. The genus contained only one species, *Cacopaurus pestis,* parasitising the walnut, *Juglans regia*, in California, USA. It was further reported from France, Spain, Italy, and Iran [59]. Although Goodey [57] synonymized *Cacopaurus* with *Paratylenchus*, the validity of *Cacopaurus* was accepted by many authors [14,69]. The first molecular characterization of *C. pestis* collected in Iran was made by Mokaram Hesar et al. [70], who concluded that the phylogenetic results obtained from the 28S and ITS rRNA gene sequence analysis did not support the validity of *Cacopaurus* because *C. pestis* clustered with *Paratylenchus* clades. Recently, Singh et al. [63] described a new species, *P. ilicis,* from holly plant (*Ilex aquifolium* L.) from The Netherlands, and this species had all typical characteristics of the genus *Cacopaurus.* The molecular data and resulting phylogenetic relationships showed that *P. pestis* (=*C. pestis*) and *P. ilicis* were embedded in two different *Paratylenchus* clades and, therefore, Singh et al. [63] stated that *Cacopaurus* could be considered a junior synonym of *Paratylenchus*. Our present phylogenetic analysis also confirmed this statement on the synonymization. However, the phylogenetic position of *P. pestis* with the *Paratylenchus* clade still remains contradictive and requires additional study. In the D2–D3 of 28S and ITS rRNA gene trees presented by Mokaram Hesar et al. [70], *P. pestis* clustered with *P. minutus* (=*P. shenzhenensis* syn. n.), whereas, in the results of our analysis, the sequences deposited by Mokaram Hesar et al. [70] in the GenBank as *P. pestis* also clustered with *P. minutus* in the D2–D3 of 28S gene tree (Clade IIIa). In the ITS rRNA gene, this species clustered with *P. peraticus* tree (Clade II). A similar pattern of *P. pestis* positions can be also observed from the phylogenetic trees given by Singh et al. [63]. We suspect that some contaminated sequence belonging to other *Paratylenchus* species was obtained and deposited in the GenBank instead of *P. pestis.* It is likely that the correct sequence and position of *P. pestis* were in the ITS rRNA gene tree, where this species clustered with *Paratylenchus* having an elongated stylet.

Raski [11] was the first who noticed that the occurrence of obese females in (*P. pestis*, *P. epacris,* and others) appeared to be associated with the elongated stylet in the female. He proposed that the obesity of these species derived from their evolution as a sessile type of parasite, which no longer migrates in search of food once feeding is started, and concluded that more information was needed to understand the nature and taxonomic importance of the swollen female stage of these various species. Our results showed that the obese shape of females reported in *P. acti*, *P. hamicaudatus*, *P. ilicis*, *P. paralatescens*, *P. peraticus*, *P. pestis*, *P. roboris,* and *P. sinensis*, which have stylets > 40 µm, are distributed within two clades (II and IV; see Figure 19). The body shape and parasitic habits of females are the most variable in the genus *Paratylenchus* than in any other genus in the family Tylenchulidae. With the exception of *Tylenchocriconema* Raski & Siddiqui, 1975 [74] containing one species and *Boomerangia* Siddiqi, 1994 [75], two species of unknown parasitic habits, females of species in the remaining four genera *Meloidoderita* Poghossian, 1966 [76], *Sphaeronema* Raski & Sher, 1952 [77], *Trophotylenchulus* Raski, 1957 [78], and *Tylenchulus* Cobb, 1913 [79] are consistently obese and sedentary semi-endo-parasites. On the contrary, *Paratylenchus* species encompass many aspects of parasitic modes by behaving as migratory ecto- or endoparasites to sedentary ecto-parasites inducing specialized feeding sites [9,19,80,81]. The biological variability of these species is not well reflected in the phylogenetic trees published in the literature. However, the results of our phylogenetic analyses show that sedentary obese species cluster together within two clades, which may indicate that these sedentary and obese species are also phylogenetically different from the sessile ones having migratory habits.

Presently, the pin nematodes comprise 142 species [68,82], and only 69 (Appendix A) of them are considered to be molecularly characterized. In the last decade, descriptions of new *Paratylenchus* species have been accomplished with one or several gene sequences obtained from type specimens. Molecular characterizations of most *Paratylenchus* species described before 2010 were made only in recent years and based on specimens identified morphologically, but they were not collected from type localities and hosts. The only exceptions are five species: *P. bilineatus*, *P. hamatus*, *P. nanus*, *P. hamicaudatus*, and *P. minutus*, which were analyzed molecularly using specimens collected from type localities or places close to type localities (Appendix A). Because the identification of *Paratylenchus* is difficult due to limited diagnostic features and morphological plasticity, incorrect species naming could not be excluded. Moreover, an analysis of available morphological, morphometric, and molecular data indicated the presence of species complexes within this genus: *P. aciculus/P. aculentus* species complex, *P. aquaticus* species complex, *P. hamatus* species complex, *P. goldeni* species complex, *P. microdorus* species complex, and *P. straeleni* species complex [14,30,66], and it makes identification really challenging work. Incorrect morphological identification would lead to the providing of non-appropriate reference sequence information for certain known species. We encourage our colleagues to pay more attention to this problem by devoting more efforts to collect the topotype materials of known *Paratylenchus* species to be used for molecular study.

## 4. Materials and Methods

### 4.1. Nematode Populations

Nematode populations used in this study were obtained from 110 soil samples collected from Florida ornamental and landscape plants and additional 15 samples from coffee, grapevine, hard wood forest trees, and strawberry from other localities listed in Table 1. Several pin nematodes collected from the USA (Alaska, California, Hawaii, Oregon, and Washington) and Spain were also included in this study. Samples consisted of moist soil and feeder roots collected under the canopy of these plants at a depth of 10–15 cm. Nematodes were extracted from soil using the centrifugal flotation method [83] and root incubation in jars [84]. The latter technique allowed the detection of ectoparasitic sedentary and endoparasitic migratory pin nematode species. Extracted specimens were used for morphological and molecular analyses.

### 4.2. Light Microscopic Study and Morphological Identification

Specimens were hand-picked in tap water, immobilized by gentle heating, and mounted in water agar on a slide for measurements and photographs using a modified technique described by Esser [85]. Additional specimens were killed and fixed using Golden’s method described in Southey [86]. Live specimens were hand-picked with an eye lash, transferred into 2–3 mL of distilled water in a watch glass, and put in an oven at 43 °C. After 15 min, the watch glass was filled with a fixative (a water solution of 3% formaldehyde and 2% glycerol) that was kept in the oven at the same temperature. The watch glass was then enclosed in a petri dish and kept in an oven to allow a slow evaporation of the fixative and the infiltration of the nematode with glycerol for three-five days or longer. Specimens were then mounted permanent glass slides to allow handling and observation under LM. Measurements of specimens were made using a Nikon (Optiphot) ocular micrometer. Photographs were taken with a Zeiss compound microscope, AXIO Scope A1, equipped with Nomarski interference contrast and an AxioCam ICc5 (Germany). Measurements of additional fixed specimens were taken with an Olympus BX51 (Olympus, Tokyo, Japan) ocular micrometer. Morphometrics included de Man’s indices and standard measurements. Drawings were made using a camera lucida attached to the microscope. Some of the best-preserved specimens were photographed with the same microscope equipped with differential interference contrast and a Canon EOS 250D digital camera (Canon, Tokyo, Japan). Digital images were edited using Adobe^®^ Photoshop^®^ CS (Adobe Systems, San Jose, CA, USA).

Species delimitation of *Paratylenchus* in this study was performed using an integrated approach that considered morphological and morphometric evaluation combined with molecular-based phylogenetic inference (tree-based methods) and sequence analyses (genetic distance methods).

### 4.3. DNA Extraction, PCR, and Sequencing

DNA was extracted from several specimens of each sample using the proteinase K protocol. DNA extraction, PCR, and cloning protocols were used as described by Subbotin [87]. The following primer set was used for PCR: the forward D2A (5′-ACA AGT ACC GTG AGG GAA AGT TG-3′) and the reverse D3B (5′-TCG GAA GGA ACC AGC TAC TA-3′) primers [88] for amplification of the D2–D3 expansion segments of the 28S rRNA gene; the forward TW81 (5′-GTT TCC GTA GGT GAA CCT GC-3′) and the reverse AB28 (5′-ATA TGC TTA AGT TCA GCG GGT-3′) primer for amplification of the ITS1-5.8-ITS2 rRNA gene; and the forward JB3 (5′-TTT TTT GGG CAT CCT GAG GTT TAT-3′) and the reverse JB4 (5′-TAA AGA AAG AAC ATA ATG AAA ATG-3′) or JB5 (5′-AGC ACC TAA ACT TAA AAC ATA ATG AAA ATG-3′) primers [89] or the forward COI-F5 (5′-AAT WTW GGT GTT GGA ACT TCT TGA AC-3′) and the reverse COI-R9 (5′-CTT AAA ACA TAA TGR AAA TGW GCW ACW ACA TAA TAA GTA TC-3′) primers [90] for amplification of the partial *COI* gene of mtDNA. The PCR products were purified using QIAquick (Qiagen) Gel or PCR extraction kits and submitted for direct sequencing or cloned using the pGEM-T Vector System II kit (Promega). Sequencing was conducted at Genewiz (CA, USA). The newly obtained sequences were submitted to the GenBank database under accession numbers as indicated in Table 1 and phylogenetic trees.

### 4.4. Phylogenetic and Sequence Analysis

The newly obtained sequences of the D2–D3 expansion segments of the 28S rRNA, the ITS of the rRNA, and partial *COI* mtDNA genes were aligned using ClustalX 1.83 [91] with corresponding published gene sequences [13,14,23,24,26,30,65,66,67,69,71,80,92,93,94,95,96,97]. Outgroup taxa for each dataset were chosen based on previously published data [30]. Three alignments containing sequences of each gene and a combined alignment containing the reference sequences of three genes for each valid species were analyzed with Bayesian inference (BI) at CIPRES Science Gateway [98], using MrBayes 3.2.7a [99]. The best-fit model of DNA evolution was obtained using jModelTest 2.1.10 [100] with the Akaike information criterion (AIC). The Akaike-supported model, the base frequency, the proportion of invariable sites, the gamma distribution shape parameters, and substitution rates in the AIC were then used in phylogenetic analyses. BI analysis under the general time reversible model with a proportion of invariable sites and a gamma-shaped distribution (GTR + I + G) was initiated with a random starting tree and run with the four Metropolis-coupled Markov chain Monte Carlo (MCMC) for 2 × 10^6^ generations. The topologies were used to generate a 50% majority rule consensus tree. Posterior probabilities (PPs) over 50% are given on appropriate clades. The trees were visualized with the program FigTree v1.4.3 and drawn with Adobe Illustrator CC. Sequence analyses of alignments were performed with PAUP∗ 4.0b 10 [101]. Pairwise divergences between taxa were computed as absolute distance values and as percentage mean distance values based on whole alignment, with adjustment for missing data.

## Figures and Tables

**Figure 1 plants-12-02770-f001:**
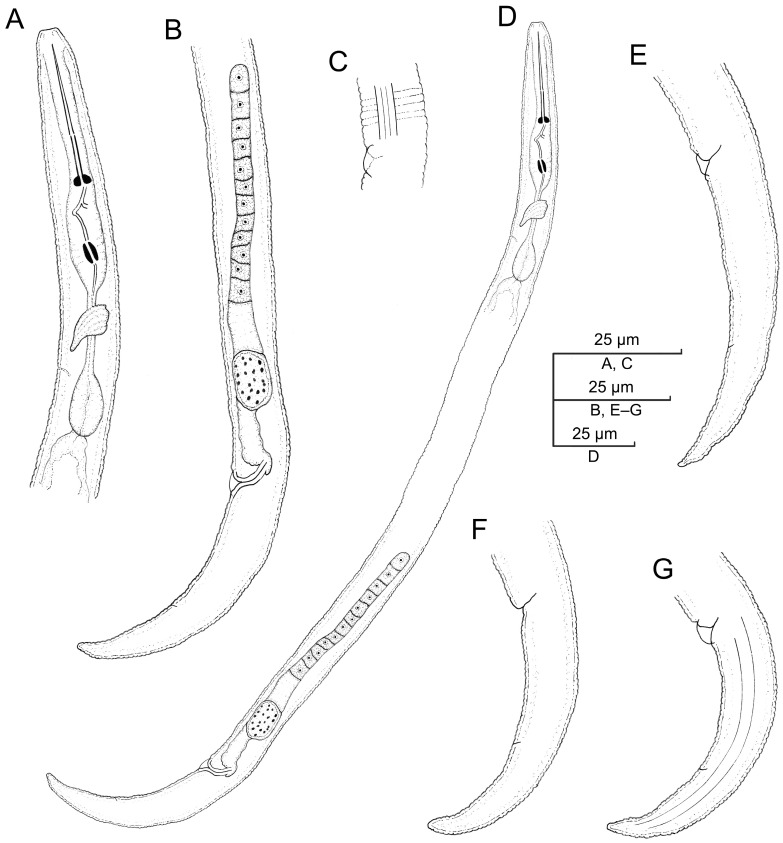
Camera lucida drawings of *Paratylenchus borealis* sp. n. female from Alaska (CD3781). (**A**): Anterior body region; (**B**): Genital track, vulval region, and posterior body region; (**C**): Lateral field; (**D**): Entire body; and (**E**–**G**): Shapes of posterior body region.

**Figure 2 plants-12-02770-f002:**
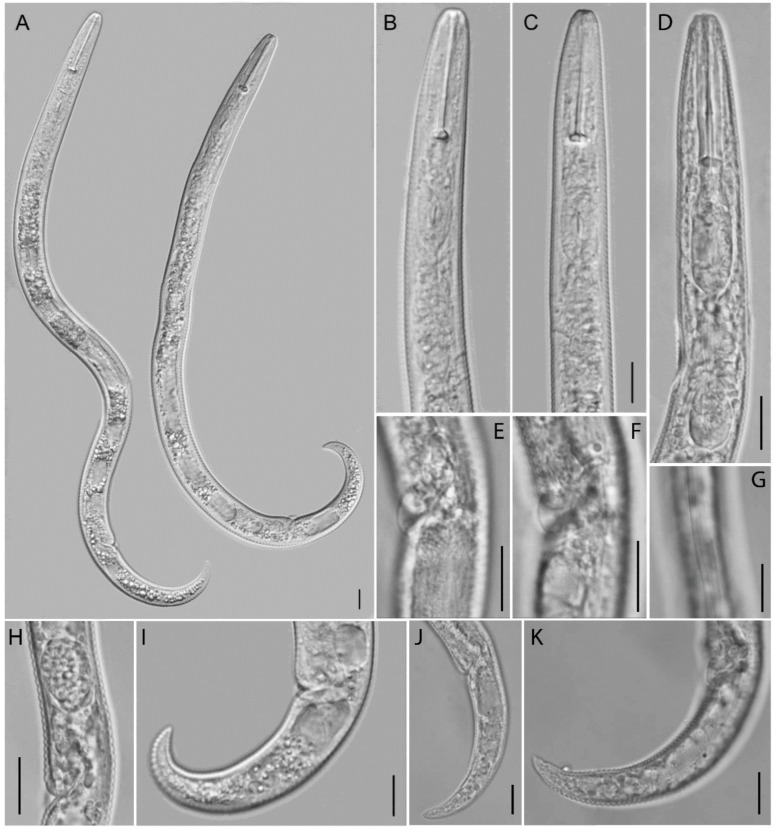
Light microscopic photos of *Paratylenchus borealis* sp. n. female from Alaska (CD3781). (**A**): Entire body; (**B**–**D**): Anterior body region; (**E**,**F**): Vulval region showing the prominent advulval flaps; (**G**): Lateral field marked by four incisures; (**H**): Vulval region and spermatheca; and (**I**–**K**): Posterior body region. (Scale bars = 10 μm).

**Figure 3 plants-12-02770-f003:**
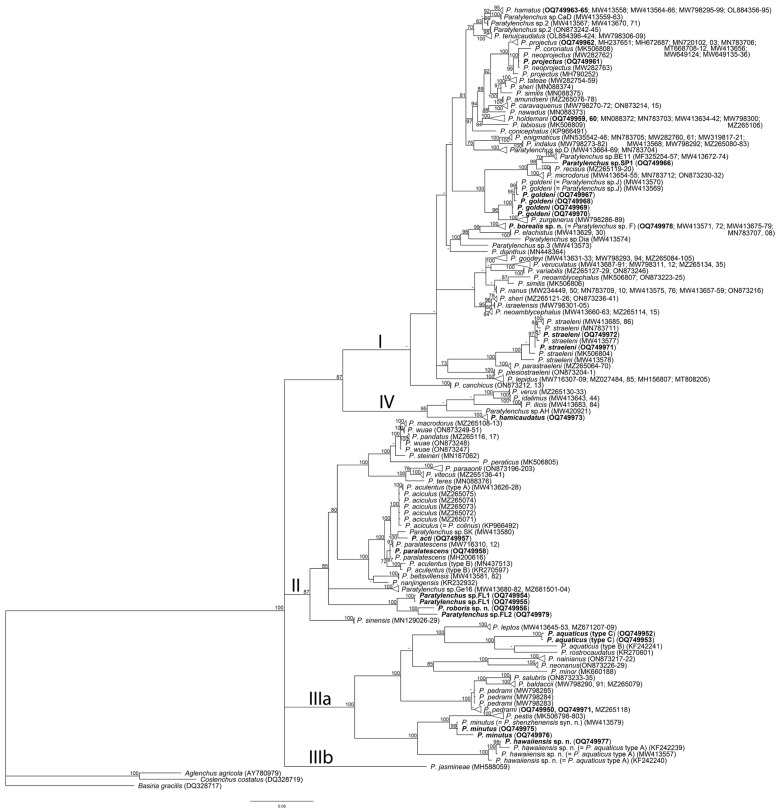
Phylogenetic relationships of *Paratylenchus* species as inferred from Bayesian analysis using the D2–D3 expansion segments of 28S rRNA gene sequence alignment under the GTR + I + G model. Posterior probability values more than 70% are given on appropriate clades. New sequences are indicated by bold letters.

**Figure 4 plants-12-02770-f004:**
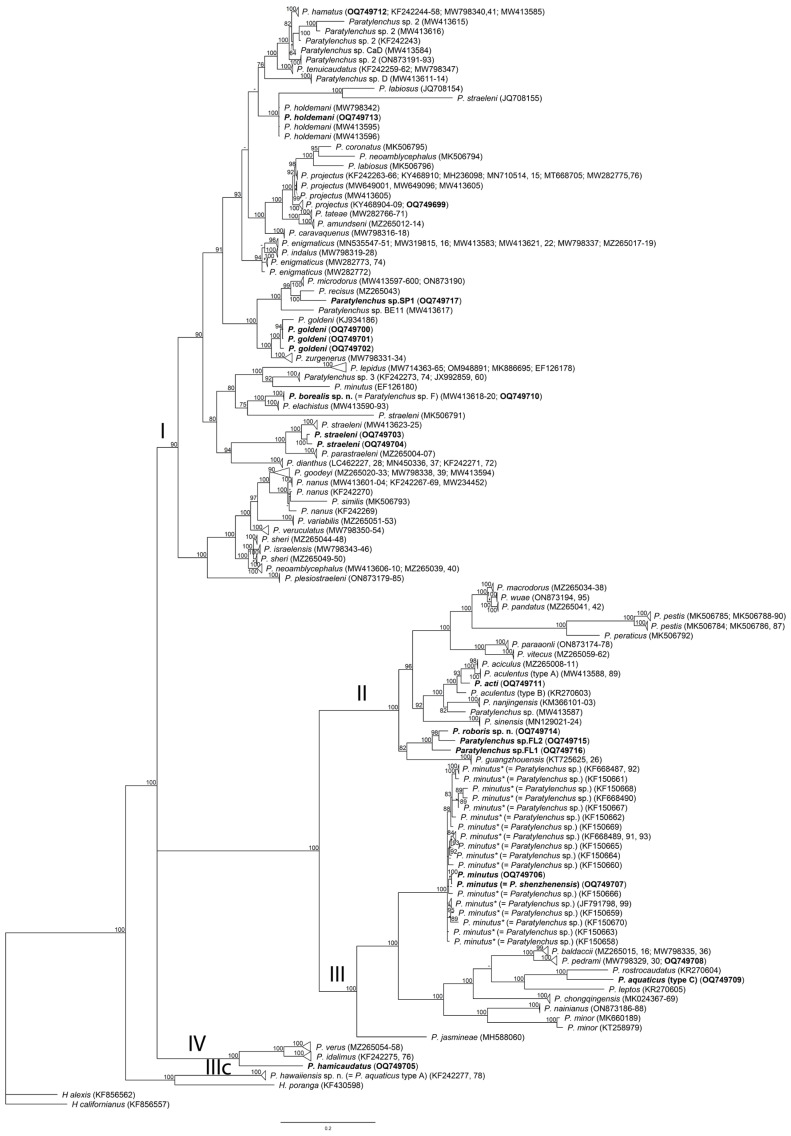
Phylogenetic relationships of *Paratylenchus* species as inferred from Bayesian analysis using the ITS rRNA gene sequence alignment under the GTR + I + G model. Posterior probability values more than 70% are given on appropriate clades. New sequences are indicated by bold letters. * Species deposited in GenBank as *Paratylenchus* sp. and herein considered as belong to *Paratylenchus minutus*.

**Figure 5 plants-12-02770-f005:**
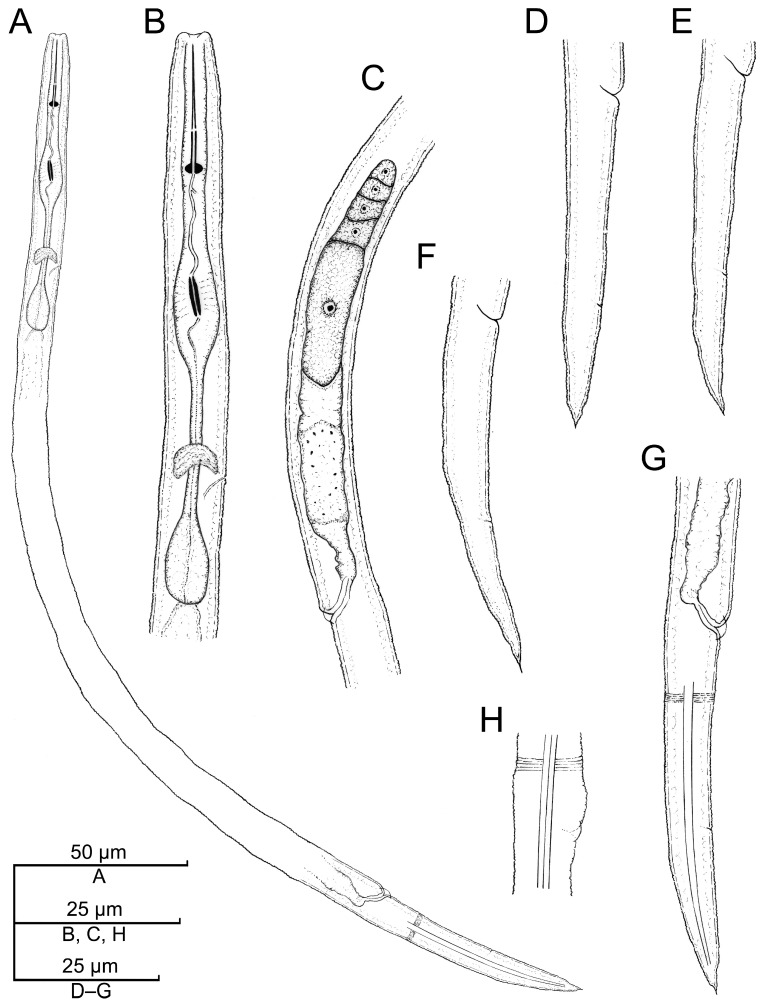
Camera lucida drawings of *Paratylenchus hawaiiensis* sp. n. female, from *Aechmea* sp. in Florida (CD3688). (**A**): Entire body; (**B**): Anterior body region; (**C**): Female vulval region and genital tract; (**D**–**G**): Posterior body; and (**H**): Lateral field.

**Figure 6 plants-12-02770-f006:**
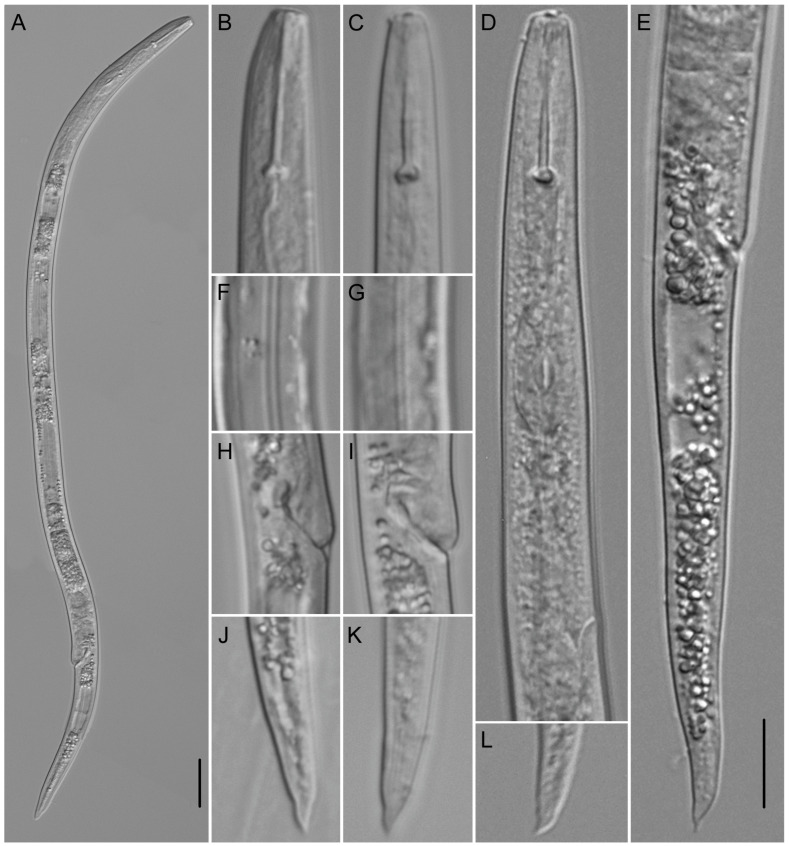
Light microscopic photos of *Paratylenchus hawaiiensis* sp. n. female from *Neoregelia* sp., Hawaii (CD619; (**A**–**C**,**F**–K)) and from *Aechmea* sp., Florida (CD3688; (**D**,**E**,**L**)). (**A**): Entire body; (**B**–**D**): Anterior body region; (**F**,**G**): Lateral field of marked by two bands (three incisures) at mid-body; (**H,I**): Vulva region; (**E**): Posterior body region; and (**J**–**L**): Tail region (Scale bars: (**A**) = 20 μm; (**B**–**L**) = 10 μm).

**Figure 12 plants-12-02770-f012:**
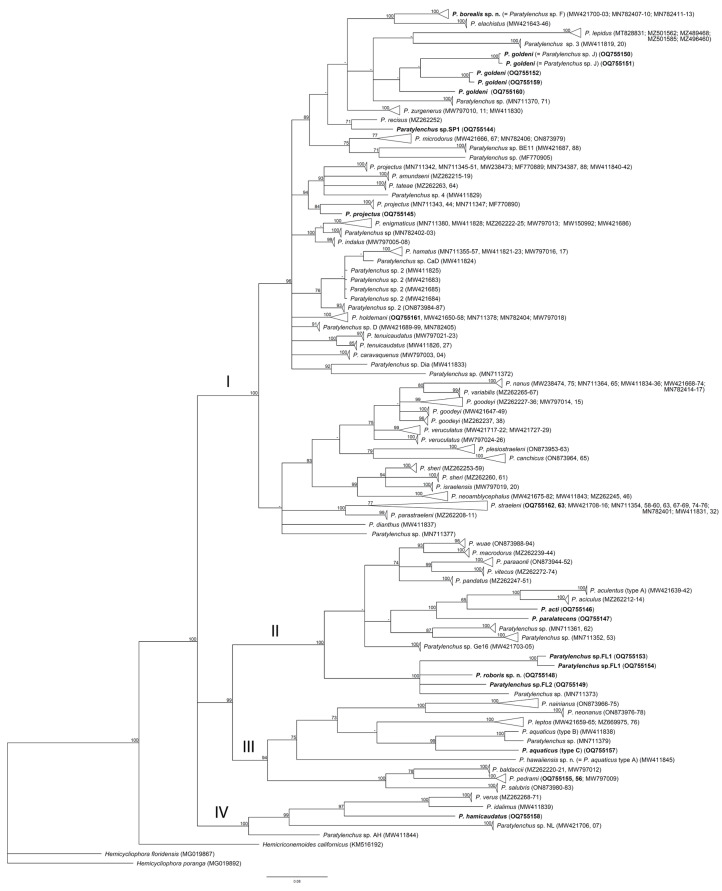
Phylogenetic relationships of *Paratylenchus* species as inferred from Bayesian analysis using the *COI* mtDNA gene sequences under the GTR + I + G model. Posterior probability more than 70% is given for appropriate clades. New sequences are indicated in bold.

**Figure 13 plants-12-02770-f013:**
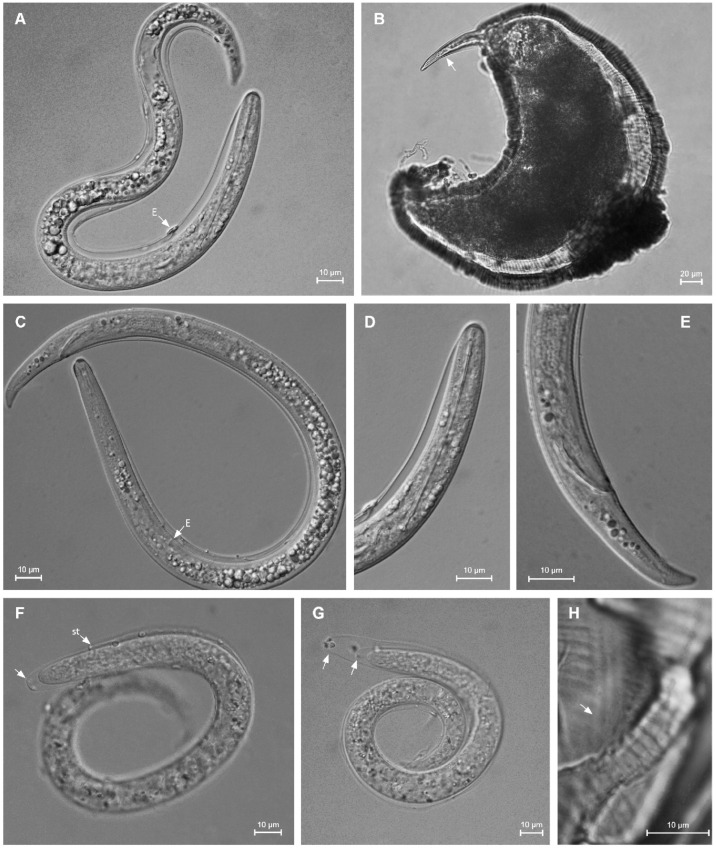
Light microscopic photos of *Paratylenchus acti* from *Andropogon virginicus* in Florida. (**A**): Entire body of vermiform female. Note excretory pore (arrowed); (**B**): Headless entire body of a dead swollen female. Note the sickle-like shape of the post-vulvar body (arrowed); (**C**): Entire body of male. Note excretory pore (arrowed); (**D**): Anterior body of vermiform female; (**E**): Posterior body of male; and (**F**): Third-stage juvenile encased in the molted cuticle of the second-stage juvenile. Note the molted stylet (st) of the J2 adhering to its molted cuticle (arrowed); (**G**): Fourth-stage juvenile encased in the molted cuticles (arrowed) of J2 and J3; and (**H**): Enlarged section of the cuticle of a swollen female showing the lateral field (arrowed) marked by four incisures or three bands.

**Figure 19 plants-12-02770-f019:**
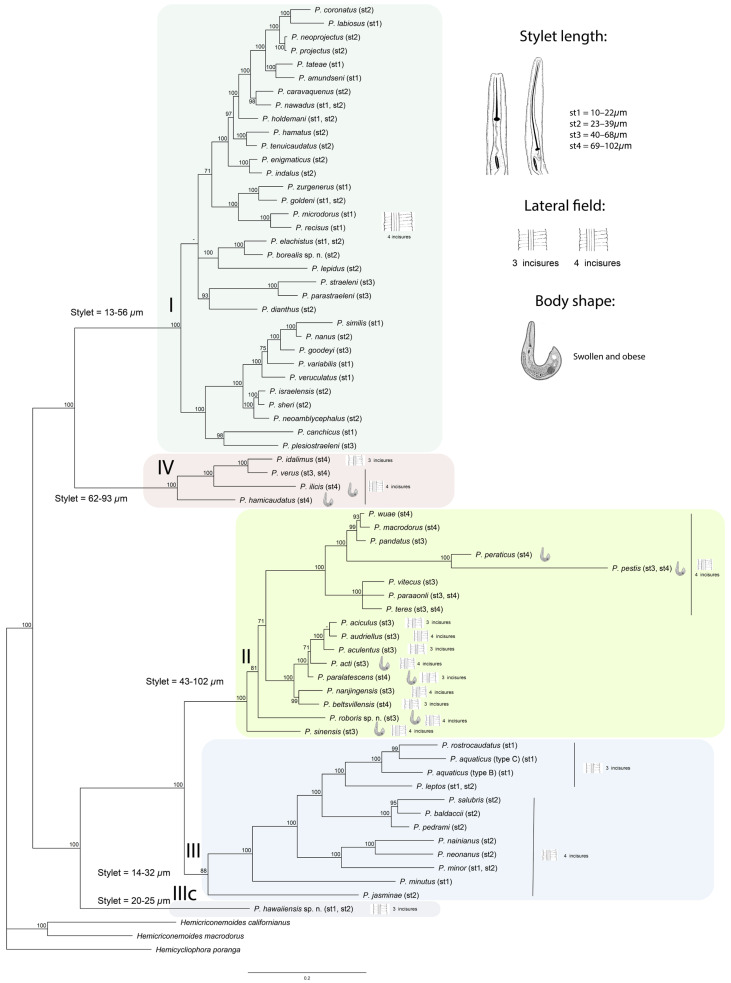
Bayesian 50% majority rule consensus tree as inferred from combined ITS rRNA, D2–D3 expansion segments of 28S rRNA, and *COI* mtDNA gene sequence alignment under the GTR + I + G model. Posterior probability more than 70% is given for appropriate clades.

**Table 1 plants-12-02770-t001:** Species and populations of the genus *Paratylenchus* characterized in the present study.

Species	Location	Host	Sample Code	GenBank Accession Number			Source
				D2–D3 of 28S rRNA	ITS rRNA	*COI*	
*Paratylenchus acti*	Avon Park, Highlands County, FL, USA	*Andropogon virginicus*	CD3566, N21-01034-2	OQ749957	OQ749711	OQ755146	R.N. Inserra & J. D. Stanley
*P. aquaticus* type C	Okeechobee, Okeechobee County, FL, USA	*Stenotaphrum secundatum*	CD3480, N21-00445	OQ749952	OQ749709	OQ755157	R.N. Inserra
*P. aquaticus* type C	Immokalee, Collier County, FL, USA	*Zoysia* sp.	CD3707, N22-00239	OQ749953	-	-	R.N. Inserra
*P. borealis* sp. n.	Anchorage, AK, USA	Unknown plants	CD3781	OQ749978	OQ749710	-	S.A. Subbotin
*P. goldeni*	Madison, Madison County, FL, USA	*Zoysia* sp.	CD3651, N21-01426-4	OQ749968	OQ749702	OQ755160	J. D. Stanley
*P. goldeni*	Morriston, Levy County, FL, USA	*Ilex* sp.	CD3443, N21-00015-4	OQ749969	OQ749700	OQ755152	R.N. Inserra
*P. goldeni*	Milton, Santa Rosa County, FL, USA	*Enydra* sp.	CD3453, N21-00238	OQ749970	OQ749701	OQ755159	R.N. Inserra
*P. goldeni*	Lewis Creek Trail, Madera County, CA, USA	Unknown plants	CD3470	OQ749967	-	-	S.A. Subbotin
*P. goldeni*	Skokomish, Mason County, WA, USA	Unknown plant	CD3216	MW413570	-	OQ755151	S.A. Subbotin
*P. goldeni*	Oakland, Douglas County, OR, USA	Unknown plant	CD3220	MW413569	-	OQ755150	S.A. Subbotin
*P. hamatus*	Kern County, CA, USA	Peach	CD3584	OQ749965	-	-	S.A. Subbotin
*P. hamatus*	Kern County, CA, USA	Grapevine	CD3625	OQ749963	-	-	S.A. Subbotin
*P. hamatus*	San Jose, Santa Clara County, CA, USA	Ornamental pear	CD3628	OQ749964	OQ749712	-	S.A. Subbotin
*P. hamicaudatus*	Lagunitas Lake (type locality), San Rafael, Marin County, CA, USA	*Sequoia sempervirens*	CD3447a,b	OQ749973, OQ749974	OQ749705	OQ755158	S.A. Subbotin
*P. hawaiiensis* sp. n.	Princeton, Miami-Dade County, FL, USA	Bromeliad (*Aechmea* sp.)	CD3688	OQ749977	-	-	S.A. Subbotin
*P. holdemani*	Marin County, CA, USA	Unknown plants	CD3469	OQ749960	-	-	S.A. Subbotin
*P. holdemani*	Área Recreativa “El Sotillo”, Villaviciosa de Odón, Madrid, Spain	*Quercus ilex*	CD3527, CD3528	OQ749959	OQ749713	OQ755161	S. Álvarez-Ortega
*P. minutus*	Kalaheo, Kauai Island, HI, USA	*Coffea* sp.	CD3538, N21-01001	OQ749976	OQ749706	-	K.H. Wang
*P. minutus*	Jasper, Hamilton County, FL, USA	*Hemerocallis* sp.	CD3465, N21-00389-1	OQ749975	OQ749707	-	R.N. Inserra
*P. paralatescens*	Gainesville, Alachua County, FL, USA	*Phyllostachys nigra*	CD3396, N20-01262-1	OQ749958	-	OQ755147	R.N. Inserra
*P. pedrami*	Yolo County, CA, USA	Grapevine	CD3609	OQ749971	OQ749708	OQ755156	S.A. Subbotin
*P. pedrami*	San Jose, Santa Clara County, CA, USA	Unknown plants	CD3461	OQ749950	-	OQ755155	S.A. Subbotin
*P. projectus*	Canada	*Fragaria* x *ananassa*	CD3623, N21-01275-1	OQ749962	-	-	R.N. Inserra
*P. projectus*	Villamanta, Madrid, Spain	Grape field	PP329	OQ749961	OQ749699	OQ755145	S. Álvarez-Ortega
*P. roboris* sp. n.	High Springs, Alachua County, FL, USA	*Q. virginiana*	CD3450, N21-00211	OQ749956	OQ749714	OQ755148	R.N. Inserra
*P. straeleni*	Ocala, Marion County, FL, USA	*Q. virginiana*	CD3633, N21-01260-3	OQ749971	OQ749704	OQ755162	R.N. Inserra
*P. straeleni*	Turnbull Hammock, Volusia County, FL, USA	*Rapidhophyllum hystrix*	CD3708, N22-00084	OQ749972	OQ749703	OQ755163	R.N. Inserra
*Paratylenchus* sp.FL1	Fort Lauderdale, Broward County, FL, USA	*Cynodon dactylon*	CD3637, N21-01410	OQ749954	OQ749716	OQ755153	R.N. Inserra
*Paratylenchus* sp.FL1	Punta Gorda, Charlotte County, FL, USA	Unidentified plant from family Palmae	CD3706, N22-00102	OQ749955	-	OQ755154	R.N. Inserra
*Paratylenchus* sp.FL2	Oviedo, Seminole County, FL, USA	Unidentified plant from family Palmae	CD3750, N22-00888	OQ749979	OQ749715	OQ755149	R.N. Inserra
*Paratylenchus* sp.SP1	Villanueva de Perales, Madrid, Spain	*Salix* sp.	PP317A	OQ749966	OQ749717	OQ755144	S. Álvarez-Ortega

**Table 3 plants-12-02770-t003:** Morphometrics of *Paratylenchus hawaiiensis* sp. n. from bromeliad in Florida compared to the population of *P. aquaticus* type A *apud* Van den Berg et al. [30] from bromeliad in Hawaii.

Reference	Van den Berg et al. [30]	This Study		Total Range
Population	Hawaii, USA (CD619)	Florida, USA (CD3688)		
Character	♀ (Fixed)	♀ (Fixed)		♀
		Holotype	Paratypes	
*n*	5	1	4	10
L	366 ± 27.1 (342–409)	362	368.6 ± 20.0 (339.0–383.0)	339–409
Stylet length (St)	21 ± 1 (20–23)	20.5	22.8 ± 2.0 (20.0–24.7)	20–25
Conus length	13 ± 0.8 (12–14)	12.5	14.0 ± 1.1 (12.5–14.8)	12–15
Stylet shaft + knob height	8 ± 0.4 (8–9)	8.0	8.8 ± 1.0 (7.5–9.9)	7.5–10.0
Knob width	3.0	3.0	4.1 ± 0.1 (4.0–4.2)	3.0–4.0
Knob height	2 ± 0.4 (1.5–2.0)	2.0	2.6 ± 0.3 (2.3–2.9)	1.5–3.0
DGO	-	4.0	4.0 ± 0.1 (3.8–4.0)	4.0
Median bulb valve length	-	5.0	5.0 ± 0.0 (5.0–5.0)	5.0
Median bulb width	-	7.0	8.3 ± 0.8 (7.5–9.4)	7.0–9.5
Isthmus length	-	22.0	18.7 ± 1.1 (17.8–20.0)	18–22
Pharynx length	90 ± 1.9 (89–92)	87	91.5 ± 5.4 (84.1–97.0)	84–97
Anterior end to excretory pore (Ep)	68 ± 3.1 (63.5–72)	68	75.4 ± 4.5 (70.3–81.2)	63–81
Max. body width	12 ± 1.1 (11–14)	12.0	13.6 ± 1.3 (12.0–15.0)	11–15
Body width at vulva	-	10.5	11.0 ± 1.4 (9.5–12.3)	9.5–12.5
Body width at anus	-	7.5	8.8 ± 0.7 (8.1–9.4)	7.5–9.5
Anterior end to median bulb base	-	51	56.1 ± 3.5 (53.4–61.3)	51–61
Lateral field width	1.8 ± 0.3 (1.5–2.0)	2.0	2.1 ± 0.1 (2.0–2.3)	1.5–2.5
Anterior end-vulva distance	-	298	300.7 ± 13.5 (281.0–311.8)	281–312
Vulva-tail terminus distance	-	64	67.9 ± 6.6 (58.0–71.3)	58–71
Genital tract length	-	115	118.6 ± 26.0 (85.0–143.5)	85–144
Vulva-anus distance	42 ± 6 (35–51)	35.5	39.8 ± 4.4 (33.5–43.5)	33–51
Tail length	27 ± 2.2 (25.5–31)	28.0	28.1 ± 2.7 (24.5–30.5)	24–31
PERCENTAGES				
*V*	81 ± 1.2 (79–82)	82	81.4 ± 0.5 (81.0–82.0)	79–82
G or T	-	32	32.3 ± 7.9 (22.6–39.5)	23–40
St/L	-	5.7	6.2 ± 0.8 (5.3–7.3)	5.5–7.5
Ep/L	18.6 ± 0.8 (17.6–19.9)	18.8	20.5 ± 1.5 (18.6–22.1)	18–22
RATIOS				
*a*	30.6 ± 3.8 (24.8–34.8)	30.0	27.4 ± 3.8 (22.6–31.2)	23–35
*b*	4.2 ± 0.2 (4.0–4.4)	4.2	4.0 ± 0.3 (3.7–4.5)	3.7–4.5
*c*	13.4 ± 0.7 (12.3–14.1)	12.5	13.1 ± 0.8 (12.2–13.8)	12–14
*c′*	3.6 ± 0.4 (3.1–4.1)	4.1	3.2 ± 0.5 (2.6–3.7)	2.6–4.1

**Table 7 plants-12-02770-t007:** Morphometrics of females and males of *Paratylenchus acti* associated with broomsedge (*Andropogon virginicus*) in Florida compared to the type population of the species from Sakhalin Island, Russia, by Eroshenko [15].

Population	Avon Park, Highlands County (CD3566—N21-01034)		Paratypes of *P. acti* from Sakhalin Island Eroshenko [15]	
Character	Vermiform ♀ (Live, Molted Cuticle Attached)	Vermiform ♂ (Live, Molted Cuticle Attached)	♀	♂
*n*	8	4	11	2
L	289.9 ± 13.8 (277.0–311.0)	299.0 ± 6.2 (291.0–305.9)	240–280	250
Stylet length (St)	60.8 ± 2.5 (55.5–64.3)	-	56–62	-
Conus length	55.0 ± 2.2 (50.0–57.4)	-	-	-
Stylet shaft + knob height	5.8 ± 0.8 (4.9–7.0)	-	-	-
Knob width	2.9 ± 0.9 (2.5–3.3)	-	-	-
Knob height	1.5 ± 0.1 (1.3–1.7)	-	-	-
DGO	5.5 ± 0.8 (5.0–7.0)	-	-	-
Median bulb valve length	9.7 ± 0.3 (9.4–10.2)	-	-	-
Median bulb width	8.7 ± 0.8 (7.5–9.8)	-	-	-
Isthmus length	15.0 ± 1.7 (12.0–17.5)	-	-	-
Pharynx length	109.5 ± 4.4 (104.0–114.8)	88.1 ± 5.1 (82.1–96.0)	-	-
Anterior end to excretory pore (Ep)	69.7 ± 5.1 (60.5–77.2)	77.8 ± 2.9 (74.2–82.1)	-	-
Max body width	12.8 ± 0.3 (12.3–13.3)	12.6 ± 0.4 (12.0–13.0)	-	-
Body width at vulva	11.3 ± 0.8 (10.0–12.1)	-	-	-
Body width at anus	6.9 ± 0.3 (6.4–7.4)	10.0 ± 0.5 (9.4–10.8)	-	-
Anterior end to median bulb base	83.6 ± 2.8 (80.0–87.2)	-	-	-
Lateral field width	-	-	-	-
Anterior end-vulva distance	212.0 ± 9.0 (202.0–228.7)	-	-	-
Vulva-tail terminus distance	79.0 ± 6.3 (71.0–92.4)	-	-	-
Genital tract length	36.6 ± 7.3 (29.0–45.5)	105.9 ± 9.8 (96.0–118.8)	-	-
Vulva-anus distance	56.9 ± 6.3 (50.0–71.2)	-	-	-
Tail length	21.1 ± 1.1 (21.8–24.8)	27.2 ± 1.1 (25.7–28.7)	25	-
Stylet base (Stb) to median bulb valve base (v)	20.4 ± 6.0 (13.0–33.0)	-	-	-
Spicule length	-	17.6 ± 0.2 (17.4–17.8)	-	16–17
Gubernaculum length	-	4.1 ± 0.1 (4.0–4.2)	-	-
PERCENTAGES				
*V*	72.8 ± 1.3 (70.0–74.4)	-	69–73	-
G or T	12.5 ± 2.2 (10.0–16.0)	35.4 ± 3.5 (31.5–40.2)	-	34–35
Stb-v/St	33.6 ± 9.5 (21.2–51.3)	-	-	-
St/L	20.6 ± 1.3 (19.0–22.4)	-	22–24	-
Ep/L	23.6 ± 1.1 (21.8–24.8)	26.0 ± 0.6 (25.4–27.0)	-	-
RATIOS				
*a*	22.5 ± 1.1 (21.0–24.6)	23.7 ± 1.1 (22.3–25.4)	19–23	23
*b*	2.6 ± 0.1 (2.4–2.9)	3.4 ± 0.2 (3.1–3.5)	2.5–2.6	3.5
*c*	13.8 ± 0.8 (12.2–14.7)	11.0 ± 0.3 (10.6–11.3)	10–11	10
*c′*	3.0 ± 0.2 (2.6–3.4)	2.7 ± 0.2 (2.4–3.0)	-	-

## Data Availability

The datasets generated during and/or analyzed during the current study are available NCBI and from the corresponding author upon reasonable request.

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
