# Peer review of "Morphological and Molecular Diversity among Pin Nematodes of the Genus Paratylenchus (Nematoda: Paratylenchidae) from Florida and Other Localities and Molecular Phylogeny of the Genus"

_plants, 2023, doi:10.3390/plants12152770_

Round 1

Reviewer 1 Report

Journal: Plants

Manuscript ID: plants-2426459

Title: Morphological and molecular diversity among pin nematodes of the genus
Paratylenchus (Nematoda: Paratylenchidae) from Florida and other localities,
and molecular phylogeny of the genus

Authors: Sergio Álvarez-Ortega et al.

The manuscript ID plants-2426459 provides a comprehensive examination of the various species of pin nematodes found in Florida and others regions in the United States, Canada and Spain. The authors have identified and described two new species, Paratylenchus hawaiiensis sp. n. and P. roboris sp. n. Additionally, the study confirms the presence of previously described species and clarifies their distribution across different states and countries. The authors use both molecular and morphological methods in species identification. I appreciate a discussion about the validity of the genus Gracilacus and Cacopaurus.  Generally, the paper is well written, presents new data, and mainly fills the gap in our knowledge. I also appreciate that the article will be published as open access, with all figures and morphological and molecular characteristics. I have very few comments and after completion of missing information and correction, I recommend the paper for publishing in Plants.

 Suggestions for authors:

Suggestions shorter title e.g. Diversity and molecular phylogeny of the genus Paratylenchus from United States, Canada and Spain.

Part MM

Line 1634: Samples from other regions in the United States, Canada and Spain were also from soil under ornamental and landscape plants?

Line 1642: Paratylenchus species were picked up from whole samples with other nematode species? Line 1645: You pick up live worms, do not they move very fast? 

Author Response

Dear Reviewer,

All your comments and suggestions are very appreciated and considered. Here are our replies:

Suggestions for authors:

Suggestions shorter title e.g. Diversity and molecular phylogeny of the genus Paratylenchus from United States, Canada and Spain.

Reply: We prefer to keep the current title.

Part MM

Line 1634: Samples from other regions in the United States, Canada and Spain were also from soil under ornamental and landscape plants?

Reply: All the samples characterized in this study are listed in table 1 and in this table we included the information about the host.

Line 1642: Paratylenchus species were picked up from whole samples with other nematode species? Line 1645: You pick up live worms, do not they move very fast?

Reply: Paratylenchus specimens were isolated and picked up individually from the samples which also contains other nematode species. The live nematodes do not move very fast.

Sincerely,

Sergio

Reviewer 2 Report

There are few studies on genus Paratylenchus and even its role in soil biology/ecology and plant pathology is rather unknown. One of the reasons is the ecarce number of nematologists devoted to its study and the need of a reliable taxonomy. Thanks the Integrative taxonomy  the nematologists are constructed a new frame of reference in which the indicated conceptual roles of this genus can be located. The authors have notably experience in Systematic and Taxonomy of Nematodes and many of their works are seminal in Integrative Taxonomy aplied to plant parasytic nematodes. The elections of this Project is also a compromise with the future because of, till now, there few paper sought in a so complete view as this is regarding Paratylenchus. My point of view is that this paper will be very important for those who record Paratylenchus in soil samples because they will have a great review and clear direccions on how to proceed. The paper is made following the canonical procedures, from a conceptual, metodological and technical point of view and results and conclussions are consecuent with it. My recommendation is to publish the paper in its present form

Author Response

Dear Reviewer,

All your comments are very appreciated.

Sincerely,

Sergio